# Advancing Wasserstein Convergence Analysis of Score-Based Models: Insights from Discretization and Second-Order Acceleration

**Yifeng Yu**
Tsinghua University
yyf22@mails.tsinghua.edu.cn

**Lu Yu** *
City University of Hong Kong
lu.yu@cityu.edu.hk

## Abstract

Score-based diffusion models have emerged as powerful tools in generative modeling, yet their theoretical foundations remain underexplored. In this work, we focus on the Wasserstein convergence analysis of score-based diffusion models. Specifically, we investigate the impact of various discretization schemes, including Euler discretization, exponential integrators, and midpoint randomization methods. Our analysis provides the first quantitative comparison of these discrete approximations, emphasizing their influence on convergence behavior. Furthermore, we explore scenarios where Hessian information is available and propose an accelerated sampler based on the local linearization method. We establish the first Wasserstein convergence analysis for such a Hessian-based method, showing that it achieves an improved convergence rate of order $\widetilde{\mathcal{O}}(\sqrt{d}/\varepsilon)$, which significantly outperforms the standard rate $\widetilde{\mathcal{O}}(d/\varepsilon^2)$ of vanilla diffusion models. Numerical experiments on synthetic data and the MNIST dataset validate our theoretical insights.

## 1  Introduction

*Diffusion models* have become a pivotal framework in modern generative modeling, achieving notable success across fields such as image generation [11, 17, 31, 34], natural language processing [30], and computational biology [1, 42]. These models add noise to data via a forward process and learn to reverse it, reconstructing data from noise. This approach enables them to capture the underlying structure of complex, high-dimensional data distributions. For a detailed review of diffusion models, we refer the readers to [6, 39, 43].

A widely adopted formulation of diffusion models is the score-based generative model (SGM), implemented using stochastic differential equations (SDEs) [36]. Broadly speaking, SGMs rely on two key stochastic processes: a forward process that gradually transforms data samples into pure noise, and a backward process that reverses this transformation, recovering the target data distribution from noise.

Despite the remarkable empirical success of diffusion models across various applications, their theoretical understanding remains limited. In recent years, there has been a rapidly expanding body of research on the convergence theory of diffusion models. Generally, these contributions can be divided into two main approaches, each focusing on different metrics and divergences. The first category investigates convergence bounds based on $\alpha$-divergence, including the Kullback–Leibler (KL) divergence and the total variation (TV) distance (see e.g., [5, 7, 8, 22, 26, 41]). Among these works, several explore acceleration techniques that leverage higher-order information about the log density (see e.g., [19, 24, 27]). The second category focuses on convergence bounds in Wasserstein

---

*Corresponding author

39th Conference on Neural Information Processing Systems (NeurIPS 2025).

distance, which is often considered more practical and informative for estimation tasks. One line of work within this category assumes strong log-concavity of the data distribution and access to accurate estimates of the score function [3, 13, 37, 38]. Another line of work focuses on specific structural assumptions of the data distribution [10, 14, 29].

Much of the existing literature on the convergence theory of diffusion models relies on the Euler discretization method. Notably, [5] compare the behavior of Euler discretization and exponential integrators [18, 47] in terms of KL divergence. Additionally, [10] provide a comparative analysis of these two schemes, though without formal theoretical guarantees. A comprehensive and systematic understanding of how different discretization schemes influence convergence performance in diffusion models remains underexplored. Furthermore, while convergence analyses of accelerated diffusion models primarily focus on TV or KL distances, studies investigating Wasserstein convergence for these accelerations remain lacking[2].

In this work, we address these challenges by analyzing the Wasserstein convergence of score-based diffusion models when the data distribution has a smooth and strongly log-concave density. Specifically, we investigate the impact of different discretization schemes on convergence behavior. Beyond the widely used Euler method and exponential integrator, we explore the midpoint randomization method. This method was initially introduced in [32] for discretizing kinetic Langevin diffusion [9] and then has been extensively studied in log-concave sampling complexity theory [16, 21, 44–46]. It was later applied to diffusion models [15, 23], showing improved KL and TV convergence performance over vanilla models and offering easy parallelization.

We also consider scenarios where accurate estimates of the Hessian of the log density are accessible. Inspired by [33], we propose a novel sampler based on the *local linearization* method. Our analysis shows that this approach significantly accelerates convergence in Wasserstein distance.

Our contribution can be summarized as follows.

- We establish convergence guarantees for SGMs in the Wasserstein-2 distance under various discretization methods, including the Euler method, exponential integrators, the midpoint randomization method, and a hybrid approach combining the latter two.

- We introduce a novel Hessian-based accelerated sampler for the stochastic diffusion process, leveraging the local linearization method. We then establish its Wasserstein convergence analysis in Theorem 4, achieving state-of-the-art order of $\widetilde{\mathcal{O}}(\sqrt{d}/\varepsilon)$.

- Section 5 compares the performance of SGMs under four discretization schemes and the proposed Hessian-based method on both synthetic data and the MNIST dataset. The results align with our theory and highlight the acceleration of the proposed second-order method.

In summary, our analysis provides a quantitative comparison of different discrete approximations, offering practical guidance for choosing discretization. Moreover, we present the first Wasserstein convergence analysis of an accelerated sampler that leverages the second-order information about log-densities. This accelerated sampler achieves a faster convergence rate $\widetilde{\mathcal{O}}(\sqrt{d}/\varepsilon)$, compared to the standard rate $\widetilde{\mathcal{O}}(d/\varepsilon^2)$ of vanilla diffusion models. These results contribute to the understanding of Wasserstein convergence in score-based models, shedding light on aspects that have not been extensively explored before.

**Notation.** Let $\mathbb{R}^d$ be the $d$-dimensional Euclidean space and $I_d$ the identity matrix. The gradient and the Hessian of a function $f : \mathbb{R}^d \to \mathbb{R}$ are denoted by $\nabla f$ and $\nabla^2 f$. Given any pair of measures $\mu$ and $\nu$, the Wasserstein-2 distance between $\mu$ and $\nu$ is defined as

$$\mathsf{W}_2(\mu, \nu) = \left( \inf_{\varrho \in \Gamma(\mu, \nu)} \int_{\mathbb{R}^d \times \mathbb{R}^d} \|x - y\|^2 \, \mathrm{d}\varrho(x, y) \right)^{1/2},$$

where the infimum is taken over all joint distributions $\varrho$ that have $\mu$ and $\nu$ as marginals. For two symmetric $d \times d$ matrices $A$ and $B$, we use $A \preccurlyeq B$ or $B \succcurlyeq A$ to denote the relation that $B - A$ is positive semi-definite. For any random vector $X$, let $\mathcal{L}(X)$ denote its law, and define $\|X\|_{\mathbb{L}_2} := \sqrt{\mathbb{E}[\|X\|^2]}$, where $\|\cdot\|$ denotes the Euclidean norm. For any random matrix $A$, define $\|A\|_{\mathbb{L}_2} := \sqrt{\mathbb{E}[\|A\|_2^2]}$, where $\|\cdot\|_2$ denotes the matrix 2-norm.

---

[2]Due to space constraints, we defer the discussion of related theoretical advancements in accelerating samplers for diffusion models to Appendix A.

## 2 Background and Our Setting

**Framework.** We consider the forward process

$$\mathrm{d}X_t = f(X_t, t)\,\mathrm{d}t + g(X_t, t)\,\mathrm{d}B_t\,, \tag{1}$$

where the initial point $X_0 \sim p_0$ follows the data distribution, and $B_t$ denotes the standard $d-$dimensional Brownian motion. Here, the drift $f : \mathbb{R}^d \times \mathbb{R}_+ \to \mathbb{R}^d$ and the function $g : \mathbb{R}^d \times \mathbb{R}_+ \to \mathbb{R}^{d \times d}$ are diffusion parameters. Some conditions are necessary to ensure that the SDE (1) is well-defined. In practice, various choices for the pair $(f, g)$ are employed, depending on the specific needs of the model; for a detailed survey, we refer to [39]. For clarity, we adopt the simplest possible choice in this work by setting $f(X_t, t) = -X_t/2$ and $g(X_t, t) = 1$. This results in the Ornstein-Uhlenbeck process, which is described by the following SDE:

$$\mathrm{d}X_t = -\frac{1}{2}X_t\,\mathrm{d}t + \,\mathrm{d}B_t\,. \tag{2}$$

The forward process (2) is run until a sufficiently large time $T > 0$, at which point the corrupted marginal distribution of $X_T$, denoted by $p_T$, is expected to approximate the standard Gaussian distribution. Then, diffusion models generate new data by reversing the SDE (2), which leads to the following backward SDE

$$\mathrm{d}X_t^{\leftarrow} = \frac{1}{2}\left(X_t^{\leftarrow} + 2\nabla \log p_{T-t}(X_t^{\leftarrow})\right)\,\mathrm{d}t + \,\mathrm{d}W_t\,, \tag{3}$$

where $X_0^{\leftarrow} \sim p_T$, and the term $\nabla \log p_t$, referred to as the *score function* for $p_t$, is represented by the gradient of the log density function of $p_t$. Additionally, $W_t$ denotes another standard Brownian motion independent of $B_t$. Under mild conditions, when initialized at $X_0^{\leftarrow} \sim p_T$, the backward process $\{X_t^{\leftarrow}\}_{0 \leqslant t \leqslant T}$ has the same distribution as the forward process $\{X_{T-t}\}_{0 \leqslant t \leqslant T}$ [2, 4]. As a result, running the reverse diffusion $X_t^{\leftarrow}$ from $t = 0$ to $T$ will generate a sample from the target data distribution $p_0$. Note that the density $p_T$ is unknown; we approximate it using the distribution

$$\hat{p}_T = \mathcal{N}(0, (1 - e^{-T})I_d)$$

as proposed in [13]. Therefore, we derive a reverse diffusion process defined by

$$\mathrm{d}Y_t = \frac{1}{2}(Y_t + 2\nabla \log p_{T-t}(Y_t))\,\mathrm{d}t + \,\mathrm{d}W_t, \quad Y_0 \sim \hat{p}_T\,. \tag{4}$$

**Score Matching.** Another challenge in working with (3) is that the score function $\nabla \log p_t$ is unknown, as the distribution $p_t$ is not explicitly available. In practice, rather than using the exact score function $\nabla \log p_{T-t}$, approximate estimates for it are learned from the data by training neural networks on a score-matching objective [20, 35, 40]. This objective is given by

$$\underset{\theta \in \Theta}{\text{minimize}} \quad \mathbb{E}[\|s_\theta(t, X_t) - \nabla \log p_t(X_t)\|^2]\,,$$

where $\{s_\theta : \theta \in \Theta\}$ is a sufficiently rich function class, such as that of neural network. Substituting the learned score estimate $s_*$ into the backward process (3), we obtain the following practical continuous-time backward SDE,

$$dX_t^{\leftarrow} = \frac{1}{2}\left(X_t^{\leftarrow} + 2s_*(T - t, X_t^{\leftarrow})\right)\,\mathrm{d}t + \,\mathrm{d}W_t\,. \tag{5}$$

Since this continuous backward SDE cannot be simulated exactly, it is typically approximated using discretization methods.

**Discretization Schemes.** We outline the four discretization methods considered in this work for solving the practical reverse SDE (5). Let $h > 0$ be the step size. Without loss of generality, we assume $T = Nh$, where $N$ is a positive integer. For simplicity, we denote $\frac{1}{2}X_t^{\leftarrow} + s_*(T - t, X_t^{\leftarrow})$ by $\gamma(T - t, X_t^{\leftarrow})$, and define

$$\Delta_h W_t := W_{t+h} - W_t, \quad \bar{\Delta}_h W_t := \int_t^{t+h} e^{\frac{t+h-s}{2}}\,\mathrm{d}W_s\,.$$

- **EULER-MARUYAMA SCHEME:** Given the step size $h$, the following approximation holds

$$X_{t+h}^{\leftarrow} = X_t^{\leftarrow} + \int_0^h \gamma(T - (t + v), X_{t+v}^{\leftarrow})\,\mathrm{d}v + \Delta_h W_t \approx X_t^{\leftarrow} + h\gamma(T - t, X_t^{\leftarrow}) + \Delta_h W_t\,.$$

We derive the following discretized process for $n = 0, \ldots, N-1$:

$$\vartheta_{n+1}^{\mathsf{EM}} = (1 + h/2)\vartheta_n^{\mathsf{EM}} + hs_*(T - nh, \vartheta_n^{\mathsf{EM}}) + \sqrt{h}\xi_n,$$

where $\vartheta_0^{\mathsf{EM}} \sim \hat{p}_T$ and $\xi_n \sim \mathcal{N}(0, I_d)$.

- **EXPONENTIAL INTEGRATOR**: Inspired by the work [18], [47] propose a more refined discretization method which solves the backward SDE (3) explicitly, yielding the following approximation

$$X_{t+h}^{\leftarrow} = e^{\frac{h}{2}}X_t^{\leftarrow} + \int_0^h e^{\frac{h-v}{2}}s_*(T - t - v, X_{t-v}^{\leftarrow})\,\mathrm{d}v + \bar{\Delta}_h W_t$$

$$\approx e^{\frac{h}{2}}X_t^{\leftarrow} + 2(e^{\frac{h}{2}} - 1)s_*(T - t, X_t^{\leftarrow}) + \bar{\Delta}_h W_t.$$

We derive the following discretized process for $n = 0, \ldots, N-1$:

$$\vartheta_{n+1}^{\mathsf{EI}} = e^{\frac{h}{2}}\vartheta_n^{\mathsf{EI}} + 2(e^{\frac{h}{2}} - 1)s_*(T - nh, \vartheta_n^{\mathsf{EI}}) + \sqrt{e^h - 1}\xi_n$$

with the initial point $\vartheta_0^{\mathsf{EI}} \sim \hat{p}_T$ and $\xi_n \sim \mathcal{N}(0, I_d)$.

- **VANILLA MIDPOINT RANDOMIZATION**: Unlike the Euler method, the midpoint randomization method evaluates the function $\gamma(T - t, X_t^{\leftarrow})$ at a random point within the time interval $[0, h]$ rather than at the start. Let $U$ be a random variable uniformly distributed in $[0, 1]$ and independent of the Brownian motion $W_t$. The randomized midpoint method exploits the approximation

$$\begin{aligned} X_{t+h}^{\leftarrow} &= X_t^{\leftarrow} + \int_0^h \gamma(T - t - v, X_{t+v}^{\leftarrow})\,\mathrm{d}v + \Delta_h W_t \\ &\approx X_t^{\leftarrow} + h\gamma(T - t - hU, X_{t+hU}^{\leftarrow}) + \Delta_h W_t. \end{aligned} \tag{6}$$

The idea behind the randomized midpoint method is to introduce an $U$ in $[0, 1]$, making $h\gamma(T - t - hU, X_{t+hU}^{\leftarrow})$ an estimator for integral $\int_0^h \gamma(T - t - v, X_{t+v}^{\leftarrow})\,\mathrm{d}v$.

Furthermore, the intermediate term $X_{t+hU}^{\leftarrow}$ is generated by employing the Euler method. We then derive the following discretized process for $n = 0, \ldots, N-1$:

**Step 1** Generate $\xi_n', \xi_n'' \sim \mathcal{N}(\mathbf{0}, I_d)$ and $U_n \sim \mathrm{Unif}[0, 1]$. Set $\xi_n = \sqrt{U_n}\xi_n' + \sqrt{1 - U_n}\xi_n''$.

**Step 2** With the initialization $\vartheta_0^{\mathsf{REM}} \sim \hat{p}_T$, define

$$\begin{aligned} \vartheta_{n+U}^{\mathsf{REM}} &= \vartheta_n^{\mathsf{REM}} + hU_n\gamma(T - nh, \vartheta_n^{\mathsf{REM}}) + \sqrt{hU_n}\xi_n', \\ \vartheta_{n+1}^{\mathsf{REM}} &= \vartheta_n^{\mathsf{REM}} + h\gamma(T - (n + U_n)h, \vartheta_{n+U}^{\mathsf{REM}}) + \sqrt{h}\xi_n. \end{aligned}$$

- **EXPONENTIAL INTEGRATOR WITH MIDPOINT RANDOMIZATION**: Combining midpoint randomization with the exponential integrator approach, we propose the following new discretization process for $n = 0, \ldots, N-1$:

**Step 1** Generate $\xi_n', \xi_n'' \sim \mathcal{N}(\mathbf{0}, I_d)$ and $U_n \sim \mathrm{Unif}[0, 1]$. Set $\xi_n = \rho_n\xi_n' + \sqrt{1 - \rho_n^2}\xi_n''$ with

$$\rho_n = e^{\frac{h(1+U_n)}{2}}\left(1 - e^{-hU_n}\right)\left[(e^{hU_n} - 1)(e^h - 1)\right]^{-1/2}.$$

**Step 2** With the initialization $\vartheta_0^{\mathsf{REI}} \sim \hat{p}_T$, define

$$\begin{aligned} \vartheta_{n+U}^{\mathsf{REI}} &= e^{hU_n/2}\vartheta_n^{\mathsf{REI}} + 2(e^{hU_n/2} - 1)s_*(T - nh, \vartheta_n^{\mathsf{REI}}) + \sqrt{e^{hU_n} - 1}\xi_n', \\ \vartheta_{n+1}^{\mathsf{REI}} &= e^{h/2}\vartheta_n^{\mathsf{REI}} + he^{(1-U_n)h/2}s_*(T - (n + U_n)h, \vartheta_{n+U}^{\mathsf{REI}}) + \sqrt{e^h - 1}\xi_n. \end{aligned}$$

The resulting discrete process is then solved to generate new samples that approximately follow the data distribution $p_0$.

## 3   Wasserstein Convergence Analysis under Various Discretization Schemes

In this section, we study the convergence of diffusion models under EM, EI, REM and REI discretizations of the continuous backward SDE (3). Specifically, we establish the upper bounds on

the Wasserstein-2 distance between the distribution of the $N$-th output of the SGMs under these discretization schemes and the target distribution

$$W_2(\mathcal{L}(\vartheta_N^\alpha), p_0), \quad \alpha \in \{\mathsf{EM}, \mathsf{EI}, \mathsf{REM}, \mathsf{REI}\}.$$

Additionally, we analyze the number of iterations $N$ required for the Wasserstein distance to achieve a pre-specified error level $\varepsilon$ under each discretization scheme. For clarity of presentation, we omit constants in the main text, retaining only the key components that affect convergence rates. Full bounds with constants are provided in the proofs.

To establish the convergence analysis, we require the following assumption on target density $p_0$.

**Assumption 1.** *The target density $p_0$ is $m_0$-strongly log-concave, and the score function $\nabla \log p_0$ is $L_0$-Lipschitz.*

Under Assumption 1, $p_t$ is $m(t)$-strongly log-concave, $\nabla \log p_t$ is $L(t)$-Lipschitz. Moreover, $m(t)$ is lower bounded by $m_{\min} = \min(1, m_0)$, and $L(t)$ is upper bounded by $L_{\max} = 1 + L_0$, as summarized in Lemma 6 and 7 (see Appendix B). We also assume that the score function $\nabla \log p_t(x)$ exhibits linear growth in $\|x\|$, as stated below.

**Assumption 2.** *There exists a constant $M_1 > 0$ such that for $n = 0, 1, \ldots, N-1$, it holds that*

$$\sup_{nh \leqslant t, s \leqslant (n+1)h} \|\nabla \log p_{T-t}(x) - \nabla \log p_{T-s}(x)\| \leqslant M_1 h(1 + \|x\|), \quad \forall x.$$

The above condition is a relaxation of the standard Lipschitz condition on the score function. Moreover, we require the following assumption on the score-matching approximation at each point $\vartheta_n$.

**Assumption 3.** *Given a small $\varepsilon_{sc} > 0$, the score estimator satisfies*

$$\sup_{0 \leqslant n \leqslant N} \|\nabla \log p_{T-nh}(\vartheta_n) - s_*(T - nh, \vartheta_n)\|_{\mathbb{L}_2} \leqslant \varepsilon_{sc}.$$

Assumption 1, 2 and 3 are standard in the Wasserstein convergence analysis of the score-based diffusion model. These assumptions were previously adopted in [12, 13] and can be easily verified in the Gaussian case.

## 3.1 Euler-Maruyama Method and Exponential Integrator

In the following theorem, we quantify the Wasserstein distance between the distribution of $\vartheta_N^\alpha, \alpha \in \{\mathsf{EM}, \mathsf{EI}\}$ and the target distribution $p_0$.

**Theorem 1.** *Suppose that Assumptions 1, 2 and 3 hold, it holds that*

$$W_2(\mathcal{L}(\vartheta_N^\alpha), p_0) \lesssim e^{-m_{\min}T} \|X_0\|_{\mathbb{L}_2} + \mathscr{C}_1^\alpha \sqrt{dh} + \mathscr{C}_2^\alpha \varepsilon_{sc}, \quad \alpha \in \{\mathsf{EM}, \mathsf{EI}\}, \tag{7}$$

$$\text{where} \quad \mathscr{C}_1^{\mathsf{EM}} = \frac{L_{\max} + 1/2}{m_{\min} - 1/2} \quad \text{and} \quad \mathscr{C}_2^{\mathsf{EM}} = \frac{1}{m_{\min} - 1/2},$$

$$\mathscr{C}_1^{\mathsf{EI}} = \frac{L_{\max}}{m_{\min} - 1/2} \quad \text{and} \quad \mathscr{C}_2^{\mathsf{EI}} = \frac{1}{m_{\min} - 1/2}$$

*with $m_{\min} = \min(1, m_0)$ and $L_{\max} = 1 + L_0$.*

Before comparing the above bound with existing ones, we state a direct consequence.

**Corollary 2.** *Given a small $\varepsilon > 0$ and $\varepsilon_{sc} = \mathcal{O}(\varepsilon)$, the Wasserstein distance satisfies $W_2(\mathcal{L}(\vartheta_N^\alpha), p_0) < \varepsilon, \alpha \in \{\mathsf{EM}, \mathsf{EI}\}$ after $N = \mathcal{O}\left(\frac{d}{\varepsilon^2}\log\left(\frac{\sqrt{d}}{\varepsilon}\right)\right)$ iterations, provided that $T = \mathcal{O}\left(\log(\frac{\sqrt{d}}{\varepsilon})\right)$ and $h = \mathcal{O}\left(\frac{\varepsilon^2}{d}\right)$.*

We now explain the upper bound in Theorem 1, which decomposes the total error into three parts: initialization, discretization, and score-matching errors. The **first term** on the right-hand side of display (7) bounds the error arising from initializing the reverse idealized continuous-time SDE (4) at $\hat{p}_T$ instead of $p_T$ The **second term** in (7) captures the discretization error from the discretization scheme, while the **third term** reflects the score matching error.

The term $-1/2$ in $m_{\min} - 1/2$ arises from the drift term of the forward SDE (2), as demonstrated in Lemma 8 in the appendix. We refer interested readers to the appendix for further details. More generally, if the forward SDE takes the form $dX_t = -\beta X_t \, dt + dB_t$, then the bound becomes $m(t) - \beta$, reflecting a dependence on the coefficient $\beta$. Although the current theory still assumes that the data distribution $p_0$ is strongly log-concave, the condition $m_0 > 1/2$ naturally generalizes to $m_0 > \beta$ for any $\beta > 0$.

The convergence rates of EM method obtained in Theorem 1 and Corollary 2 align with Theorem 2 and Proposition 5 in [13]. Moreover, we note that the convergence rate of EM and EI schemes are comparable, which is consistent with the error bounds for these two schemes in KL divergence established in Theorem 1 of [5], where $N = \Omega(T^2)$ in their setting.

## 3.2 Randomized Midpoint Method

Since the randomized midpoint method involves the i.i.d. uniformly distributed random variable $U_n$, the resulting score matching function $s_*(T - t, x)$ can be evaluated at any $t \in [0, T]$. To proceed, we impose a regularity condition on the deviation of the estimated score from the true score at these points. For this, we introduce the following auxiliary stochastic processes. Given $U_n = u$, define the conditional realization of the random vector $\vartheta_{n+U}^{\mathsf{REM}}$ via

$$\vartheta_{n+u}^{\mathsf{REM}} := \left(1 + \frac{uh}{2}\right)\vartheta_n^{\mathsf{REM}} + uh s_*(T - nh, \vartheta_n^{\mathsf{REM}}) + \sqrt{uh}\,\xi_n \,.$$

Similarly, we define the following conditional realization of $\vartheta_{n+U}^{\mathsf{REI}}$

$$\vartheta_{n+u}^{\mathsf{REI}} := e^{uh/2}\vartheta_n^{\mathsf{REI}} + 2(e^{uh/2} - 1)s_*(T - nh, \vartheta_n^{\mathsf{REI}}) + \sqrt{e^{uh} - 1}\,\xi_n' \,.$$

We impose the following assumption on score estimates, which extends Assumption 3.

**Assumption 4.** *There exists a constant $\varepsilon_{sc} > 0$ such that for any $u \in [0, 1]$ and $n = 0, \ldots, N$,*

$$\left\|\nabla \log p_{T-(n+u)h}(\vartheta_{n+u}^{\alpha}) - s_*(T - (n+u)h, \vartheta_{n+u}^{\alpha})\right\|_{\mathbb{L}_2} \leqslant \varepsilon_{sc}, \quad \alpha \in \{\mathsf{REM}, \mathsf{REI}\}.$$

In what follows, we provide the upper bound for the Wasserstein-2 distance between the law of $\vartheta_N^{\alpha}, \alpha \in \{\mathsf{REM}, \mathsf{REI}\}$ and the data distribution $p_0$.

**Theorem 3.** *Suppose that Assumptions 1, 2 and 4 hold, then for $\alpha \in \{\mathsf{REM}, \mathsf{REI}\}$,*

$$\mathsf{W}_2(\mathcal{L}(\vartheta_N^{\alpha}), p_0) \lesssim e^{-m_{\min}T}\|X_0\|_{\mathbb{L}_2} + \mathscr{C}_1^{\alpha}(d)\sqrt{h} + \mathscr{C}_2^{\alpha}\varepsilon_{sc}\,,$$

$$\text{where} \quad \mathscr{C}_1^{\mathsf{REM}}(d) = \frac{\sqrt{d/3}L_{\max} + 1/2\sqrt{3}}{m_{\min} - 1/2} \quad \text{and} \quad \mathscr{C}_2^{\mathsf{REM}} = \frac{3}{m_{\min} - 1/2}\,,$$

$$\mathscr{C}_1^{\mathsf{REI}}(d) = \frac{\sqrt{d/3}L_{\max}}{(m_{\min} - 1/2)} \quad \text{and} \quad \mathscr{C}_2^{\mathsf{REI}} = \frac{3}{m_{\min} - 1/2}$$

*with $L_{\max}$ and $m_{\min}$ as defined in Theorem 1.*

The convergence rate of these two schemes is generally consistent with EM, EI methods, differing only in the coefficients. As shown in display (6), the key idea behind the randomized midpoint method is to introduce a uniformly distributed random variable $U_n$ to evaluate the term $\gamma(T - t, X_t^{\leftarrow})$ at a random point within the time interval $[0, h]$. In the proofs provided in Appendix B.3 and B.4 (corresponding to $\alpha = \mathsf{REM}$ and $\alpha = \mathsf{REI}$, respectively), we demonstrate that $\mathbb{E}_{U_n}[\vartheta_{n+1}^{\alpha}]$ yields an accurate approximation to the true distribution, with an error of order $\mathcal{O}(h^{5/2}) + \mathcal{O}(h)\varepsilon_{sc}$. However, this randomization also introduces an variance term $\|\vartheta_{n+1}^{\alpha} - \mathbb{E}_{U_n}[\vartheta_{n+1}^{\alpha}]\|_{\mathbb{L}_2}$, of order $\mathcal{O}(h^{3/2}) + \mathcal{O}(h)\varepsilon_{sc}$, which obscures the benefits of the improved estimation. As a result, the midpoint randomization offers no improvement in convergence over EM and EI methods.

Although midpoint randomization itself does not improve convergence in our setting, it enables parallel computation [15, 23], significantly reducing computational complexity and enhancing efficiency.

## 4 Second-order Acceleration

In this section, we propose an accelerated sampler that leverages Hessian estimation. The core idea behind the acceleration is the *Local Linearization Method*, introduced in [33], which approximates the drift term of an SDE using its Itô expansion over small time intervals. To illustrate this, we begin with a general framework for the backward process.

$$\mathrm{d}x_t = \gamma(T - t, x_t)\,\mathrm{d}t + \sigma\,\mathrm{d}W_t, \tag{8}$$

where $\sigma > 0$ and $W_t$ is the $d$-dimensional Brownian motion. We assume that $\gamma(t, x) \in C^{1,3}(\mathbb{R}_+ \times \mathbb{R}^d)$ and approximate it by a linear function in both state and time within each discretization step. Applying Itô's formula to $\gamma(T - t, x)$, we derive the following approximation for $\gamma(T - t, x_t) - \gamma(T - s, x_s)$

$$\left( \frac{\sigma^2}{2} \frac{\partial^2 \gamma}{\partial x^2}(T - s, x_s) - \frac{\partial \gamma}{\partial t}(T - s, x_s) \right)(t - s) + \frac{\partial \gamma}{\partial x}(T - s, x_s) \cdot (x_t - x_s).$$

Here and henceforth, we abbreviate the partial derivative $\frac{\partial^\alpha g(z)}{\partial z^\alpha}\big|_{z=z_0}$ as $\frac{\partial^\alpha g}{\partial z^\alpha}(z_0)$. This allows us to express $\gamma(T - t, x_t)$ in the following form

$$\gamma(T - t, x_t) \approx \gamma(T - s, x_s) + L_s(x_t - x_s) + M_s(t - s), \tag{9}$$

with

$$L_s = \frac{\partial \gamma}{\partial x}(T - s, x_s) \qquad \text{and} \qquad M_s = \frac{\sigma^2}{2} \frac{\partial^2 \gamma}{\partial x^2}(T - s, x_s) - \frac{\partial \gamma}{\partial t}(T - s, x_s).$$

Here, $L_s$ is the first-order spatial derivative of $\gamma$, capturing its local variation with respect to position $x$. The term $M_s$ represents the temporal evolution of $\gamma$, incorporating information about how its shape changes over both space and time. Substituting display (9) into the original SDE (8), we obtain

$$dx_t = [\gamma(T - s, x_s) + L_s(x_t - x_s) + M_s(t - s)]\,\mathrm{d}t + \sigma\,\mathrm{d}W_t.$$

This formulation ensures that the discretized process preserves the essential structure of the original dynamics while remaining computationally tractable. Unlike discretization schemes in Section 2, which rely on direct numerical integration, this transformed SDE allows for analytical solutions within each small time interval.

Setting $\gamma(T - t, x) = x/2 + \nabla \log p_{T-t}(x)$, $\sigma = 1$, let $\Delta t \in [0, h]$ and $s = nh$. By Itô's formula, we derive the analytical expression for $x_t$. In the resulting expression, we denote $x_s$ by $\vartheta_n^{\mathsf{SO}}$. Then, we obtain that for any $t \in [nh, (n+1)h]$, it holds that

$$x_t = \vartheta_n^{\mathsf{SO}} + \int_{nh}^t \left( \frac{1}{2}\vartheta_n^{\mathsf{SO}} + \nabla \log p_{T-nh}(\vartheta_n^{\mathsf{SO}}) + L_n(x_u - \vartheta_n^{\mathsf{SO}}) + M_n(u - nh) \right) \mathrm{d}u + \int_{nh}^t \mathrm{d}W_u \tag{10}$$

where

$$L_n = \frac{1}{2}I_d + \nabla^2 \log p_{T-nh}(\vartheta_n^{\mathsf{SO}}), \quad M_n = \frac{1}{2}\sum_{j=1}^d \frac{\partial^2}{\partial x_j^2} \nabla \log p_{T-nh}(\vartheta_n^{\mathsf{SO}}) - \frac{\partial}{\partial t} \nabla \log p_{T-nh}(\vartheta_n^{\mathsf{SO}}).$$

Thus, $L_n$ contains the Hessian of the score function. The term $M_n$ measures the difference between the spatial and temporal changes in the score function, reflecting the balance between curvature effects and temporal adaptation in the diffusion process. Notice that $x_{(n+1)h}$ is the point we aim to approximate. For this, we need to estimate the score function and its higher-order derivatives to obtain accurate estimates of $L_n$ and $M_n$, denoted by $s_*^{(L)}$ and $s_*^{(M)}$, respectively.

By the work of [28], higher-order derivatives of $\log p_t(x)$ with respect to $x$ can be accurately estimated. Moreover, we show in Appendix C that $\partial_t \nabla \log p_t(x)$ can be expressed as a combination of up to second-order partial derivatives of $\log p_t(x)$ with respect to $x$. Thus, estimating $\partial_t \nabla \log p_t(x)$ requires no additional assumptions beyond those for accurately estimating $\partial_x^2 \nabla \log p_t(x)$. We also require the following assumption.

**Assumption 5.** *For some constants $\varepsilon_{sc}^{(L)}, \varepsilon_{sc}^{(M)} > 0$, the estimate for high-order derivatives of the score function satisfies that*

$$\sup_{0 \leqslant n \leqslant N-1} \left\| s_*^{(L)}(T - nh, \vartheta_n^{\mathsf{SO}}) - L_n \right\|_{\mathbb{L}_2} \leqslant \varepsilon_{sc}^{(L)},$$

$$\sup_{0 \leqslant n \leqslant N-1} \left\| s_*^{(M)}(T - nh, \vartheta_n^{\mathsf{SO}}) - M_n \right\|_{\mathbb{L}_2} \leqslant \varepsilon_{sc}^{(M)}.$$

This assumption has been adopted in prior works [24] and [27]. Substituting these estimates into display (10) then gives

$$x_t = \vartheta_n^{\mathsf{SO}} + \int_{nh}^t \left[ \gamma(T - nh, \vartheta_n^{\mathsf{SO}}) + s_*^{(L)}(T - nh, \vartheta_n^{\mathsf{SO}})(x_u - \vartheta_n^{\mathsf{SO}}) \right.$$
$$\left. + s_*^{(M)}(T - nh, \vartheta_n^{\mathsf{SO}})(u - nh) \right] \mathrm{d}u + \int_{nh}^t \mathrm{d}W_u \,.$$

Let $\vartheta_{n+1}^{\mathsf{SO}}$ denote $x_{(n+1)h}$. The second-order discretization scheme is given by[3]

$$\vartheta_{n+1}^{\mathsf{SO}} = \vartheta_n^{\mathsf{SO}} + s_*^{(L)}(T - nh, \vartheta_n^{\mathsf{SO}})^{-1} \left( e^{s_*^{(L)}(T - nh, \vartheta_n^{\mathsf{SO}})h} - I_d \right) \left( \frac{1}{2} \vartheta_n^{\mathsf{SO}} + s_*(T - nh, \vartheta_n^{\mathsf{SO}}) \right)$$
$$+ s_*^{(L)}(T - nh, \vartheta_n^{\mathsf{SO}})^{-2} \left( e^{s_*^{(L)}(T - nh, \vartheta_n^{\mathsf{SO}})h} - s_*^{(L)}(T - nh, \vartheta_n^{\mathsf{SO}})h - I_d \right) s_*^{(M)}(T - nh, \vartheta_n^{\mathsf{SO}})$$
$$+ \int_{nh}^{(n+1)h} e^{s_*^{(L)}(T - nh, \vartheta_n^{\mathsf{SO}})[(n+1)h - t]} \, \mathrm{d}W_t \,.$$

We assume additional smoothness on the score function.

**Assumption 6.** *Let $\| \cdot \|_F$ denote the Frobenius norm. There exists a positive constant $L_F$ such that*

$$\left\| \nabla^2 \log p_t(x) - \nabla^2 \log p_t(y) \right\|_F \leqslant L_F \left\| x - y \right\|, \ \ \forall x, y \in \mathbb{R}^d \,.$$

As shown in Theorems 4 and 5 of [27], this condition plays a crucial role in bounding the Wasserstein distance for Hessian estimates and can be easily verified in the Gaussian case.

**Assumption 7.** *There exists a constant $M_2 > 0$ such that, for any $n = 0, \ldots, N - 1$ and $t \in [nh, (n+1)h]$, it holds that*

$$\left\| \nabla^2 \log p_{T-t}(x) - \nabla^2 \log p_{T-nh}(x) \right\| \leqslant M_2 h(1 + \|x\|), \ \ \forall x \in \mathbb{R}^d \,.$$

We now quantify the $\mathsf{W}_2$ distance between the generated distribution $\mathcal{L}(\vartheta_N^{\mathsf{SO}})$ and the target $p_0$.

**Theorem 4.** *Suppose that Assumptions 1, 3, 5, 6 and 7 hold, then*

$$\mathsf{W}_2(\mathcal{L}(\vartheta_N^{\mathsf{SO}}), p_0) \lesssim e^{-m_{\min}T} \|X_0\|_{\mathbb{L}_2} + \mathscr{C}_1^{\mathsf{SO}}(d)h + \mathscr{C}_2^{\mathsf{SO}} \left( \varepsilon_{sc} + \frac{2}{3} \sqrt{hd} \varepsilon_{sc}^{(L)} + \frac{1}{2} h \varepsilon_{sc}^{(M)} \right) \quad (11)$$

*where $\mathscr{C}_1^{\mathsf{SO}}(d) = e^{(L_{\max} - 1/2)h} \cdot \dfrac{\sqrt{d}(L_{\max}^{3/2} + \sqrt{2}L_F/4)}{m_{\min} - 1/2}$ and $\mathscr{C}_2^{\mathsf{SO}} = \dfrac{e^{(L_{\max} - 1/2)h}}{m_{\min} - 1/2}$ with $L_{\max}$ and $m_{\min}$ as defined in Theorem 1.*

Before discussing the results of this theorem, let us state its direct consequence.

**Corollary 5.** *For a given $\varepsilon > 0$, the Wasserstein distance satisfies $\mathsf{W}_2(\mathcal{L}(\vartheta_N^{\mathsf{SO}}), p_0) < \varepsilon$ after $N = \mathcal{O}\left( \frac{\sqrt{d}}{\varepsilon} \log\left( \frac{\sqrt{d}}{\varepsilon} \right) \right)$ iterations, provided that $T = \mathcal{O}\left( \log(\frac{\sqrt{d}}{\varepsilon}) \right)$ and $h = \mathcal{O}(\frac{\varepsilon}{\sqrt{d}})$.*

In the above, we present the first Wasserstein convergence analysis of an accelerated sampler that utilizes accurate score function estimation and second-order information about log-densities. The total error arises from the same resources as in Theorem 1. The **first term** on the right-hand side of display (11) captures the initialization error. The **second term** reflects the benefit of second-order acceleration, where improved discretization reduces the error in approximating the reverse SDE. The **third term** accounts for errors in estimating both the spatial and temporal components of the score function, as well as the higher order terms $L_n$ and $M_n$.

The accelerated convergence of this method is driven by two key innovations: approximating the drift term through its Itô expansion rather than endpoint evaluations, and deriving a closed-form solution to the integral equation (10) using the Itô formula, akin to Exponential Integrator techniques. Compared to the four schemes described in Section 3, the proposed second-order algorithm offers a clear computational advantage: it requires only $\widetilde{\mathcal{O}}(1/\varepsilon)$ iterations (v.s. $\widetilde{\mathcal{O}}(1/\varepsilon^2)$), and permits a

---

[3]Due to space limitations, we refer readers to Appendix E for a complete derivation.

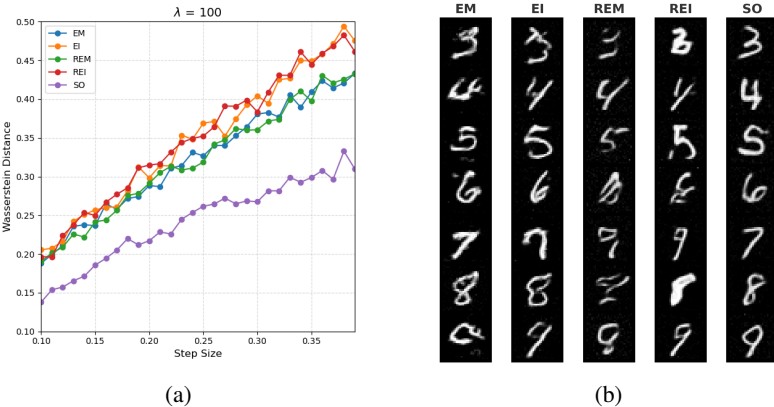

(a)            (b)

Figure 1: **(a)**: Errors of SGMs under EM, EI, REM, REI and SO with different choices of $h$. **(b)**: Samples generated by five different algorithms on the MNIST dataset.

larger step size $h = \mathcal{O}(\varepsilon)$, enabling faster progression in each iteration instead of $h = \mathcal{O}(\varepsilon^2)$. As a result, the second-order method achieves the same accuracy with fewer iterations and a larger step size, making it a more efficient method for approximating the target distribution.

We note that the convergence rate $\widetilde{\mathcal{O}}(1/\varepsilon)$ matches that of the accelerated Denoising Diffusion Probabilistic Models (DDPM) sampler [17] proposed in [27], which achieves this rate in KL divergence and relies on Hessian estimation. Additionally, this rate aligns with the iteration complexity results in TV distance presented in [19] when setting $p = 1$ in their framework.

## 5 Numerical Studies

In this section, we compare the performance of the SGMs under EM, EI, REM, REI discretization schemes described in Section 2, as well as the second-order acceleration method (SO) proposed in Section 4, evaluated on both synthetic data and the MNIST dataset.

### 5.1 Experiments on Synthetic Data

We apply the five algorithms to the posterior of penalized logistic regression, defined by $p_0(\theta) \propto \exp(-f(\theta))$, with the potential function $f$ defined via

$$f(\theta) = \frac{\lambda}{2} \|\theta\|^2 + \frac{1}{n_{\mathsf{data}}} \sum_{i=1}^{n_{\mathsf{data}}} \log \left(1 + \exp(-y_i x_i^\top \theta)\right),$$

where $\lambda > 0$ denotes the tuning parameter. The data $\{x_i, y_i\}_{i=1}^{n_{\mathsf{data}}}$, composed of binary labels $y_i \in \{-1, 1\}$ and features $x_i = (x_{i,1}, \cdots, x_{i,d})^\top \in \mathbb{R}^d$ generated from $x_{i,j} \overset{\mathrm{i.i.d.}}{\sim} \mathcal{N}(0, 100)$. Set $\lambda = 100$ with $d = 20$ and $n_{\mathsf{data}} = 100$. Appendix F provides additional results and implementation details. Figure 1a presents the $\mathsf{W}_2$ distance measured along the first dimension between the empirical distributions of the samples from the five algorithms and the target distribution under different choices of $h$. These results support our theoretical findings: all discretization schemes from Section 2 exhibit similar convergence behavior, while the proposed Hessian-based sampler consistently achieves superior performance.

### 5.2 Real Data Analysis

We apply the four SGM discretization schemes and the second-order algorithm to the MNIST dataset. To accelerate the SO algorithm, we use Hessian-vector products (HVPs) instead of explicitly computing the Hessian. The results in Figure 1b demonstrate that SO outperforms the others. Additional results and implementation details are provided in Appendix G.

# 6 Discussion

The Wasserstein-2 distance, used in this paper, serves as a natural and practical metric for measuring errors in diffusion models. However, recent work on the convergence theory of diffusion models has also explored alternative metrics such as total variation distance and KL divergence. A promising direction for future research is to establish convergence guarantees with respect to these alternative distances.

Moreover, while this work makes progress in provably accelerating SDE-based diffusion sampling in Wasserstein distance, it would also be valuable to explore deterministic samplers based on probability flow ODEs.

Finally, for clarity and simplicity, we focus on a specific choice of drift functions $f$ and $g$ in the forward process, corresponding to the Ornstein–Uhlenbeck process. Extending this analysis to a more general framework with broader choices of $(f, g)$ is an interesting avenue for future research.

## Acknowledgments and Disclosure of Funding

Lu Yu is supported by the City University of Hong Kong Startup Fund and Hong Kong RGC Grant 21306325.

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

# A Discussion of Theoretical Advances in Accelerating Samplers for Diffusion Models

Score-based diffusion models can be formulated using either SDEs or their deterministic counterparts, known as probability flow ODEs [36]. While SDE-based samplers generate samples through stochastic simulation, ODE-based samplers provide a deterministic alternative. Theoretical advancements in accelerating these samplers have emerged only recently. A significant step toward designing provably accelerated, training-free methods was made by [24], who propose and analyze acceleration for both ODE- and SDE-based samplers. Their accelerated SDE sampler leverages higher-order expansions of the conditional density to enhance efficiency. This was followed by the work of [25], which provided convergence guarantees for probability flow ODEs. Furthermore, [19] studies the convergence properties of deterministic samplers based on probability flow ODEs, using the Runge-Kutta integrator; [41] propose and analyze a training-free acceleration algorithm for SDE-based samplers, based on the stochastic Runge-Kutta method. [27] proposes a novel accelerated SDE-based sampler when Hessian information is available. Another line of work involves the midpoint randomized method. In particular, [15] explore ODE acceleration by incorporating a randomized midpoint method, leveraging its advantages in parallel computation. A more recent work by [23] improved upon the ODE sampler proposed by [15], achieving the state-of-the-art convergence rate.

We note that all of these works provide convergence analysis in terms of either KL divergence or TV distance. Among these, [27] accelerates the stochastic DDPM sampler by leveraging precise score and Hessian estimations of the log density, even for possibly non-smooth target distributions. This is achieved through a novel Bayesian approach based on tilting factor representation and Tweedie's formula. [19] accelerates the ODE sampler by utilizing $p$-th ($p \geqslant 1$) order information of the score function, with the target distribution supported on a compact set and employing early stopping. These two works are the most similar to our proposed accelerated sampler in that they all rely on the Hessian information of the log density. However, their settings differ from ours, and their convergence analyses are neither directly applicable to our framework nor precisely expressed in terms of Wasserstein distance.[4].

# B Proof of Section 3

We define several stochastic processes associated with the backward process $X_t^{\leftarrow}$ and the sample path $\vartheta_n$. First, recall that $X_t^{\leftarrow}$ is described by the following SDE:

$$\mathrm{d}X_t^{\leftarrow} = \left(\frac{1}{2}X_t^{\leftarrow} + \nabla \log p_t(X_t^{\leftarrow})\right)\mathrm{d}t + \mathrm{d}W_t, \quad X_0^{\leftarrow} \sim p_T,$$

and $\{\vartheta_n^{\alpha}, 0 \leqslant n \leqslant N\}$ satisfies the iterative law:

$$\vartheta_{n+1}^{\alpha} = \mathcal{G}_h^{\alpha}(\vartheta_n^{\alpha}, \{W_t\}_{nh \leqslant t \leqslant (n+1)h}),$$

where $\alpha \in \{\mathsf{EM}, \mathsf{EI}, \mathsf{REM}, \mathsf{REI}, \mathsf{SO}\}$.

Based on $X_t^{\leftarrow}$, we define the following two processes, $\{Y_t, 0 \leqslant t \leqslant T\}$ and $\{\tilde{Y}_t, 0 \leqslant t \leqslant h\}$. $Y_t$ satisfies the SDE

$$\mathrm{d}Y_t = \left(\frac{1}{2}Y_t + \nabla \log p_{T-t}(Y_t)\right)\mathrm{d}t + \mathrm{d}W_t, \quad Y_0 \sim \hat{p}_T.$$

$\tilde{Y}_t$ actually relies on $X_t^{\leftarrow}$ on the time interval $[nh, (n+1)h]$ for each $n$. However, we only need this notation in the proof of one-step discretization error, then we allow for some slight abuse of notation by omitting $n$, since it will not lead to any confusion. Therefore, $\{\tilde{Y}_t, 0 \leqslant t \leqslant h\}$ satisfies

$$\mathrm{d}\tilde{Y}_t = \left(\frac{1}{2}\tilde{Y}_t + \nabla \log p_{T-t}(\tilde{Y}_t)\right)\mathrm{d}t + \mathrm{d}W_t, \quad \tilde{Y}_0 = \vartheta_n. \tag{12}$$

---

[4]When the target distribution is compactly supported, Pinsker's inequality allows translating TV or KL divergence into Wasserstein distance. However, this often yields loose bounds, especially in high dimensions, where the actual Wasserstein distance may be much smaller.

Recall that we have defined two processes $\vartheta_{n+u}^{\mathsf{REM}}$ and $\vartheta_{n+u}^{\mathsf{REI}}$ in Section 3.2, that is, for any $u \in [0,1]$ and $n = 0, \cdots, N-1$,

$$\vartheta_{n+u}^{\mathsf{REM}} := (1 + \frac{uh}{2})\vartheta_n^{\mathsf{REM}} + uhs_*(T - nh, \vartheta_n^{\mathsf{REM}}) + \sqrt{uh}\xi_n \,,$$

$$\vartheta_{n+u}^{\mathsf{REI}} := e^{uh/2}\vartheta_n^{\mathsf{REI}} + 2(e^{uh/2} - 1)s_*(T - nh, \vartheta_n^{\mathsf{REI}}) + \sqrt{e^{uh} - 1}\xi_n' \,.$$

This section is devoted to proving the convergence rate of the diffusion model under various discretization schemes. To this end, we need the following auxiliary lemmas.

**Lemma 6** (Lemma 9 in [13]). *Suppose that Assumption 1 holds. Then, $\nabla \log p_t(x)$ is $L(t)$-Lipschitz, where $L(t)$ is given by*

$$L(t) = \min\{(1 - e^{-t})^{-1}, e^t L_0\} = \begin{cases} e^t L_0 & \text{if } t \leqslant \log(1 + \frac{1}{L_0}) \\ (1 - e^{-t})^{-1} & \text{if } t > \log(1 + \frac{1}{L_0}) \end{cases} \,.$$

*Therefore,*

$$L(t) \leqslant L_0 + 1 \,.$$

**Lemma 7** (Proof of Proposition 7 in [13]). *Suppose that Assumption 1 holds. Then, $\nabla \log p_t(x)$ is $m(t)$-strongly log-concave, where $m(t)$ is given by*

$$m(t) = \frac{1}{e^{-t}/m_0 + (1 - e^{-t})} \,.$$

*Therefore,*

$$m(t) \geqslant \min\{1, m_0\} \,.$$

Combining these two lemmas, we conclude that the Hessian matrix of $\log p_t$ satisfies the following condition

$$-L(t)I_d \preccurlyeq \nabla^2 \log p_t(\cdot) \preccurlyeq -m(t)I_d \,.$$

We will frequently use Grönwall's inequality in the proof. Below, we present a specialized form tailored to the relevant processes.

**Lemma 8.** *Suppose the Assumption 1 holds, consider two stochastic processes $H_t$ and $G_t$ defined on the time interval $[t_1, t_2]$, if they satisfy the same SDE, especially motivated by the same Brownian motion, which means that*

$$\mathrm{d}H_t = \left(\frac{1}{2}H_t + \nabla \log p_{T-t}(H_t)\right) \mathrm{d}t + \mathrm{d}W_t \,,$$

$$\mathrm{d}G_t = \left(\frac{1}{2}G_t + \nabla \log p_{T-t}(G_t)\right) \mathrm{d}t + \mathrm{d}W_t \,.$$

*then for each $t \in [t_1, t_2]$,*

$$\|H_t - G_t\|_{\mathbb{L}_2} \leqslant e^{-\int_{t_1}^t (m(T-s) - \frac{1}{2})\,\mathrm{d}s} \|H_{t_1} - G_{t_1}\|_{\mathbb{L}_2} \,.$$

Applying Lemma 8 to different processes and time intervals, we derive the following inequalities essential for our proof.

$$\left\|Y_{nh+t} - \tilde{Y}_t\right\|_{\mathbb{L}_2} \leqslant e^{-\int_{nh}^{nh+t}(m(T-s) - \frac{1}{2})\,\mathrm{d}s} \left\|Y_{nh} - \tilde{Y}_0\right\|_{\mathbb{L}_2}, \quad \forall t \in [0, h] \,. \tag{13}$$

$$\|Y_t - X_t^\leftarrow\|_{\mathbb{L}_2} \leqslant e^{-\int_0^t (m(T-s) - \frac{1}{2})\,\mathrm{d}s} \|Y_0 - X_0^\leftarrow\|_{\mathbb{L}_2}, \quad \forall t \in [0, T] \,. \tag{14}$$

which follow from the fact that $\{Y_t, 0 \leqslant t \leqslant T\}$, $\{\tilde{Y}_t, 0 \leqslant t \leqslant h\}$, $\{X_t^\leftarrow, 0 \leqslant t \leqslant T\}$ all satisfy the same SDE as in Lemma 8, by applying a time-shifting operator to $\tilde{Y}_t$.

## B.1 Proof of Theorem 1: Part I

In this part, we provide the first part of the proof of Theorem 1, with respect to Euler-Maruyama method. To achieve this, we first prove the one-step discretization error in the following proposition.

**Proposition 9.** *Suppose that Assumption 1, 2 and 3 are satisfied. Then, the following two claims hold.*

*(1) Firstly, it holds that*

$$\left\|\tilde{Y}_h - \vartheta_{n+1}^{\mathsf{EM}}\right\|_{\mathbb{L}_2} \leqslant h^2 (C_1(n)^2 + M_1) \left\|Y_{nh} - \vartheta_n^{\mathsf{EM}}\right\|_{\mathbb{L}_2}$$
$$+ h^2 \left[ C_1(n) \left( C_1(n)C_2(n) + \frac{1}{2}C_4 + C_3(n) \right) + M_1 \left( 1 + C_2(n) + C_4 \right) \right]$$
$$+ h^{3/2} \sqrt{d} C_1(n)$$
$$+ h \varepsilon_{sc} ,$$

*where*

$$C_1(n) = \frac{1}{2} + \frac{1}{h} \int_{nh}^{(n+1)h} L(T - t) \, \mathrm{d}t ,$$
$$C_2(n) = e^{- \int_0^{nh} (m(T-t) - \frac{1}{2}) \, \mathrm{d}t} \left\|Y_0 - X_T\right\|_{\mathbb{L}_2} ,$$
$$C_3(n) = \frac{1}{h} \int_{nh}^{(n+1)h} (dL(T - t))^{1/2} \, \mathrm{d}t ,$$
$$C_4 = \sup_{0 \leqslant t \leqslant T} \left\|X_t\right\|_{\mathbb{L}_2} .$$

*(2) As a result,*

$$\left\|Y_{(n+1)h} - \vartheta_{n+1}^{\mathsf{EM}}\right\|_{\mathbb{L}_2} \leqslant r_n^{\mathsf{EM}} \left\|Y_{nh} - \vartheta_n^{\mathsf{EM}}\right\|_{\mathbb{L}_2} + h^2 C_n^{\mathsf{EM}} + h^{3/2} \sqrt{d} C_1(n) + h \varepsilon_{sc} ,$$

*where*

$$r_n^{\mathsf{EM}} = e^{- \int_{nh}^{(n+1)h} (m(T-t) - \frac{1}{2}) \, \mathrm{d}t} + h^2 (C_1(n)^2 + M_1) ,$$
$$C_n^{\mathsf{EM}} = C_1(n) \left( C_1(n)C_2(n) + \frac{1}{2}C_4 + C_3(n) \right) + M_1 \left( 1 + C_2(n) + C_4 \right) .$$

*Proof.* We prove the two claims sequentially.
**Proof of Claim (1).** Rewrite display (12) in the integral form,

$$\tilde{Y}_h = \tilde{Y}_0 + \int_0^h \left( \frac{1}{2} \tilde{Y}_t + \nabla \log p_{T-nh-t}(\tilde{Y}_t) \right) \mathrm{d}t + \int_{nh}^{(n+1)h} \mathrm{d}W_t .$$

For Euler-Maruyama method, we can write $\vartheta_{n+1}^{\mathsf{EM}}$ in integral form as follows

$$\vartheta_{n+1}^{\mathsf{EM}} = \vartheta_n^{\mathsf{EM}} + \int_{nh}^{(n+1)h} \left( \frac{1}{2} \vartheta_n^{\mathsf{EM}} + s_*(T - nh, \vartheta_n^{\mathsf{EM}}) \right) \mathrm{d}t + \int_{nh}^{(n+1)h} \mathrm{d}W_t .$$

Note that $\tilde{Y}_0 = \vartheta_n^{\mathsf{EM}}$, then, it holds that

$$
\left\| \tilde{Y}_h - \vartheta_{n+1}^{\mathsf{EM}} \right\|_{\mathbb{L}_2} = \left\| \frac{1}{2} \int_0^h (\tilde{Y}_t - \vartheta_n^{\mathsf{EM}}) \, \mathrm{d}t + \int_0^h (\nabla \log p_{T-nh-t}(\tilde{Y}_t) - s_*(T - nh, \vartheta_n^{\mathsf{EM}})) \, \mathrm{d}t \right\|_{\mathbb{L}_2}
$$

$$
\leqslant \underbrace{\left\| \frac{1}{2} \int_0^h (\tilde{Y}_t - \tilde{Y}_0) \, \mathrm{d}t + \int_0^h (\nabla \log p_{T-nh-t}(\tilde{Y}_t) - \nabla \log p_{T-nh-t}(\tilde{Y}_0) \, \mathrm{d}t \right\|_{\mathbb{L}_2}}_{\mathrm{I}}
$$

$$
+ \underbrace{\left\| \int_0^h (\nabla \log p_{T-nh-t}(\vartheta_n^{\mathsf{EM}}) - \nabla \log p_{T-nh}(\vartheta_n^{\mathsf{EM}})) \, \mathrm{d}t \right\|_{\mathbb{L}_2}}_{\mathrm{II}}
$$

$$
+ \underbrace{\left\| \int_0^h (\nabla \log p_{T-nh}(\vartheta_n^{\mathsf{EM}}) - s_*(T - nh, \vartheta_n^{\mathsf{EM}})) \, \mathrm{d}t \right\|_{\mathbb{L}_2}}_{\mathrm{III}} .
$$

$$(15)$$

Here, we decompose the term $\left\| \tilde{Y}_h - \vartheta_{n+1}^{\mathsf{EM}} \right\|_{\mathbb{L}_2}$ into a sum of three terms and then control each term individually.

For the term I of inequality (15), by Assumption 1 and Lipschitzness of $\nabla \log p_t$, we obtain

$$
\mathrm{I} = \left\| \frac{1}{2} \int_0^h (\tilde{Y}_t - \tilde{Y}_0) \, \mathrm{d}t + \int_0^h (\nabla \log p_{T-nh-t}(\tilde{Y}_t) - \nabla \log p_{T-nh-t}(\tilde{Y}_0)) \, \mathrm{d}t \right\|_{\mathbb{L}_2}
$$

$$
\leqslant \frac{1}{2} \int_0^h \left\| \tilde{Y}_t - \tilde{Y}_0 \right\|_{\mathbb{L}_2} \, \mathrm{d}t + \int_0^h L(T - nh - t) \left\| \tilde{Y}_t - \tilde{Y}_0 \right\|_{\mathbb{L}_2} \, \mathrm{d}t
$$

$$
\leqslant \left( \frac{1}{2}h + \int_{nh}^{(n+1)h} L(T - t) \, \mathrm{d}t \right) \sup_{0 \leqslant t \leqslant h} \left\| \tilde{Y}_t - \tilde{Y}_0 \right\|_{\mathbb{L}_2} .
$$

We then proceed to derive the upper bound for the term $\sup_{0 \leqslant t \leqslant h} \left\| \tilde{Y}_t - \tilde{Y}_0 \right\|_{\mathbb{L}_2}$.

**Lemma 10.** *When $p_0$ satisfies Assumption 1, it holds that*

$$
\sup_{0 \leqslant t \leqslant h} \left\| \tilde{Y}_t - \tilde{Y}_0 \right\|_{\mathbb{L}_2} \leqslant \left( \frac{1}{2}h + \int_{nh}^{(n+1)h} L(T - t) \, \mathrm{d}t \right) \left\| Y_{nh} - \vartheta_n^{\mathsf{EM}} \right\|_{\mathbb{L}_2}
$$

$$
+ \left( \frac{1}{2}h + \int_{nh}^{(n+1)h} L(T - t) \, \mathrm{d}t \right) e^{- \int_0^{nh} (m(T-t) - \frac{1}{2}) \, \mathrm{d}t} \left\| Y_0 - X_T \right\|_{\mathbb{L}_2}
$$

$$
+ \frac{1}{2}h \sup_{0 \leqslant t \leqslant T} \left\| X_t \right\|_{\mathbb{L}_2} + \int_{nh}^{(n+1)h} (dL(T - t))^{1/2} \, \mathrm{d}t + \sqrt{dh} .
$$

Notice that we have no initial limit on the $\tilde{Y}_t$ in Lemma 10, which means that we can use this lemma to any discretization scheme.

For the term II of (15), we first rely on Assumption 2 to derive

$$
\mathrm{II} = \left\| \int_0^h (\nabla \log p_{T-nh-t}(\vartheta_n^{\mathsf{EM}}) - \nabla \log p_{T-nh}(\vartheta_n^{\mathsf{EM}})) \, \mathrm{d}t \right\|_{\mathbb{L}_2}
$$

$$
\leqslant \int_0^h \left\| \nabla \log p_{T-nh-t}(\vartheta_n^{\mathsf{EM}}) - \nabla \log p_{T-nh}(\vartheta_n^{\mathsf{EM}}) \right\|_{\mathbb{L}_2} \, \mathrm{d}t
$$

$$
\leqslant h^2 M_1 (1 + \left\| \vartheta_n^{\mathsf{EM}} \right\|_{\mathbb{L}_2}) .
$$

Using the triangle inequality and (14), we obtain

$$
\begin{aligned}
\left\|\vartheta_n^{\mathsf{EM}}\right\|_{\mathbb{L}_2} &\leqslant \left\|Y_{nh} - \vartheta_n^{\mathsf{EM}}\right\|_{\mathbb{L}_2} + \left\|Y_{nh} - X_{nh}^{\leftarrow}\right\|_{\mathbb{L}_2} + \left\|X_{nh}^{\leftarrow}\right\|_{\mathbb{L}_2} \\
&\leqslant \left\|Y_{nh} - \vartheta_n^{\mathsf{EM}}\right\|_{\mathbb{L}_2} + e^{-\int_0^{nh}(m(T-t)-\frac{1}{2})\,\mathrm{d}t}\left\|Y_0 - X_T\right\|_{\mathbb{L}_2} + \sup_{0\leqslant t\leqslant T}\left\|X_t\right\|_{\mathbb{L}_2}.
\end{aligned}
\tag{16}
$$

For the term III of (15), it follows from Assumption 3 that

$$
\begin{aligned}
\text{III} &= \left\|\int_0^h \left(\nabla \log p_{T-nh}(\vartheta_n^{\mathsf{EM}}) - s_*(T-nh, \vartheta_n^{\mathsf{EM}})\right)\mathrm{d}t\right\|_{\mathbb{L}_2} \\
&\leqslant \int_0^h \left\|\nabla \log p_{T-nh}(\vartheta_n^{\mathsf{EM}}) - s_*(T-nh, \vartheta_n^{\mathsf{EM}})\right\|_{\mathbb{L}_2}\mathrm{d}t \\
&\leqslant h\varepsilon_{sc}.
\end{aligned}
$$

Combining these terms above, we obtain that

$$
\begin{aligned}
\left\|\tilde{Y}_h - \vartheta_{n+1}^{\mathsf{EM}}\right\|_{\mathbb{L}_2} &\leqslant h^2(C_1(n)^2 + M_1)\left\|Y_{nh} - \vartheta_n^{\mathsf{EM}}\right\|_{\mathbb{L}_2} \\
&\quad + h^2\left[C_1(n)\left(C_1(n)C_2(n) + \frac{1}{2}C_4 + C_3(n)\right) + M_1\left(1 + C_2(n) + C_4\right)\right] \\
&\quad + h^{3/2}\sqrt{d}C_1(n) \\
&\quad + h\varepsilon_{sc},
\end{aligned}
\tag{17}
$$

where

$$
\begin{aligned}
C_1(n) &= \frac{1}{2} + \frac{1}{h}\int_{nh}^{(n+1)h} L(T-t)\,\mathrm{d}t, \\
C_2(n) &= e^{-\int_0^{nh}(m(T-t)-\frac{1}{2})\,\mathrm{d}t}\left\|Y_0 - X_T\right\|_{\mathbb{L}_2}, \\
C_3(n) &= \frac{1}{h}\int_{nh}^{(n+1)h} (dL(T-t))^{1/2}\,\mathrm{d}t, \\
C_4 &= \sup_{0\leqslant t\leqslant T}\left\|X_t\right\|_{\mathbb{L}_2}.
\end{aligned}
$$

This completes the proof of Claim **(1)**.

**Proof of Claim (2)**. By the triangle inequality, we have

$$
\left\|Y_{(n+1)h} - \vartheta_{n+1}^{\mathsf{EM}}\right\|_{\mathbb{L}_2} \leqslant \left\|Y_{(n+1)h} - \tilde{Y}_h\right\|_{\mathbb{L}_2} + \left\|\tilde{Y}_h - \vartheta_{n+1}^{\mathsf{EM}}\right\|_{\mathbb{L}_2}.
\tag{18}
$$

Applying (13) to the first term of (18), we obtain that

$$
\left\|Y_{(n+1)h} - \tilde{Y}_h\right\|^2 \leqslant e^{-\int_{nh}^{(n+1)h}(2m(T-t)-1)\,\mathrm{d}t}\left\|Y_{nh} - \tilde{Y}_0\right\|^2.
\tag{19}
$$

Notice that $\tilde{Y}_0 = \vartheta_n^{\mathsf{EM}}$, it then follows that

$$
\left\|Y_{(n+1)h} - \tilde{Y}_h\right\|_{\mathbb{L}_2} \leqslant e^{-\int_{nh}^{(n+1)h}(m(T-t)-\frac{1}{2})\,\mathrm{d}t}\left\|Y_{nh} - \vartheta_n^{\mathsf{EM}}\right\|_{\mathbb{L}_2}.
$$

Claim **(2)** follows directly from our previous results and Claim **(1)**. Since this step is independent of the discretization method, it applies to all the schemes discussed in this section. In the following analysis, we omit this step and proceed directly with the proof of the first claim. $\qquad\square$

We now proceed to derive the upper bound of the Wasserstein distance between the sample distribution generated after $N$ iterations and the target distribution $p_0$, based on the one-step discretization error bound given by Proposition 9.

First, note that

$$
\mathsf{W}_2(\mathcal{L}(\vartheta_N^{\mathsf{EM}}), p_0) \leqslant \left\|\vartheta_N^{\mathsf{EM}} - X_0\right\|_{\mathbb{L}_2} \leqslant \left\|Y_{Nh} - \vartheta_N^{\mathsf{EM}}\right\|_{\mathbb{L}_2} + \left\|Y_{Nh} - X_0\right\|_{\mathbb{L}_2}.
$$

Invoking Proposition 7 of [13], we have

$$\left\| Y_{Nh} - X_0 \right\|_{\mathbb{L}_2} \leqslant e^{-\int_0^T m(t)\, dt} \left\| X_0 \right\|_{\mathbb{L}_2}. \tag{20}$$

According to Proposition 9, by induction, we obtain that

$$
\begin{aligned}
\left\| Y_{Nh} - \vartheta_N^{\mathsf{EM}} \right\|_{\mathbb{L}_2} &\leqslant r_{N-1}^{\mathsf{EM}} \left\| Y_{(N-1)h} - \vartheta_{N-1}^{\mathsf{EM}} \right\|_{\mathbb{L}_2} + \left( h^2 C_{N-1}^{\mathsf{EM}} + h^{3/2}\sqrt{d} C_1(N-1) + h\varepsilon_{sc} \right) \\
&\leqslant \left( \prod_{j=0}^{N-1} r_j^{\mathsf{EM}} \right) \left\| Y_0 - \vartheta_0^{\mathsf{EM}} \right\|_{\mathbb{L}_2} + \sum_{k=0}^{N-1} \left( \prod_{j=k+1}^{N-1} r_j^{\mathsf{EM}} \right) \left( h^2 C_k^{\mathsf{EM}} + h^{3/2}\sqrt{d} C_1(k) + h\varepsilon_{sc} \right) \\
&= \sum_{k=0}^{N-1} \left( \prod_{j=k+1}^{N-1} r_j^{\mathsf{EM}} \right) \left( h^2 C_k^{\mathsf{EM}} + h^{3/2}\sqrt{d} C_1(k) + h\varepsilon_{sc} \right),
\end{aligned}
\tag{21}
$$

where we define $\prod_{j=N}^{N-1} r_j^{\mathsf{EM}} = 1$. Notice that

$$
\begin{aligned}
\prod_{j=k+1}^{N-1} r_j^{\mathsf{EM}} &= \prod_{j=k+1}^{N-1} \left( e^{-\int_{jh}^{(j+1)h} (m(T-t)-\frac{1}{2})\, dt} + h^2 (C_1(k)^2 + M_1) \right) \\
&\lesssim \prod_{j=k+1}^{N-1} e^{-h(m_{\min}-\frac{1}{2})} = e^{-h(m_{\min}-\frac{1}{2})(N-k-1)}.
\end{aligned}
$$

Therefore, we have

$$
\begin{aligned}
\left\| Y_{Nh} - \vartheta_N^{\mathsf{EM}} \right\|_{\mathbb{L}_2} &\lesssim \sum_{k=0}^{N-1} e^{-h(m_{\min}-\frac{1}{2})(N-k-1)} \left( h^2 C_k^{\mathsf{EM}} + h^{3/2}\sqrt{d} C_1(k) + h\varepsilon_{sc} \right) \\
&\leqslant \frac{1}{1 - e^{-h(m_{\min}-\frac{1}{2})}} \left( h^2 \max_{0\leqslant k \leqslant N-1} C_k^{\mathsf{EM}} + h^{3/2}\sqrt{d} \max_{0\leqslant k \leqslant N-1} C_1(k) + h\varepsilon_{sc} \right) \\
&\lesssim \frac{1}{m_{\min} - 1/2} \left( \sqrt{d}h \max_{0\leqslant k \leqslant N-1} C_1(k) + \varepsilon_{sc} \right).
\end{aligned}
\tag{22}
$$

Recall the definition of $C_1(k)$ and the upper bound of $L(t)$, it follows that

$$\max_{0\leqslant k \leqslant N-1} C_1(k) \leqslant \frac{1}{2} + L_{\max},$$

and thus we obtain that

$$\left\| Y_{Nh} - \vartheta_N^{\mathsf{EM}} \right\|_{\mathbb{L}_2} \lesssim \sqrt{d}h \cdot \frac{L_{\max} + 1/2}{m_{\min} - 1/2} + \varepsilon_{sc} \cdot \frac{1}{m_{\min} - 1/2}.$$

Plugging this back into the previous display then we have

$$\mathsf{W}_2(\mathcal{L}(\vartheta_N^{\mathsf{EM}}), p_0) \lesssim e^{-\int_0^T m(t)\, dt} \left\| X_0 \right\|_{\mathbb{L}_2} + \sqrt{d}h \cdot \frac{L_{\max} + 1/2}{m_{\min} - 1/2} + \varepsilon_{sc} \cdot \frac{1}{m_{\min} - 1/2},$$

which completes the first part of the proof for Theorem 1.

## B.2 Proof of Theorem 1: Part II

This part aims to prove the Wasserstein convergence result for the Exponential Integrator (EI) scheme. We will prove this theorem using the same method as in Theorem 1. Following this approach, we first establish the one-step discretization error in the proposition below.

**Proposition 11.** *Suppose that Assumption 1, 3 and 2 hold, then one-step discretization error for Exponential Integrator scheme is obtained from the following two bounds.*

*(1)* *It holds that*

$$\left\|\tilde{Y}_h - \vartheta_{n+1}^{\mathsf{EI}}\right\|_{\mathbb{L}_2} \leqslant h^2 \left(C_5(n)C_1(n) + M_1 \frac{2(e^{h/2}-1)}{h}\right) \left\|Y_{nh} - \vartheta_n^{\mathsf{EM}}\right\|_{\mathbb{L}_2}$$

$$+ h^2 \left[C_5(n)\left(C_1(n)C_2(n) + \frac{1}{2}C_4 + C_3(n)\right) + \frac{2(e^{h/2}-1)}{h}M_1(1 + C_2(n) + C_4)\right]$$

$$+ h^{3/2}\sqrt{d}C_5(n)$$

$$+ h \cdot \frac{2(e^{h/2}-1)}{h}\varepsilon_{sc},$$

*where*

$$C_5(n) = \frac{1}{h}\int_{nh}^{(n+1)h} e^{\frac{1}{2}((n+1)h-t)}L(T-t)\,\mathrm{d}t \approx C_1(n) - \frac{1}{2}.$$

*(2)* *Therefore, we have the bound for one-step discretization error*

$$\left\|Y_{(n+1)h} - \vartheta_{n+1}^{\mathsf{EI}}\right\|_{\mathbb{L}_2} \leqslant r_n^{\mathsf{EI}}\left\|Y_{nh} - \vartheta_n^{\mathsf{EI}}\right\|_{\mathbb{L}_2} + h^2 C_n^{\mathsf{EI}} + h^{3/2}\sqrt{d}C_5(n) + h \cdot \frac{2(e^{h/2}-1)}{h}\varepsilon_{sc},$$

*where*

$$r_n^{\mathsf{EI}} = e^{-\int_{nh}^{(n+1)h}(m(T-t)-\frac{1}{2})\,\mathrm{d}t} + h^2\left(C_5(n)C_1(n) + M_1\frac{2(e^{h/2}-1)}{h}\right),$$

$$C_n^{\mathsf{EI}} = C_5(n)\left(C_1(n)C_2(n) + \frac{1}{2}C_4 + C_3(n)\right) + \frac{2(e^{h/2}-1)}{h}M_1(1 + C_2(n) + C_4).$$

*Here, the constants $C_i, i = 1, 2, 3, 4$ are as defined in Proposition 9.*

*Proof.* We prove two claims in succession.

**Proof of Claim (1).** Consider the process defined in (12), which satisfies the SDE

$$\mathrm{d}\tilde{Y}_t = \left[\frac{1}{2}\tilde{Y}_t + \nabla\log p_{T-nh-t}(\tilde{Y}_t)\right]\mathrm{d}t + \mathrm{d}W_t,$$

Instead of integrating both sides of the SDE, we use Itô's formula to $e^{-\frac{t}{2}}\tilde{Y}_t$, then we have

$$\mathrm{d}(e^{-\frac{t}{2}}\tilde{Y}_t) = -\frac{1}{2}e^{-\frac{t}{2}}\tilde{Y}_t + e^{-\frac{t}{2}}\mathrm{d}\tilde{Y}_t = e^{-\frac{t}{2}}\left(\nabla\log p_{T-nh-t}(\tilde{Y}_t)\,\mathrm{d}t + \mathrm{d}W_t\right),$$

and we notice that we can write it in an integral form.

$$\tilde{Y}_t = e^{t/2}\tilde{Y}_0 + \int_0^t e^{\frac{1}{2}(t-s)}\nabla\log p_{T-nh-s}(\tilde{Y}_s)\,\mathrm{d}s + \int_{nh}^{nh+t} e^{\frac{1}{2}((n+1)h-s)}\,\mathrm{d}W_s.$$

Then we obtain that

$$\tilde{Y}_h - \vartheta_{n+1}^{\mathsf{EI}} = \int_0^h e^{\frac{1}{2}(h-t)}(\nabla\log p_{T-nh-t}(\tilde{Y}_t) - s_*(T-nh, \vartheta_n^{\mathsf{EI}}))\,\mathrm{d}t.$$

We make decomposition the same as the one in (15), that is

$$\nabla\log p_{T-nh-t}(\tilde{Y}_t) - s_*(T-nh, \vartheta_n^{\mathsf{EI}}) = \nabla\log p_{T-nh-t}(\tilde{Y}_t) - \nabla\log p_{T-nh-t}(\tilde{Y}_0)$$

$$+ \nabla\log p_{T-nh-t}(\vartheta_n^{\mathsf{EI}}) - \nabla\log p_{T-nh}(\vartheta_n^{\mathsf{EI}})$$

$$+ \nabla\log p_{T-nh}(\vartheta_n^{\mathsf{EI}}) - s_*(T-nh, \vartheta_n^{\mathsf{EI}}).$$

It then follows that

$$\left\|\tilde{Y}_h - \vartheta_{n+1}^{\mathsf{EI}}\right\|_{\mathbb{L}_2} \leqslant \int_0^h e^{\frac{1}{2}(h-t)}\left\|\nabla\log p_{T-nh-t}(\tilde{Y}_t) - s_*(T-nh, \vartheta_n^{\mathsf{EI}})\right\|_{\mathbb{L}_2}\mathrm{d}t$$

$$\leqslant \int_0^h e^{\frac{1}{2}(h-t)}\left\|\nabla\log p_{T-nh-t}(\tilde{Y}_t) - \nabla\log p_{T-nh-t}(\tilde{Y}_0)\right\|_{\mathbb{L}_2}\mathrm{d}t$$

$$+ \int_0^h e^{\frac{1}{2}(h-t)}\left\|\nabla\log p_{T-nh-t}(\vartheta_n^{\mathsf{EI}}) - \nabla\log p_{T-nh}(\vartheta_n^{\mathsf{EI}})\right\|_{\mathbb{L}_2}\mathrm{d}t$$

$$+ \int_0^h e^{\frac{1}{2}(h-t)}\left\|\nabla\log p_{T-nh}(\vartheta_n^{\mathsf{EI}}) - s_*(T-nh, \vartheta_n^{\mathsf{EI}})\right\|_{\mathbb{L}_2}\mathrm{d}t.$$

Note that apart from the exponential term $e^{\frac{1}{2}(h-t)}$, the derivation of the remaining parts is completely consistent with that of (15), until we encounter the term involving $\vartheta_n^{\mathsf{EI}}$, at which point we obtain

$$\left\|\tilde{Y}_h - \vartheta_{n+1}^{\mathsf{EI}}\right\|_{\mathbb{L}_2} \leqslant \left(\int_0^h e^{\frac{1}{2}(h-t)}L(T - nh - t)\,\mathrm{d}t\right) \sup_{0\leqslant t\leqslant h}\left\|\tilde{Y}_t - \tilde{Y}_0\right\|_{\mathbb{L}_2}$$
$$+ \left(\int_0^h e^{\frac{1}{2}(h-t)}\,\mathrm{d}t\right) M_1 h\left(1 + \left\|\vartheta_n^{\mathsf{EI}}\right\|_{\mathbb{L}_2}\right)$$
$$+ \left(\int_0^h e^{\frac{1}{2}(h-t)}\,\mathrm{d}t\right)\varepsilon_{sc}.$$

By Lemma 10, we can bound the first term on the right-hand side of the previous display. Moreover, from (16), $\left\|\vartheta_n^{\mathsf{EI}}\right\|_{\mathbb{L}_2}$ can be bounded similarly. Substituting all coefficients with $C_i(n)$ from Proposition 9, we obtain

$$\left\|\tilde{Y}_h - \vartheta_{n+1}^{\mathsf{EI}}\right\|_{\mathbb{L}_2} \leqslant h^2 \cdot C_5(n)\left[C_1(n)\left\|Y_{nh} - \vartheta_n^{\mathsf{EI}}\right\|_{\mathbb{L}_2} + C_1(n)C_2(n) + \frac{1}{2}C_4 + C_3(n)\right]$$
$$+ h^2 \cdot \frac{2(e^{h/2} - 1)}{h}M_1\left[1 + \left\|Y_{nh} - \vartheta_n^{\mathsf{EM}}\right\|_{\mathbb{L}_2} + C_2(n) + C_4\right]$$
$$+ h^{3/2}\sqrt{d}C_5(n)$$
$$+ h \cdot \frac{2(e^{h/2} - 1)}{h}\varepsilon_{sc}$$
$$= h^2\left(C_5(n)C_1(n) + M_1\frac{2(e^{h/2} - 1)}{h}\right)\left\|Y_{nh} - \vartheta_n^{\mathsf{EM}}\right\|_{\mathbb{L}_2}$$
$$+ h^2\left[C_5(n)\left(C_1(n)C_2(n) + \frac{1}{2}C_4 + C_3(n)\right) + \frac{2(e^{h/2} - 1)}{h}M_1(1 + C_2(n) + C_4)\right]$$
$$+ h^{3/2}\sqrt{d}C_5(n)$$
$$+ h \cdot \frac{2(e^{h/2} - 1)}{h}\varepsilon_{sc},$$

where

$$C_5(n) = \frac{1}{h}\int_{nh}^{(n+1)h} e^{\frac{1}{2}((n+1)h - t)}L(T - t)\,\mathrm{d}t \approx C_1(n) - \frac{1}{2}.$$

**Proof of Claim (2).** The proof is omitted for brevity, as it merely requires incorporating $\left\|Y_{(n+1)h} - \tilde{Y}_h\right\|_{\mathbb{L}_2}$ into the conclusion of Claim (1), following a similar argument as in the proof of Claim (2) in Proposition 9.

$\square$

For the second part of the proof for Theorem 1, recall that in the first part, the three key steps (20), (21) and (22) lead to the desired result. We now revisit these steps within the framework of other discretization schemes.

Since (20) is independent of the discretization scheme, we can directly apply it throughout the proofs of Theorems 1, 3 and 4. For (21), we note that the $h^2$ term in $r_j^\alpha$ is neglected, which results in the same upper bound for $\prod_{j=k+1}^{N-1} r_j^\alpha$ across all discretization schemes.

Given the consistency of these two steps, for the remaining discretization schemes, we can directly derive an analogue of (22) from Claim (2). Therefore, in the subsequent proofs of these theorems, after establishing the corresponding proposition, we proceed directly from an expression similar to (22).

For this theorem, we begin the proof with the following inequality

$$\left\|Y_{Nh} - \vartheta_N^{\mathsf{EI}}\right\|_{\mathbb{L}_2} \lesssim \frac{1}{m_{\min} - 1/2}\left(hC_n^{\mathsf{EI}} + h^{1/2}\sqrt{d}\max_{0\leqslant k\leqslant N-1}C_5(k) + \varepsilon_{sc}\right)$$

$$\lesssim \frac{1}{m_{\min} - 1/2}\left(\sqrt{dh}\max_{0\leqslant k\leqslant N-1}C_5(k) + \varepsilon_{sc}\right)$$

$$\leqslant \sqrt{dh}\cdot\frac{L_{\max}}{m_{\min} - 1/2} + \varepsilon_{sc}\cdot\frac{1}{m_{\min} - 1/2}.$$

Combining this with the bound of $\|X_0 - Y_{Nh}\|_{\mathbb{L}_2}$, we obtain

$$\mathsf{W}_2(\mathcal{L}(\vartheta_N^{\mathsf{EI}}), p_0) \lesssim e^{-m_{\min}T}\|X_0\|_{\mathbb{L}_2} + \sqrt{dh}\cdot\frac{L_{\max}}{m_{\min} - 1/2} + \varepsilon_{sc}\cdot\frac{1}{m_{\min} - 1/2}$$

as desired.

### B.3  Proof of Theorem 3: Part I

In this section, we prove the Wasserstein distance between the generated distribution $\mathcal{L}(\vartheta_N^{\mathsf{REM}})$ and the target distribution. The following proposition is established for the one-step discretization error.

**Proposition 12.** *Suppose that Assumptions 1, 2 and 4 are satisfied, the following two claims hold.*

*(1) It holds that*

$$\left\|\tilde{Y}_h - \vartheta_{n+1}^{\mathsf{REM}}\right\|_{\mathbb{L}_2}$$

$$\leqslant h^2\left\{\left[\int_0^1\int_0^1\left[|u-v|L(T-(n+u)h)\left(\frac{1}{2}+L(T-nh)\right)+M_1\right]^2 \mathrm{d}u\,\mathrm{d}v\right]^{1/2}\right.$$

$$\left.+\frac{1}{4\sqrt{3}}L(T-nh)+\frac{1}{8\sqrt{3}}\right\}\left\|Y_{nh}-\vartheta_n^{\mathsf{REM}}\right\|_{\mathbb{L}_2}$$

$$+h^2\left\{\left\{\int_0^1\int_0^1\left\{(u-v)\left[\left(\frac{1}{2}+L(T-nh)\right)C_2(n)+\frac{1}{2}C_4+(dL(T-nh))^{1/2}\right]L(T-(n+u)h)\right.\right.$$

$$\left.\left.+M_1(1+C_2(n)+C_4)\right\}^2 \mathrm{d}u\,\mathrm{d}v\right\}^{1/2}$$

$$\left.+\frac{1}{8\sqrt{3}}(C_2(n)+C_4)+\frac{1}{4\sqrt{3}}\left(L(T-nh)C_2(n)+(dL(T-nh))^{1/2}\right)\right\}$$

$$+h^{3/2}\left\{\sqrt{d}\left[\int_0^1\int_0^1 L(T-(n+u)h)^2|u-v|\,\mathrm{d}u\,\mathrm{d}v\right]^{1/2}+\frac{1}{2\sqrt{3}}\right\}$$

$$+2h\varepsilon_{sc}.$$

*(2) Moreover, it holds that*

$$\left\|Y_{(n+1)h}-\vartheta_{n+1}^{\mathsf{REM}}\right\| \leqslant r_n^{\mathsf{REM}}\left\|Y_{nh}-\vartheta_n\right\|_{\mathbb{L}_2}+h^2 C_{n,1}^{\mathsf{REM}}+h^{3/2}C_{n,2}^{\mathsf{REM}}+3h\varepsilon_{sc},$$

*where*

$$r_n^{\mathsf{REM}}=e^{-\int_{nh}^{(n+1)h}(m(T-t)-\frac{1}{2})\,\mathrm{d}t}$$

$$+h^2\left\{\left[\int_0^1\int_0^1\left[|u-v|L(T-(n+u)h)\left(\frac{1}{2}+L(T-nh)\right)+M_1\right]^2 \mathrm{d}u\,\mathrm{d}v\right]^{1/2}\right.$$

$$\left.+\frac{1}{4\sqrt{3}}L(T-nh)+\frac{1}{8\sqrt{3}}\right\},$$

$$C_{n,1}^{\mathsf{REM}} = \left\{ \int_0^1 \int_0^1 \left\{ (u-v) \left[ \left( \frac{1}{2} + L(T-nh) \right) C_2(n) + \frac{1}{2} C_4 + (dL(T-nh))^{1/2} \right] L(T-(n+u)h) \right. \right.$$
$$\left. \left. + M_1(1 + C_2(n) + C_4) \right\}^2 \, du \, dv \right\}^{1/2}$$
$$+ \frac{1}{8\sqrt{3}} (C_2(n) + C_4) + \frac{1}{4\sqrt{3}} \left( L(T-nh)C_2(n) + (dL(T-nh))^{1/2} \right) ,$$
$$C_{n,2}^{\mathsf{REM}} = \sqrt{d} \left[ \int_0^1 \int_0^1 L(T-(n+u)h)^2 |u-v| \, du \, dv \right]^{1/2} + \frac{1}{2\sqrt{3}} .$$

*Proof of Proposition 12.* **Proof of Claim (1).** We make the following decomposition of one-step discretization error

$$\left\| \tilde{Y}_h - \vartheta_{n+1}^{\mathsf{REM}} \right\|_{\mathbb{L}_2} \leqslant \left\| \tilde{Y}_h - \mathbb{E}_{U_n} \left[ \vartheta_{n+1}^{\mathsf{REM}} \right] \right\|_{\mathbb{L}_2} + \left\| \mathbb{E}_{U_n} \left[ \vartheta_{n+1}^{\mathsf{REM}} \right] - \vartheta_{n+1}^{\mathsf{REM}} \right\|_{\mathbb{L}_2} . \tag{23}$$

We first derive the upper bound for the term $\left\| \tilde{Y}_h - \mathbb{E}_{U_n} \left[ \vartheta_{n+1}^{\mathsf{REM}} \right] \right\|_{\mathbb{L}_2}$. By the definitions of $\vartheta_n^{\mathsf{REM}}$ and $\tilde{Y}_h$, we have

$$\left\| \tilde{Y}_h - \mathbb{E}_{U_n} \left[ \vartheta_{n+1}^{\mathsf{REM}} \right] \right\|_{\mathbb{L}_2}$$
$$= \left\| \frac{1}{2} \int_0^h \tilde{Y}_t \, dt + \int_0^h \nabla \log p_{T-nh-t}(\tilde{Y}_t) \, dt - \frac{1}{2} h \mathbb{E}_{U_n} (\vartheta_{n+U}^{\mathsf{REM}}) - h \mathbb{E}_{U_n} [s_*(T-(n+U_n)h, \vartheta_{n+U}^{\mathsf{REM}})] \right\|_{\mathbb{L}_2} .$$

Notice that

$$\int_0^h \tilde{Y}_t \, dt = h \mathbb{E}_{U_n} [\tilde{Y}_{U_n h}], \qquad \int_0^h \nabla \log p_{T-nh-t}(\tilde{Y}_t) \, dt = h \mathbb{E}_{U_n} [\nabla \log p_{T-(n+U_n)h}(\tilde{Y}_{U_n h})] .$$

Plugging this back into the previous display then gives

$$\left\| \tilde{Y}_h - \mathbb{E}_{U_n} [\vartheta_{n+1}^{\mathsf{REM}}] \right\|_{\mathbb{L}_2}$$
$$= \left\| \frac{1}{2} h \mathbb{E}_{U_n} [\tilde{Y}_{U_n h}] + h \mathbb{E}_{U_n} [\nabla \log p_{T-(n+U_n)h}(\tilde{Y}_{U_n h})] - \frac{1}{2} h \mathbb{E}_{U_n} (\vartheta_{n+U}^{\mathsf{REM}}) - h \mathbb{E}_{U_n} (s_*(T-(n+U_n)h, \vartheta_{n+U}^{\mathsf{REM}})) \right\|_{\mathbb{L}_2}$$
$$\leqslant \frac{1}{2} h \left\| \mathbb{E}_{U_n} [\tilde{Y}_{U_n h} - \vartheta_{n+U_n}^{\mathsf{REM}}] \right\|_{\mathbb{L}_2} + h \left\| \mathbb{E}_{U_n} [\nabla \log p_{T-(n+U_n)h}(\tilde{Y}_{U_n h}) - s_*(T-(n+U_n)h, \vartheta_{n+U_n}^{\mathsf{REM}})] \right\|_{\mathbb{L}_2} .$$

By the definition of $\tilde{Y}_{U_n h}$ and $\vartheta_{n+U_n}^{\mathsf{REM}}$, we have

$$\left\| \mathbb{E}_{U_n} [\tilde{Y}_{U_n h} - \vartheta_{n+U_n}^{\mathsf{REM}}] \right\|_{\mathbb{L}_2}$$
$$= \left\| \mathbb{E}_{U_n} \left[ \frac{1}{2} \int_0^{U_n h} (\tilde{Y}_t - \vartheta_n^{\mathsf{REM}}) \, dt + \int_0^{U_n h} (\nabla \log p_{T-nh-t}(\tilde{Y}_t) - s_*(T-nh, \vartheta_n^{\mathsf{REM}})) \, dt \right] \right\|_{\mathbb{L}_2}$$
$$\leqslant \left\| \mathbb{E}_{U_n} \left[ \frac{1}{2} \int_0^{U_n h} \left\| \tilde{Y}_t - \vartheta_n^{\mathsf{REM}} \right\| \, dt + \int_0^{U_n h} \left\| \nabla \log p_{T-nh-t}(\tilde{Y}_t) - s_*(T-nh, \vartheta_n^{\mathsf{REM}}) \right\| \, dt \right] \right\|_{\mathbb{L}_2}$$
$$\leqslant \left\| \mathbb{E}_{U_n} \left[ \frac{1}{2} \int_0^h \left\| \tilde{Y}_t - \vartheta_n^{\mathsf{REM}} \right\| \, dt + \int_0^h \left\| \nabla \log p_{T-nh-t}(\tilde{Y}_t) - s_*(T-nh, \vartheta_n^{\mathsf{REM}}) \right\| \, dt \right] \right\|_{\mathbb{L}_2}$$
$$\leqslant \frac{1}{2} \int_0^h \left\| \tilde{Y}_t - \vartheta_n^{\mathsf{REM}} \right\|_{\mathbb{L}_2} \, dt + \int_0^h \left\| \nabla \log p_{T-nh-t}(\tilde{Y}_t) - s_*(T-nh, \vartheta_n^{\mathsf{REM}}) \right\|_{\mathbb{L}_2} \, dt .$$

The second inequality arises because the integrand is non-negative, the last inequality follows from the fact that the random variables inside the inner expectation $\mathbb{E}_{U_n}$ are independent of $U_n$, and thus the inner expectation can be ignored. Then using the same argument as in the proof of Proposition 9,

especially adopting the same procedure as the one following (15), we can apply the conclusion of Proposition 9 to the term above, then we obtain that

$$\left\|\mathbb{E}_{U_n}\left[\tilde{Y}_{U_n h} - \vartheta_{n+U_n}^{\mathsf{REM}}\right]\right\|_{\mathbb{L}_2} \leqslant \left(\frac{1}{2}h + \int_{nh}^{(n+1)h} L(T-t)\,\mathrm{d}t\right) \sup_{0 \leqslant t \leqslant h} \left\|\tilde{Y}_t - \tilde{Y}_0\right\|_{\mathbb{L}_2}$$

$$+ \int_{nh}^{(n+1)h} \left\|\nabla \log p_{T-nh-t}(\vartheta_n^{\mathsf{REM}}) - s_*(T-nh, \vartheta_n^{\mathsf{REM}})\right\|_{\mathbb{L}_2}\,\mathrm{d}t$$

$$\leqslant h^2(C_1(n)^2 + M_1)\left\|Y_{nh} - \vartheta_n^{\mathsf{REM}}\right\|_{\mathbb{L}_2}$$

$$+ h^2\left[C_1(n)\left(C_1(n)C_2(n) + \frac{1}{2}C_4 + C_3(n)\right) + M_1(1 + C_2(n) + C_4)\right]$$

$$+ h^{3/2}\sqrt{d}C_1(n)$$

$$+ h\varepsilon_{sc}$$

$$\triangleq h^2 r_1 \left\|Y_{nh} - \vartheta_n^{\mathsf{REM}}\right\|_{\mathbb{L}_2} + h^2 r_2 + h^{3/2}\sqrt{d}C_1(n) + h\varepsilon_{sc},$$

where

$$r_1 = C_1(n)^2 + M_1,$$

$$r_2 = C_1(n)\left(C_1(n)C_2(n) + \frac{1}{2}C_4 + C_3(n)\right) + M_1(1 + C_2(n) + C_4).$$

We now derive the upper bound of the second term in (23). Note that

$$\left\|\mathbb{E}_{U_n}\left[\nabla \log p_{T-(n+U_n)h}(\tilde{Y}_{(n+U_n)h}) - s_*(T-(n+U_n)h, \vartheta_{n+U_n}^{\mathsf{REM}})\right]\right\|_{\mathbb{L}_2}$$

$$= \left\|\int_0^1 \left(\nabla \log p_{T-(n+u)h}(\tilde{Y}_{(n+u)h}) - s_*(T-(n+u)h, \vartheta_{n+u}^{\mathsf{REM}})\right)\mathrm{d}u\right\|_{\mathbb{L}_2}$$

$$\leqslant \int_0^1 \left\|\nabla \log p_{T-(n+u)h}(\tilde{Y}_{(n+u)h}) - s_*(T-(n+u)h, \vartheta_{n+u}^{\mathsf{REM}})\right\|_{\mathbb{L}_2}\,\mathrm{d}u$$

$$\leqslant \int_0^1 \left(\left\|\nabla \log p_{T-(n+u)h}(\tilde{Y}_{(n+u)h}) - \nabla \log p_{T-(n+u)h}(\vartheta_{n+u}^{\mathsf{REM}})\right\|_{\mathbb{L}_2}\right.$$

$$\left. + \left\|\nabla \log p_{T-(n+u)h}(\vartheta_{n+u}^{\mathsf{REM}}) - s_*(T-(n+u)h, \vartheta_{n+u}^{\mathsf{REM}})\right\|_{\mathbb{L}_2}\right)\mathrm{d}u$$

$$\leqslant \int_0^1 L(T-(n+u)h)\left\|\tilde{Y}_{(n+u)h} - \vartheta_{n+u}^{\mathsf{REM}}\right\|_{\mathbb{L}_2}\,\mathrm{d}u + \varepsilon_{sc},$$

(24)

the second inequality follows from the triangle inequality, and the last inequality depends on Assumption 1 and 4. By (17), changing the value of $h$ to $uh$, we have

$$\left\|\tilde{Y}_{(n+u)h} - \vartheta_{n+u}^{\mathsf{REM}}\right\|_{\mathbb{L}_2}$$

$$\leqslant (uh)^2(C_{1,n}(u)^2 + M_1)\left\|Y_{nh} - \vartheta_n^{\mathsf{REM}}\right\|_{\mathbb{L}_2}$$

$$+ (uh)^2\left[C_{1,n}(u)\left(C_{1,n}(u)C_2(n) + \frac{1}{2}C_4 + C_{3,n}(u)\right) + M_1(1 + C_2(n) + C_4)\right]$$

$$+ (uh)^{3/2}\sqrt{d}C_{1,n}(u)$$

$$+ uh\varepsilon_{sc},$$

(25)

where $C_{1,n}(u)$ and $C_{3,n}(u)$ is the $uh$-version of $C_1(n)$ and $C_3(n)$, respectively, that is

$$C_{1,n}(u) = \frac{1}{2} + \frac{1}{uh}\int_{nh}^{(n+u)h} L(T-t)\,\mathrm{d}t,$$

$$C_{3,n}(u) = \frac{1}{uh}\int_{nh}^{(n+u)h} (dL(T-t))^{1/2}\,\mathrm{d}t,$$

Plugging the previous display (25) back into display (24), then rearranging and simplifying the expression, yields

$$\left\| \mathbb{E}_{U_n} [\nabla \log p_{T-(n+U_n)h}(\tilde{Y}_{(n+U_n)h}) - s_*(T - (n + U_n)h, \vartheta^{\mathsf{REM}}_{n+U_n})] \right\|_{\mathbb{L}_2}$$

$$\leqslant h^2 \left( \int_0^1 L(T - (n + u)h)u^2(C_{1,n}(u)^2 + M_1)\, \mathrm{d}u \right) \left\| Y_{nh} - \vartheta^{\mathsf{EM}}_n \right\|_{\mathbb{L}_2}$$

$$+ h^2 \left\{ \int_0^1 L(T - (n + u)h)u^2 \left[ C_{1,n}(u) \left( C_{1,n}(u)C_2(n) + \frac{1}{2}C_4 + C_{3,n}(u) \right) + M_1(1 + C_2(n) + C_4) \right] \mathrm{d}u \right\}$$

$$+ h^{3/2} \left( \int_0^1 L(T - (n + u)h)u^{3/2}\, \mathrm{d}u \right) \sqrt{d}C_1(n)$$

$$+ h \left( \int_0^1 L(T - (n + u)h)u\, \mathrm{d}u \right) \varepsilon_{sc}$$

$$+ \varepsilon_{sc}$$

$$\triangleq h^2 r_3 \left\| Y_{nh} - \vartheta^{\mathsf{EM}}_n \right\|_{\mathbb{L}_2} + h^2 r_4 + h^{3/2} r_5 + h r_6 \varepsilon_{sc} + \varepsilon_{sc}\,,$$

where

$$r_3 = \int_0^1 L(T - (n + u)h)u^2(C_{1,n}(u)^2 + M_1)\, \mathrm{d}u\,,$$

$$r_4 = \int_0^1 L(T - (n + u)h)u^2 \left[ C_{1,n}(u) \left( C_{1,n}(u)C_2(n) + \frac{1}{2}C_4 + C_{3,n}(u) \right) \right] \mathrm{d}u + M_1(1 + C_2(n) + C_4)\,,$$

$$r_5 = \left( \int_0^1 L(T - (n + u)h)u^{3/2}\, \mathrm{d}u \right) \sqrt{d}C_1(n)\,,$$

$$r_6 = \int_0^1 L(T - (n + u)h)u\, \mathrm{d}u\,.$$

From the bounds we have obtained for two terms, it follows that

$$\left\| \tilde{Y}_h - \mathbb{E}_{U_n}[\vartheta^{\mathsf{REM}}_{n+1}] \right\|_{\mathbb{L}_2}$$

$$\leqslant h^3 (\frac{1}{2}r_1 + r_3) \left\| Y_{nh} - \vartheta^{\mathsf{EM}}_n \right\|_{\mathbb{L}_2} + h^3 (\frac{1}{2}r_2 + r_4) + h^{5/2}(\frac{1}{2}\sqrt{d}C_1(n) + r_5) + h^2(\frac{1}{2} + r_6)\varepsilon_{sc} + h\varepsilon_{sc}\,.$$

$$\tag{26}$$

Considering the second term of one-step discretization error

$$\vartheta^{\mathsf{REM}}_{n+1} - \mathbb{E}_{U_n}[\vartheta^{\mathsf{REM}}_{n+1}]$$

$$= \frac{1}{2}h \left[ \vartheta^{\mathsf{REM}}_{n+U} - \mathbb{E}_{U_n}[\vartheta^{\mathsf{REM}}_{n+U}] \right] + h \left[ s_*(T - (n + U_n)h, \vartheta^{\mathsf{REM}}_{n+U}) - \mathbb{E}_{U_n}[s_*(T - (n + U_n)h, \vartheta^{\mathsf{REM}}_{n+U})] \right]$$

$$= \frac{1}{2}h \left[ \frac{1}{2}h(U_n - \frac{1}{2})\vartheta^{\mathsf{REM}}_n + h(U_n - \frac{1}{2})s_*(T - nh, \vartheta^{\mathsf{REM}}_n) \right]$$

$$+ \frac{1}{2}h \left[ \int_{nh}^{(n+U_n)h} \mathrm{d}W_t - \int_0^1 \left( \int_{nh}^{(n+u)h} \mathrm{d}W_t \right) \mathrm{d}u \right]$$

$$+ h \left[ s_*(T - (n + U_n)h, \vartheta^{\mathsf{REM}}_{n+U}) - \mathbb{E}_{U_n}[s_*(T - (n + U_n)h, \vartheta^{\mathsf{REM}}_{n+U})] \right]\,.$$

$$\tag{27}$$

The second equality follows from the fact that

$$\mathbb{E}_{U_n}[\vartheta^{\mathsf{REM}}_{n+U}] = \vartheta^{\mathsf{REM}}_n + \frac{1}{2}h\mathbb{E}_{U_n}[U_n]\vartheta^{\mathsf{REM}}_n + h\mathbb{E}_{U_n}[U_n]s_*(T - nh, \vartheta^{\mathsf{REM}}_n) + \mathbb{E}_{U_n} \int_{nh}^{(n+U_n)h} \mathrm{d}W_t$$

$$= \vartheta^{\mathsf{REM}}_n + \frac{1}{4}h\vartheta^{\mathsf{REM}}_n + \frac{1}{2}hs_*(T - nh, \vartheta^{\mathsf{REM}}_n) + \int_0^1 \left( \int_{nh}^{(n+u)h} \mathrm{d}W_t \right) \mathrm{d}u\,,$$

since $U_n$ is independent of $\vartheta^{\mathsf{REM}}_n$.

We proceed to bound each term in (27). For the first term, still notice that the independence between

$U_n$ and $\vartheta_n^{\mathsf{REM}}$, then we find that

$$\left\|(U_n - \frac{1}{2})\vartheta_n^{\mathsf{REM}}\right\|_{\mathbb{L}_2} = \left\{\mathbb{E}\left[\mathbb{E}_{U_n}\left\|(U_n - \frac{1}{2})\vartheta_n^{\mathsf{REM}}\right\|^2\right]\right\}^{1/2}$$

$$= \left\{\mathbb{E}\left[\mathbb{E}_{U_n}[(U_n - \frac{1}{2})^2] \cdot \left\|\vartheta_n^{\mathsf{REM}}\right\|^2\right]\right\}^{1/2}$$

$$= \left\{\mathbb{E}\left[\frac{1}{12}\left\|\vartheta_n^{\mathsf{REM}}\right\|^2\right]\right\}^{1/2}$$

$$= \frac{1}{2\sqrt{3}}\left\|\vartheta_n^{\mathsf{REM}}\right\|_{\mathbb{L}_2}.$$

The bounding of another part of the first term follows in a similar manner, we obtain that

$$\left\|(U_n - \frac{1}{2})s_*(T - nh, \vartheta_n^{\mathsf{REM}})\right\|_{\mathbb{L}_2} = \frac{1}{2\sqrt{3}}\left\|s_*(T - nh, \vartheta_n^{\mathsf{REM}})\right\|_{\mathbb{L}_2}.$$

For the second term of (27), notice that due to Itô's isometry formula, for any well-defined stochastic process $X_t$ and its Itô stochastic integral $I_t(X) = \int_0^t X_u \, dM_u$, we have

$$\mathbb{E}[I_t(X)^2] = \mathbb{E}\int_0^t X_u^2 \, d\langle M\rangle_u, \tag{28}$$

then we can establish a lemma.

**Lemma 13.** *Suppose $W_t$ is a d-dim standard Brownian motion, then*

$$\left\|\int_{nh}^{(n+U_n)h} dW_t - \int_0^1\left(\int_{nh}^{(n+u)h} dW_t\right) du\right\|_{\mathbb{L}_2}^2 \leqslant \frac{h}{3}.$$

For the third term of (27), we get

$$\left\|s_*(T - (n + U_n)h, \vartheta_{n+U}^{\mathsf{REM}}) - \mathbb{E}_{U_n}[s_*(T - (n + U_n)h, \vartheta_{n+U}^{\mathsf{REM}})]\right\|_{\mathbb{L}_2}$$

$$= \left\|\int_0^1 s_*(T - (n + U_n)h, \vartheta_{n+U}^{\mathsf{REM}}) - s_*(T - (n + v)h, \vartheta_{n+v}^{\mathsf{REM}}) \, dv\right\|_{\mathbb{L}_2}$$

$$= \left\{\mathbb{E}\int_0^1\left[\int_0^1 s_*(T - (n + u)h, \vartheta_{n+u}^{\mathsf{REM}}) - s_*(T - (n + v)h, y_{(n+v)h}) \, dv\right]^2 du\right\}^{1/2}$$

$$\leqslant \left\{\mathbb{E}\int_0^1\int_0^1\left[s_*(T - (n + u)h, \vartheta_{n+u}^{\mathsf{REM}}) - s_*(T - (n + v)h, \vartheta_{n+v}^{\mathsf{REM}})\right]^2 du \, dv\right\}^{1/2}$$

$$= \left\{\int_0^1\int_0^1\left\|s_*(T - (n + u)h, \vartheta_{n+u}^{\mathsf{REM}}) - s_*(T - (n + v)h, \vartheta_{n+v}^{\mathsf{REM}})\right\|_{\mathbb{L}_2}^2 du \, dv\right\}^{1/2}.$$

Then by the triangle inequality and Assumption 4, we have

$$\left\|s_*(T - (n + u)h, \vartheta_{n+u}^{\mathsf{REM}}) - s_*(T - (n + v)h, \vartheta_{n+v}^{\mathsf{REM}})\right\|_{\mathbb{L}_2}$$

$$\leqslant \left\|s_*(T - (n + u)h, \vartheta_{n+u}^{\mathsf{REM}}) - \nabla\log p_{T-(n+u)h}(\vartheta_{n+u}^{\mathsf{REM}})\right\|_{\mathbb{L}_2}$$

$$+ \left\|s_*(T - (n + v)h, \vartheta_{n+v}^{\mathsf{REM}}) - \nabla\log p_{T-(n+v)h}(\vartheta_{n+v}^{\mathsf{REM}})\right\|_{\mathbb{L}_2} \tag{29}$$

$$+ \left\|\nabla\log p_{T-(n+u)h}(\vartheta_{n+u}^{\mathsf{REM}}) - \nabla\log p_{T-(n+v)h}(\vartheta_{n+v}^{\mathsf{REM}})\right\|_{\mathbb{L}_2}$$

$$\leqslant 2\varepsilon_{sc} + \left\|\nabla\log p_{T-(n+u)h}(\vartheta_{n+u}^{\mathsf{REM}}) - \nabla\log p_{T-(n+v)h}(\vartheta_{n+v}^{\mathsf{REM}})\right\|_{\mathbb{L}_2}.$$

Combining the three terms of (27) together, we have

$$\left\|\vartheta_{n+1}^{\mathsf{REM}} - \mathbb{E}_{U_n}[\vartheta_{n+1}^{\mathsf{REM}}]\right\|_{\mathbb{L}_2} \leqslant \frac{1}{8\sqrt{3}}h^2\left\|\vartheta_n^{\mathsf{REM}}\right\|_{\mathbb{L}_2} + \frac{1}{4\sqrt{3}}h^2\left\|s_*(T - nh, \vartheta_n^{\mathsf{REM}})\right\|_{\mathbb{L}_2} + \frac{1}{2\sqrt{3}}h^{3/2}$$

$$+ h\left\{\int_0^1\int_0^1\left\|\nabla\log p_{T-(n+v)h}(\vartheta_{n+v}^{\mathsf{REM}}) - \nabla\log p_{T-(n+u)h}(\vartheta_{n+v}^{\mathsf{REM}})\right\|_{\mathbb{L}_2}^2 du \, dv\right\}^{1/2}$$

$$+ 2h\varepsilon_{sc}.$$

By applying the same technique used in the proofs of Proposition 9 and Proposition 11, the upper bounds for $\left\|\vartheta_n^{\mathsf{REM}}\right\|_{\mathbb{L}_2}$ and $\left\|s_*(T - nh, \vartheta_n^{\mathsf{REM}})\right\|_{\mathbb{L}_2}$ follows readily. Thus, the proposition follows immediately from the bound on the second last term. We now consider the case for $u > v$, due to Assumptions 1 and 2,

$$\left\|\nabla \log p_{T-(n+u)h}(\vartheta_{n+u}^{\mathsf{REM}}) - \nabla \log p_{T-(n+v)h}(\vartheta_{n+v}^{\mathsf{REM}})\right\|_{\mathbb{L}_2}$$
$$\leqslant L(T - (n+u)h) \left\|\vartheta_{n+u}^{\mathsf{REM}} - \vartheta_{n+v}^{\mathsf{REM}}\right\|_{\mathbb{L}_2} + M_1 h \left(1 + \left\|\vartheta_{n+v}^{\mathsf{REM}}\right\|_{\mathbb{L}_2}\right).$$

Since

$$\left\|\vartheta_{n+u}^{\mathsf{REM}} - \vartheta_{n+v}^{\mathsf{REM}}\right\|_{\mathbb{L}_2} \leqslant \frac{1}{2}(u-v)h \left\|\vartheta_n^{\mathsf{REM}}\right\|_{\mathbb{L}_2} + (u-v)h \left\|s_*(T - nh, \vartheta_n^{\mathsf{REM}})\right\|_{\mathbb{L}_2} + \left\|\int_{(n+v)h}^{(n+u)h} \mathrm{d}W_t\right\|_{\mathbb{L}_2}$$

$$\leqslant \frac{1}{2}(u-v)h \left(\left\|Y_{nh} - \vartheta_n^{\mathsf{REM}}\right\|_{\mathbb{L}_2} + C_2(n) + C_4\right)$$
$$+ (u-v)h \left[\varepsilon_{sc} + L(T - nh) \left(\left\|Y_{nh} - \vartheta_n^{\mathsf{REM}}\right\|_{\mathbb{L}_2} + C_2(n)\right) + (dL(T - nh))^{1/2}\right]$$
$$+ \sqrt{(u-v)h}$$

$$\leqslant (u-v)h \left[\frac{1}{2} + L(T - nh)\right] \left\|Y_{nh} - \vartheta_n^{\mathsf{REM}}\right\|_{\mathbb{L}_2}$$
$$+ (u-v)h \left[(\frac{1}{2} + L(T - nh))C_2(n) + \frac{1}{2}C_4 + (dL(T - nh))^{1/2}\right]$$
$$+ \sqrt{(u-v)dh} + (u-v)h\varepsilon_{sc}.$$

The second inequality follows from (16), Assumptions 1, 3 and Lemma 18. Similarly,

$$\left\|\vartheta_{n+v}^{\mathsf{REM}}\right\|_{\mathbb{L}_2} \leqslant \left\|\vartheta_{n+v}^{\mathsf{REM}} - \vartheta_n^{\mathsf{REM}}\right\|_{\mathbb{L}_2} + \left\|\vartheta_n^{\mathsf{REM}}\right\|_{\mathbb{L}_2}$$

$$\leqslant vh \left[\frac{1}{2} + L(T - nh)\right] \left\|Y_{nh} - \vartheta_n^{\mathsf{REM}}\right\|_{\mathbb{L}_2}$$
$$+ vh \left[(\frac{1}{2} + L(T - nh))C_2(n) + \frac{1}{2}C_4 + (dL(T - nh))^{1/2}\right]$$
$$+ \sqrt{vdh} + vh\varepsilon_{sc}$$
$$+ \left\|Y_{nh} - \vartheta_n^{\mathsf{REM}}\right\|_{\mathbb{L}_2} + C_2(n) + C_4.$$

Therefore, we obtain that

$$\left\|\nabla \log p_{T-(n+u)h}(\vartheta_{n+u}^{\mathsf{REM}}) - \nabla \log p_{T-(n+v)h}(\vartheta_{n+v}^{\mathsf{REM}})\right\|_{\mathbb{L}_2}$$
$$\leqslant h \left\{(u-v) \left[\frac{1}{2} + L(T - nh)\right] L(T - (n+u)h) + M_1\right\} \left\|Y_{nh} - \vartheta_n^{\mathsf{REM}}\right\|_{\mathbb{L}_2}$$
$$+ h^2 M_1 v \left[\frac{1}{2} + L(T - nh)\right] \left\|Y_{nh} - \vartheta_n^{\mathsf{REM}}\right\|_{\mathbb{L}_2}$$
$$+ h^2 v M_1 \left[\left(\frac{1}{2} + L(T - nh)\right) C_2(n) + \frac{1}{2}C_4 + (dL(T - nh))^{1/2}\right]$$
$$+ h^{3/2} M_1 \sqrt{vd}$$
$$+ h \left\{(u-v) \left[\left(\frac{1}{2} + L(T - nh)\right) C_2(n) + \frac{1}{2}C_4 + (dL(T - nh))^{1/2}\right] L(T - (n+u)h)\right.$$
$$\left. + M_1(1 + C_2(n) + C_4)\right\}$$
$$+ h^{1/2} L(T - (n+u)h) \sqrt{(u-v)d}$$
$$+ h(u-v)L(T - (n+u)h)\varepsilon_{sc} + h^2 M_1 v\varepsilon_{sc}.$$

We claim that we only consider the lowest order of each part, which means the relative higher order term with the combination of $d$ and $h$ will be ignored. Then take the supremum with respect to $v$,

$$\left\{ \int_0^1 \int_0^1 \left\| \nabla \log p_{T-(n+v)h}(\vartheta_{n+v}^{\mathsf{REM}}) - \nabla \log p_{T-(n+u)h}(\vartheta_{n+v}^{\mathsf{REM}}) \right\|_{\mathbb{L}_2}^2 \, \mathrm{d}u \, \mathrm{d}v \right\}^{1/2}$$

$$\leqslant h \left\{ \int_0^1 \int_0^1 \left[ |u-v| L(T-(n+u)h)\left(\frac{1}{2} + L(T-nh)\right) + M_1 \right]^2 \mathrm{d}u \, \mathrm{d}v \right\}^{1/2} \left\| Y_{nh} - \vartheta_n^{\mathsf{REM}} \right\|_{\mathbb{L}_2}$$

$$+ h^{1/2}\sqrt{d} \left[ \int_0^1 \int_0^1 L(T-(n+u)h)^2 |u-v| \, \mathrm{d}u \, \mathrm{d}v \right]^{1/2}$$

$$+ h \left[ \int_0^1 \int_0^1 (u-v)^2 L(T-(n+u)h)^2 \, \mathrm{d}u \, \mathrm{d}v \right]^{1/2} \varepsilon_{sc}.$$

$$(30)$$

Combining the above,

$$\left\| \vartheta_{n+1}^{\mathsf{REM}} - \mathbb{E}_{U_n}[\vartheta_{n+1}^{\mathsf{REM}}] \right\|_{\mathbb{L}_2}$$

$$\leqslant \frac{1}{8\sqrt{3}} h^2 \left( \left\| Y_{nh} - \vartheta_n^{\mathsf{REM}} \right\|_{\mathbb{L}_2} + C_2(n) + C_4 \right)$$

$$+ \frac{1}{4\sqrt{3}} h^2 \left[ \varepsilon_{sc} + L(T-nh) \left( \left\| Y_{nh} - \vartheta_n^{\mathsf{REM}} \right\|_{\mathbb{L}_2} + C_2(n) \right) + (dL(T-nh))^{1/2} \right]$$

$$+ \frac{1}{2\sqrt{3}} h^{3/2}$$

$$+ h^2 \left\{ \int_0^1 \int_0^1 \left[ |u-v| L(T-(n+u)h)\left(\frac{1}{2} + L(T-nh)\right) + M_1 \right]^2 \mathrm{d}u \, \mathrm{d}v \right\}^{1/2} \left\| Y_{nh} - \vartheta_n^{\mathsf{REM}} \right\|_{\mathbb{L}_2}$$

$$+ h^2 \left\{ \int_0^1 \int_0^1 \left\{ (u-v)\left[ \left(\frac{1}{2} + L(T-nh)\right) C_2(n) + \frac{1}{2}C_4 + (dL(T-nh))^{1/2} \right] L(T-(n+u)h) \right.\right.$$

$$\left.\left. + M_1(1 + C_2(n) + C_4) \right\}^2 \mathrm{d}u \, \mathrm{d}v \right\}^{1/2}$$

$$+ h^{3/2}\sqrt{d} \left[ \int_0^1 \int_0^1 L(T-(n+u)h)^2 |u-v| \, \mathrm{d}u \, \mathrm{d}v \right]^{1/2}$$

$$+ h^2 \left[ \int_0^1 \int_0^1 (u-v)^2 L(T-(n+u)h)^2 \, \mathrm{d}u \, \mathrm{d}v \right]^{1/2} \varepsilon_{sc}$$

$$+ 2h\varepsilon_{sc}$$

$$\lesssim h^2 \left\{ \left[ \int_0^1 \int_0^1 \left[ |u-v| L(T-(n+u)h)\left(\frac{1}{2} + L(T-nh)\right) + M_1 \right]^2 \mathrm{d}u \, \mathrm{d}v \right]^{1/2} \right.$$

$$\left. + \frac{1}{4\sqrt{3}} L(T-nh) + \frac{1}{8\sqrt{3}} \right\} \left\| Y_{nh} - \vartheta_n^{\mathsf{REM}} \right\|_{\mathbb{L}_2}$$

$$+ h^2 \left\{ \left\{ \int_0^1 \int_0^1 \left\{ (u-v)\left[ \left(\frac{1}{2} + L(T-nh)\right) C_2(n) + \frac{1}{2}C_4 + (dL(T-nh))^{1/2} \right] L(T-(n+u)h) \right.\right.\right.$$

$$\left.\left. + M_1(1 + C_2(n) + C_4) \right\}^2 \mathrm{d}u \, \mathrm{d}v \right\}^{1/2}$$

$$\left. + \frac{1}{8\sqrt{3}}(C_2(n) + C_4) + \frac{1}{4\sqrt{3}}(L(T-nh)C_2(n) + (dL(T-nh))^{1/2}) \right\}$$

$$+ h^{3/2} \left\{ \sqrt{d} \left[ \int_0^1 \int_0^1 L(T-(n+u)h)^2 |u-v| \, \mathrm{d}u \, \mathrm{d}v \right]^{1/2} + \frac{1}{2\sqrt{3}} \right\}$$

$$+ 2h\varepsilon_{sc}.$$

$$(31)$$

Compared to the term $\tilde{Y}_h - \mathbb{E}_{U_n}[\vartheta_{n+1}^{\mathsf{REM}}]$, we can focus on the lower-order terms, ignoring the score matching error. Therefore, we have

$$\left\| \tilde{Y}_{(n+1)h} - \vartheta_{n+1}^{\mathsf{REM}} \right\|_{\mathbb{L}_2}$$

$$\leqslant h^2 \left\{ \left[ \int_0^1 \int_0^1 \left[ |u-v| L(T-(n+u)h)\left(\frac{1}{2} + L(T-nh)\right) + M_1 \right]^2 \mathrm{d}u\,\mathrm{d}v \right]^{1/2} \left\| Y_{nh} - \vartheta_n^{\mathsf{REM}} \right\|_{\mathbb{L}_2} \right.$$

$$\left. + \frac{1}{4\sqrt{3}} L(T-nh) + \frac{1}{8\sqrt{3}} \right\}$$

$$+ h^2 \left\{ \left\{ \int_0^1 \int_0^1 \left\{ (u-v) \left[ \left(\frac{1}{2} + L(T-nh)\right) C_2(n) + \frac{1}{2} C_4 + (dL(T-nh))^{1/2} \right] L(T-(n+u)h) \right. \right. \right.$$

$$\left. \left. + M_1(1 + C_2(n) + C_4) \right\}^2 \mathrm{d}u\,\mathrm{d}v \right\}^{1/2}$$

$$\left. + \frac{1}{8\sqrt{3}}(C_2(n) + C_4) + \frac{1}{4\sqrt{3}}\left( L(T-nh)C_2(n) + (dL(T-nh))^{1/2} \right) \right\}$$

$$+ h^{3/2} \left\{ \sqrt{d} \left[ \int_0^1 \int_0^1 L(T-(n+u)h)^2 |u-v|\,\mathrm{d}u\,\mathrm{d}v \right]^{1/2} + \frac{1}{2\sqrt{3}} \right\}$$

$$+ 3h\varepsilon_{sc}.$$

$\square$

Returning to the proof of Theorem 3, by the conclusion of Proposition 12, we have

$$\left\| Y_{Nh} - \vartheta_N^{\mathsf{REM}} \right\|_{\mathbb{L}_2} \lesssim \frac{1}{m_{\min} - 1/2} \left( h \max_{0 \leqslant k \leqslant N-1} C_{k,1}^{\mathsf{REM}} + h^{1/2} \max_{0 \leqslant k \leqslant N-1} C_{k,2}^{\mathsf{REM}} + 3\varepsilon_{sc} \right)$$

$$\lesssim \sqrt{h} \cdot \frac{\sqrt{d/3}L_{\max} + \frac{1}{2\sqrt{3}}}{m_{\min} - 1/2} + \varepsilon_{sc} \cdot \frac{3}{m_{\min} - 1/2}.$$

This completes the first part of proof for Theorem 3.

### B.4  Proof of Theorem 3: Part II

We begin with the following proposition.

**Proposition 14.** *Suppose that Assumptions 1, 2 and 4 are satisfied, the following two claims hold*

*(1) It holds that*

$$\left\| \tilde{Y}_h - \vartheta_{n+1}^{\mathsf{REI}} \right\|_{\mathbb{L}_2}$$

$$\leqslant h^2 \left\{ \int_0^1 \int_0^1 \left[ |u-v| L(T-(n+u)h)\left(\frac{1}{2} + L(T-nh)\right) + M_1 \right. \right.$$

$$\left. \left. + \frac{1}{2}|u-v| L(T-(n+v)h) r_n^{\mathsf{EI}}(v) \right]^2 \mathrm{d}u\,\mathrm{d}v \right\}^{1/2} \left\| Y_{nh} - \vartheta_n^{\mathsf{REI}} \right\|_{\mathbb{L}_2}$$

$$+ h^2 \left\{ \frac{e^{\frac{1}{2}(1-v)h} - e^{\frac{1}{2}(1-u)h}}{h} e^{\frac{1}{2}vh} L(T-(n+u)h) \left[ C_2(n) + C_4 + 2L(T-nh)C_2(n) + (dL(T-nh))^{1/2} \right] \right.$$

$$+ e^{\frac{1}{2}(1-u)h} M_1 \left[ 1 + 2e^{\frac{1}{2}vh}\left( L(T-nh)C_2(n) + (dL(T-nh))^{1/2} \right) + C_2(n) + C_4 \right]$$

$$\left. + \frac{|e^{\frac{1}{2}(1-u)h} - e^{\frac{1}{2}(1-v)h}|}{h} \left( L(T-(n+v)h)C_2(n) + (dL(T-(n+v)h))^{1/2} \right) \right\}$$

$$+ h^{3/2} \sqrt{d} \left\{ \int_0^1 \int_0^1 L(T-(n+u)h)^2 |u-v| \, \mathrm{d}u \, \mathrm{d}v \right\}^{1/2}$$

$$+ 3h\varepsilon_{sc} \, .$$

*(2) Furthermore, it holds that*

$$\left\| Y_{(n+1)h} - \vartheta_{n+1}^{\mathsf{REI}} \right\|_{\mathbb{L}_2} \leqslant r_n^{\mathsf{REI}} \left\| Y_{nh} - \vartheta_n^{\mathsf{REI}} \right\|_{\mathbb{L}_2} + h^2 C_{n,1}^{\mathsf{REI}} + h^{3/2} C_{n,2}^{\mathsf{REI}} + 3h\varepsilon_{sc} \, ,$$

*where*

$$r_n^{\mathsf{REI}} = e^{-\int_{nh}^{(n+1)h}(m(T-t)-\frac{1}{2}) \, \mathrm{d}t}$$

$$+ h^2 \left\{ \int_0^1 \int_0^1 \left[ |u-v| L(T-(n+u)h)(\frac{1}{2} + L(T-nh)) \right. \right.$$

$$\left. \left. + M_1 + \frac{1}{2} |u-v| L(T-(n+v)h) r_n^{\mathsf{EI}}(v) \right]^2 \, \mathrm{d}u \, \mathrm{d}v \right\}^{1/2} \, ,$$

$$C_{n,1}^{\mathsf{REI}} = \frac{e^{\frac{1}{2}(1-v)h} - e^{\frac{1}{2}(1-u)h}}{h} e^{\frac{1}{2}vh} L(T-(n+u)h) \left[ C_2(n) + C_4 + 2L(T-nh)C_2(n) + (dL(T-nh))^{1/2} \right]$$

$$+ e^{\frac{1}{2}(1-u)h} M_1 \left[ 1 + 2e^{\frac{1}{2}vh} \left( L(T-nh)C_2(n) + (dL(T-nh))^{1/2} \right) + C_2(n) + C_4 \right]$$

$$+ \frac{|e^{\frac{1}{2}(1-u)h} - e^{\frac{1}{2}(1-v)h}|}{h} \left( L(T-(n+v)h)C_2(n) + (dL(T-(n+v)h))^{1/2} \right) \, ,$$

$$C_{n,2}^{\mathsf{REI}} = \sqrt{d} \left\{ \int_0^1 \int_0^1 L(T-(n+u)h)^2 |u-v| \, \mathrm{d}u \, \mathrm{d}v \right\}^{1/2} \, .$$

*Proof of Proposition 14.* This proposition can be proven following the same approach as in the proof of Proposition 12, with the only difference being the inclusion of the exponential coefficient term. However, this term does not significantly affect the overall proof.

Similarly, we make a decomposition as

$$\left\| \tilde{Y}_h - \vartheta_{n+1}^{\mathsf{REI}} \right\|_{\mathbb{L}_2} \leqslant \left\| \tilde{Y}_h - \mathbb{E}_{U_n}[\vartheta_{n+1}^{\mathsf{REI}}] \right\|_{\mathbb{L}_2} + \left\| \mathbb{E}_{U_n}[\vartheta_{n+1}^{\mathsf{REI}}] \right\|_{\mathbb{L}_2} \, . \tag{32}$$

Note that

$$\tilde{Y}_h - \mathbb{E}_{U_n}[\vartheta_{n+1}^{\mathsf{REI}}] = \int_0^h e^{\frac{1}{2}(h-t)} \nabla \log p_{T-nh-t}(\tilde{Y}_t) \, \mathrm{d}t - h\mathbb{E}_{U_n}\left[ e^{\frac{1}{2}(1-U_n)h} s_*(T-nh-U_nh, \vartheta_{n+U_n}^{\mathsf{REI}}) \right]$$

$$= h \int_0^1 e^{\frac{1}{2}(1-u)h} \left( \nabla \log p_{T-nh-uh}(\tilde{Y}_{uh}) - s_*(T-nh-uh, \vartheta_{n+u}^{\mathsf{REI}}) \right) \, \mathrm{d}u$$

$$= h \int_0^1 e^{\frac{1}{2}(1-u)h} \left( \nabla \log p_{T-nh-uh}(\tilde{Y}_{uh}) - \nabla \log p_{T-nh-uh}(\vartheta_{n+u}^{\mathsf{REI}}) \right) \, \mathrm{d}u$$

$$+ h \int_0^1 e^{\frac{1}{2}(1-u)h} \left( \nabla \log p_{T-nh-uh}(\vartheta_{n+u}^{\mathsf{REI}}) - s_*(T-nh-uh, \vartheta_{n+u}^{\mathsf{REI}}) \right) \, \mathrm{d}u \, .$$

Then, we obtain

$$\left\| \tilde{Y}_h - \mathbb{E}_{U_n}[\vartheta_{n+1}^{\mathsf{REI}}] \right\|_{\mathbb{L}_2}$$

$$\leqslant h \int_0^1 \left\| e^{\frac{1}{2}(1-u)h}(\nabla \log p_{T-nh-uh}(\tilde{Y}_{uh}) - s_*(T-nh-uh, \vartheta_{n+u}^{\mathsf{REI}})) \right\|_{\mathbb{L}_2} \, \mathrm{d}u$$

$$\leqslant h \int_0^1 e^{\frac{1}{2}(1-u)h} L(T-nh-uh) \left\| \tilde{Y}_{uh} - \vartheta_{n+u}^{\mathsf{REI}} \right\|_{\mathbb{L}_2} \, \mathrm{d}u + h \int_0^1 e^{\frac{1}{2}(1-u)h} \, \mathrm{d}u \, \varepsilon_{sc}$$

$$\leqslant h^3 \int_0^1 e^{\frac{1}{2}(1-u)h} L(T-nh-uh) u^2 \left( C_{5,n}(u)C_{1,n}(u) + M_1 \frac{2(e^{uh/2}-1)}{uh} \right) \, \mathrm{d}u \left\| Y_{nh} - \vartheta_n^{\mathsf{REI}} \right\|_{\mathbb{L}_2}$$

$$+ h^3 \int_0^1 e^{\frac{1}{2}(1-u)h} L(T-nh-uh) u^2 \left[ C_{5,n}(u) \left( C_{1,n}(u)C_2(n) + \frac{1}{2}C_4 + C_{3,n}(u) \right) \right.$$

$$+ \frac{2(e^{uh} - 1)}{uh} M_1(1 + C_2(n) + C_4) \Bigg] \mathrm{d}u$$

$$+ h^{5/2} \int_0^1 e^{\frac{1}{2}(1-u)h} L(T - nh - uh) u^{3/2} C_{5,n}(u) \, \mathrm{d}u \, \sqrt{d}$$

$$+ h^2 \int_0^1 e^{\frac{1}{2}(1-u)h} L(T - nh - uh) \frac{2(e^{uh/2} - 1)}{h} \, \mathrm{d}u \, \varepsilon_{sc} + h \frac{2(e^{h/2} - 1)}{h} \varepsilon_{sc} \,, \tag{33}$$

where

$$C_{5,n}(u) = \frac{1}{uh} \int_{nh}^{(n+u)h} e^{\frac{1}{2}((n+u)h - t)} L(T - t) \, \mathrm{d}t \,.$$

In the third inequality, we can directly bound $\left\| \tilde{Y}_{uh} - \vartheta_{n+u}^{\mathsf{REI}} \right\|_{\mathbb{L}_2}$, as it is a special case of Proposition 11, where the step size is replaced by $uh$.

For the second term of (32), we have

$$\vartheta_{n+1}^{\mathsf{REI}} - \mathbb{E}_{U_n}[\vartheta_{n+1}^{\mathsf{REI}}]$$

$$= h e^{\frac{1}{2}(1-U_n)h} s_*(T - nh - U_n h, \vartheta_{n+U}^{\mathsf{REI}}) - h \mathbb{E}_{U_n}\left[ e^{\frac{1}{2}(1-U_n)h} s_*(T - nh - U_n h, \vartheta_{n+U}^{\mathsf{REI}}) \right]$$

$$= h \int_0^1 \left[ e^{\frac{1}{2}(1-U_n)h} s_*(T - nh - U_n h, \vartheta_{n+U_n}^{\mathsf{REI}}) - e^{\frac{1}{2}(1-v)h} s_*(T - nh - vh, \vartheta_{n+v}^{\mathsf{REI}}) \right] \mathrm{d}v \,.$$

Similar to display (29), we then obtain

$$\left\| \vartheta_{n+1}^{\mathsf{REI}} - \mathbb{E}_{U_n}[\vartheta_{n+1}^{\mathsf{REI}}] \right\|_{\mathbb{L}_2}$$

$$\leqslant \left\{ \mathbb{E} \int_0^1 \left[ h \int_0^1 e^{\frac{1}{2}(1-u)h} s_*(T - (n+u)h, \vartheta_{n+u}^{\mathsf{REI}}) - e^{\frac{1}{2}(1-v)h} s_*(T - (n+v)h, \vartheta_{n+v}^{\mathsf{REI}}) \, \mathrm{d}v \right]^2 \mathrm{d}u \right\}^{1/2}$$

$$\leqslant h \left\{ \int_0^1 \int_0^1 \left\| e^{\frac{1}{2}(1-u)h} s_*(T - (n+u)h, \vartheta_{n+u}^{\mathsf{REI}}) - e^{\frac{1}{2}(1-v)h} s_*(T - (n+v)h, \vartheta_{n+v}^{\mathsf{REI}}) \right\|_{\mathbb{L}_2}^2 \mathrm{d}u \, \mathrm{d}v \right\}^{1/2}$$

$$\leqslant h \left\{ \int_0^1 \int_0^1 \left\| e^{\frac{1}{2}(1-u)h} \nabla \log p_{T-(n+u)h}(\vartheta_{n+u}^{\mathsf{REI}}) - e^{\frac{1}{2}(1-v)h} \nabla \log p_{T-(n+v)h}(\vartheta_{n+v}^{\mathsf{REI}}) \right\|_{\mathbb{L}_2}^2 \mathrm{d}u \, \mathrm{d}v \right\}^{1/2}$$

$$+ 2h \left( \int_0^1 e^{(1-u)h} \, \mathrm{d}u \right)^{1/2} \varepsilon_{sc} \,,$$

Using the same strategy as in display (30), we arrive at

$$\left\| e^{\frac{1}{2}(1-u)h} \nabla \log p_{T-(n+u)h}(\vartheta_{n+u}^{\mathsf{REI}}) - e^{\frac{1}{2}(1-v)h} \nabla \log p_{T-(n+v)h}(\vartheta_{n+v}^{\mathsf{REI}}) \right\|_{\mathbb{L}_2}$$

$$\leqslant e^{\frac{1}{2}(1-u)h} \left\| \nabla \log p_{T-(n+u)h}(\vartheta_{n+u}^{\mathsf{REI}}) - \nabla \log p_{T-(n+u)h}(\vartheta_{n+v}^{\mathsf{REI}}) \right\|_{\mathbb{L}_2}$$

$$+ e^{\frac{1}{2}(1-u)h} \left\| \nabla \log p_{T-(n+u)h}(\vartheta_{n+v}^{\mathsf{REI}}) - \nabla \log p_{T-(n+v)h}(\vartheta_{n+v}^{\mathsf{REI}}) \right\|_{\mathbb{L}_2}$$

$$+ \left| e^{\frac{1}{2}(1-u)h} - e^{\frac{1}{2}(1-v)h} \right| \left\| \nabla \log p_{T-(n+v)h}(\vartheta_{n+v}^{\mathsf{REI}}) \right\|_{\mathbb{L}_2}$$

$$\leqslant e^{\frac{1}{2}(1-u)h} L(T - (n+u)h) \left\| \vartheta_{n+u}^{\mathsf{REI}} - \vartheta_{n+v}^{\mathsf{REI}} \right\|_{\mathbb{L}_2}$$

$$+ e^{\frac{1}{2}(1-u)h} M_1 h (1 + \left\| \vartheta_{n+v}^{\mathsf{REI}} \right\|_{\mathbb{L}_2})$$

$$+ \left| e^{\frac{1}{2}(1-u)h} - e^{\frac{1}{2}(1-v)h} \right| \left[ L(T - (n+v)h) \left( \left\| Y_{(n+v)h} - \vartheta_{n+v}^{\mathsf{REI}} \right\|_{\mathbb{L}_2} + C_2(n) \right) + (dL(T - (n+v)h))^{1/2} \right] \,.$$

The second inequality follows from Assumptions 1 and 2. We bound the term $\left\| \nabla \log p_{T-(n+v)h}(\vartheta_{n+v}^{\mathsf{REI}}) \right\|_{\mathbb{L}_2}$ by decomposing it as follows

$$\left\| \nabla \log p_{T-(n+v)h}(\vartheta_{n+v}^{\mathsf{REI}}) \right\|_{\mathbb{L}_2} \leqslant \left\| \nabla \log p_{T-(n+v)h}(\vartheta_{n+v}^{\mathsf{REI}}) - \nabla \log p_{T-(n+v)h}(Y_{(n+v)h}) \right\|_{\mathbb{L}_2}$$

$$+ \left\| \nabla \log p_{T-(n+v)h}(Y_{(n+v)h}) - \nabla \log p_{T-(n+v)h}(X_{((n+v)h)}^{\leftarrow}) \right\|_{\mathbb{L}_2}$$

$$+ \left\| \nabla \log p_{T-(n+v)h}(X_{T-(n+v)h}) \right\|_{\mathbb{L}_2} \,.$$

Without loss of generality, we consider the case where $u > v$; the other case follows similarly.

$$\left\|\vartheta_{n+u}^{\mathsf{REI}} - \vartheta_{n+v}^{\mathsf{REI}}\right\|_{\mathbb{L}_2}$$

$$= (e^{\frac{1}{2}uh} - e^{\frac{1}{2}vh})\left\|\vartheta_n^{\mathsf{REI}}\right\|_{\mathbb{L}_2} + \int_{vh}^{uh} e^{\frac{1}{2}t}\,\mathrm{d}t\left\|s_*(T - nh, \vartheta_n^{\mathsf{REI}})\right\|_{\mathbb{L}_2}$$

$$+ \left\|\int_{nh}^{(n+u)h} e^{\frac{1}{2}((n+u)h-t)}\,\mathrm{d}W_t - \int_{nh}^{(n+v)h} e^{\frac{1}{2}((n+v)h-t)}\,\mathrm{d}W_t\right\|_{\mathbb{L}_2}$$

$$\leqslant \left(e^{\frac{1}{2}(u-v)h} - 1\right)e^{\frac{1}{2}vh}\left(\left\|Y_{nh} - \vartheta_n^{\mathsf{REI}}\right\|_{\mathbb{L}_2} + C_2(n) + C_4\right)$$

$$+ 2\left(e^{\frac{1}{2}(u-v)h} - 1\right)e^{\frac{1}{2}vh}\left[\varepsilon_{sc} + L(T - nh)\left(\left\|Y_{nh} - \vartheta_n^{\mathsf{REI}}\right\|_{\mathbb{L}_2} + C_2(n)\right) + (dL(T - nh))^{1/2}\right]$$

$$+ \left[(e^{uh} - 1) + (e^{vh} - 1) - 2(e^{\frac{u+v}{2}h} - e^{\frac{u-v}{2}h})\right]^{1/2}\sqrt{d}\,. \tag{34}$$

Here, we apply the formula in (28) to bound the last term.

$$\left\|\int_{nh}^{(n+u)h} \mathbf{1}_{\{t\leqslant(n+v)h\}}(e^{\frac{1}{2}((n+u)h-t)} - e^{\frac{1}{2}((n+v)h-t)}) + \mathbf{1}_{\{t>(n+v)h\}}e^{\frac{1}{2}((n+u)h-t)}\,\mathrm{d}W_t\right\|_{\mathbb{L}_2}$$

$$= \sqrt{d}\left[\int_{nh}^{(n+v)h} (e^{\frac{1}{2}((n+u)h-t)} - e^{\frac{1}{2}((n+v)h-t)})^2\,\mathrm{d}t + \int_{(n+v)h}^{(n+u)h} e^{(n+u)h-t}\,\mathrm{d}t\right]^{1/2}$$

$$= \sqrt{d}\left[(e^{uh} - 1) + (e^{vh} - 1) - 2(e^{\frac{u+v}{2}h} - e^{\frac{u-v}{2}h})\right]^{1/2},$$

We then bound the term $\left\|\vartheta_{n+v}^{\mathsf{REI}}\right\|_{\mathbb{L}_2}$ following display (34) above. To this end, let $u = 0$, we then have

$$\left\|\vartheta_{n+v}^{\mathsf{REI}}\right\|_{\mathbb{L}_2} \leqslant (e^{\frac{1}{2}vh} - 1)(\left\|Y_{nh} - \vartheta_n^{\mathsf{REI}}\right\|_{\mathbb{L}_2} + C_2(n) + C_4)$$

$$+ 2(e^{\frac{1}{2}vh} - 1)\left[\varepsilon_{sc} + L(T - nh)\left(\left\|Y_{nh} - \vartheta_n^{\mathsf{REI}}\right\|_{\mathbb{L}_2} + C_2(n)\right) + (dL(T - nh))^{1/2}\right]$$

$$+ \sqrt{d}(e^{vh} - 1)^{1/2}$$

$$+ \left\|Y_{nh} - \vartheta_n^{\mathsf{REI}}\right\|_{\mathbb{L}_2} + C_2(n) + C_4\,.$$

Additionally, we can bound $\left\|Y_{(n+v)h} - \vartheta_{n+v}^{\mathsf{REI}}\right\|_{\mathbb{L}_2}$, as it is a special case of the one-step discretization error under the Exponential Integrator scheme, where the step size is replaced by $vh$. Specifically, we have

$$\left\|Y_{(n+v)h} - \vartheta_{n+v}^{\mathsf{REI}}\right\|_{\mathbb{L}_2}$$

$$\leqslant r_n^{\mathsf{EI}}(v)\left\|Y_{nh} - \vartheta_n^{\mathsf{REI}}\right\|_{\mathbb{L}_2} + h^2 C_n^{\mathsf{EI}}(v) + h^{3/2}u^{3/2}\sqrt{d}C_{5,n}(v) + vh\frac{2(e^{\frac{1}{2}vh} - 1)}{vh}\varepsilon_{sc}\,,$$

where

$$r_n^{\mathsf{EI}}(v) = e^{-\int_{nh}^{(n+v)h}(m(T-t)-\frac{1}{2})\,\mathrm{d}t} + v^2h^2\left(C_{5,n}(v)C_{1,n}(v) + M_1\frac{2(e^{\frac{1}{2}vh} - 1)}{vh}\right),$$

$$C_n^{\mathsf{EI}}(v) = C_{5,n}(v)\left(C_{1,n}(v)C_2(n) + \frac{1}{2}C_4 + C_{3,n}(v)\right) + \frac{2(e^{\frac{1}{2}vh} - 1)}{vh}M_1(1 + C_2(n) + C_4)\,.$$

Then, we obtain

$$\left\|e^{\frac{1}{2}(1-u)h}\nabla\log p_{T-(n+u)h}(\vartheta_{n+u}^{\mathsf{REI}}) - e^{\frac{1}{2}(1-v)h}\nabla\log p_{T-(n+v)h}(\vartheta_{n+v}^{\mathsf{REI}})\right\|_{\mathbb{L}_2}$$

$$\leqslant \left\{2(e^{\frac{1}{2}(1-v)h} - e^{\frac{1}{2}(1-u)h})e^{\frac{1}{2}vh}L(T - (n + u)h)\left[\frac{1}{2} + L(T - nh)\right]\right.$$

$$+ e^{\frac{1}{2}(1-u)h}M_1h\left[(e^{\frac{1}{2}vh} - 1) + 2(e^{\frac{1}{2}vh} - 1)L(T - nh) + 1\right]$$

$$+ \left| e^{\frac{1}{2}(1-u)h} - e^{\frac{1}{2}(1-v)h} \right| L(T - (n+v)h) r_n^{\mathsf{EI}}(v) \Bigg\} \left\| Y_{nh} - \vartheta_n^{\mathsf{REI}} \right\|_{\mathbb{L}_2}$$

$$+ h^3 L(T - (n+v)h) \frac{|e^{\frac{1}{2}(1-u)h} - e^{\frac{1}{2}(1-v)h}|}{h} C_n^{\mathsf{EI}}(v)$$

$$+ h^{5/2} L(T - (n+v)h) \frac{|e^{\frac{1}{2}(1-u)h} - e^{\frac{1}{2}(1-v)h}|}{h} u^{3/2} \sqrt{d} C_{5,n}(v)$$

$$+ h^2 M_1 e^{\frac{1}{2}(1-u)h} \frac{e^{\frac{1}{2}vh} - 1}{h} (C_2(n) + C_4)$$

$$+ h^{3/2} M_1 e^{\frac{1}{2}(1-u)h} \sqrt{vd} \left( \frac{e^{vh} - 1}{vh} \right)^{1/2}$$

$$+ h \Bigg\{ \frac{e^{\frac{1}{2}(1-v)h} - e^{\frac{1}{2}(1-u)h}}{h} e^{\frac{1}{2}vh} L(T - (n+u)h) \left[ C_2(n) + C_4 + 2L(T - nh)C_2(n) + (dL(T-nh))^{1/2} \right]$$

$$+ e^{\frac{1}{2}(1-u)h} M_1 \left[ 1 + 2e^{\frac{1}{2}vh} \left( L(T - nh)C_2(n) + (dL(T-nh))^{1/2} \right) + C_2(n) + C_4 \right]$$

$$+ \frac{|e^{\frac{1}{2}(1-u)h} - e^{\frac{1}{2}(1-v)h}|}{h} \left( L(T - (n+v)h)C_2(n) + (dL(T - (n+v)h))^{1/2} \right) \Bigg\}$$

$$+ h^{1/2} L(T - (n+u)h) e^{\frac{1}{2}(1-u)h} \left[ \frac{(e^{uh} - 1) + (e^{vh} - 1) - 2(e^{\frac{u+v}{2}h} - e^{\frac{u-v}{2}h})}{h} \right]^{1/2} \sqrt{d}$$

$$+ h^2 \varepsilon_{sc} L(T - (n+v)h) \frac{|e^{\frac{1}{2}(1-u)h} - e^{\frac{1}{2}(1-v)h}|}{h} \frac{2(e^{\frac{1}{2}vh} - 1)}{h}$$

$$+ h\varepsilon_{sc} \cdot 2e^{\frac{1}{2}(1-u)h} \left[ L(T - (n+u)h) \frac{e^{\frac{1}{2}uh} - e^{\frac{1}{2}vh}}{h} + M_1 e^{\frac{1}{2}vh} \right].$$

Ignoring the higher-order terms, we take the supremum with respect to $v$ and substitute it back into the original expression, yielding

$$\left\| \vartheta_{n+1}^{\mathsf{REI}} - \mathbb{E}_{U_n}[\vartheta_{n+1}^{\mathsf{REI}}] \right\|_{\mathbb{L}_2}$$

$$\leqslant h \Bigg\{ \int_0^1 \int_0^1 \left\| e^{\frac{1}{2}(1-u)h} s_*(T - (n+u)h, \vartheta_{n+u}^{\mathsf{REI}}) - e^{\frac{1}{2}(1-v)h} s_*(T - (n+v)h, \vartheta_{n+v}^{\mathsf{REI}}) \right\|_{\mathbb{L}_2}^2 \, du \, dv \Bigg\}^{1/2}$$

$$\leqslant h^2 \Bigg\{ \int_0^1 \int_0^1 \Big[ |u-v| L(T - (n+u)h) \left( \frac{1}{2} + L(T - nh) \right) + M_1$$

$$+ \frac{1}{2} |u-v| L(T - (n+v)h) r_n^{\mathsf{EI}}(v) \Big]^2 \, du \, dv \Bigg\}^{1/2} \left\| Y_{nh} - \vartheta_n^{\mathsf{REI}} \right\|_{\mathbb{L}_2}$$

$$+ h^{3/2} \Bigg\{ \int_0^1 \int_0^1 dL(T - (n+u)h)^2 |u-v| \, du \, dv \Bigg\}^{1/2} + 2h\varepsilon_{sc}.$$

$$(35)$$

This completes the proof.

$\square$

Now, we have

$$\left\| Y_{Nh} - \vartheta_N^{\mathsf{REI}} \right\|_{\mathbb{L}_2} \lesssim \frac{1}{m_{\min} - 1/2} \left( h \max_{0 \leqslant k \leqslant N-1} C_{n,1}^{\mathsf{REI}} + \sqrt{h} \max_{0 \leqslant k \leqslant N-1} C_{n,2}^{\mathsf{REI}} + 3\varepsilon_{sc} \right)$$

$$\lesssim \sqrt{dh} \frac{L_{\max}}{\sqrt{3}(m_{\min} - 1/2)} + \varepsilon_{sc} \frac{3}{m_{\min} - 1/2}.$$

The desired result follows readily.

## C  The proof of the upper bound of error of the second-order acceleration scheme

This section is dedicated to proving the Wasserstein convergence result for second-order acceleration. To this end, we first establish the following proposition.

**Proposition 15.** *Suppose that Assumptions 1, 3, 5, 6, 7 are satisfied, the following results hold.*

*(1) First, we have an upper bound for $\left\|\tilde{Y}_h - \vartheta^{\mathsf{SO}}_{n+1}\right\|_{\mathbb{L}_2}$ as follows,*

$$
\left\|\tilde{Y}_h - \vartheta^{\mathsf{SO}}_{n+1}\right\|_{\mathbb{L}_2} \leqslant A_{n,1} e^{(L(nh)-\frac{1}{2})h} h^2 \left\|Y_{nh} - \vartheta^{\mathsf{SO}}_n\right\|_{\mathbb{L}_2} + A_{n,2} e^{(L(nh)-\frac{1}{2})h} h^2
$$
$$
+ \left(h\varepsilon_{sc} + \frac{2}{3}\sqrt{d}h^{3/2}\varepsilon^{(L)}_{sc} + \frac{1}{2}h^2\varepsilon^{(M)}_{sc}\right) e^{(L(T-nh)-\frac{1}{2})h},
$$

*where*

$$
A_{n,1} = \sup_{nh\leqslant t\leqslant(n+1)h} \frac{1}{t^2} \int_0^t \left(\int_0^s \left[(1+L(T-nh-u))L(T-nh-u)\right.\right.
$$
$$
\left.\left. + (1+L(T-nh))L(T-nh)\right] \mathrm{d}u\right) \mathrm{d}s,
$$

$$
A_{n,2} = \sup_{nh\leqslant t\leqslant(n+1)h} \frac{1}{t^2} \left[\int_0^t \left(\int_0^s \left[(1+L(T-nh-u))L(T-nh-u)\right.\right.\right.
$$
$$
\left.\left. + (1+L(T-nh))L(T-nh)\right] \mathrm{d}u\right) \mathrm{d}s \cdot C_2(n)
$$
$$
+ \sqrt{d} \int_0^t \left(\int_0^s \left[\left(\frac{1}{2}+L(T-nh-u)\right)L(T-nh-u)^{1/2}\right.\right.
$$
$$
\left.\left. + \left(\frac{1}{2}+L(T-nh)\right)L(T-nh)^{1/2}\right] \mathrm{d}u\right) \mathrm{d}s
$$
$$
\left. + \int_0^t \left(\int_0^s \frac{1}{2}(L(T-nh-u)+L(T-nh)) \mathrm{d}u\right) \mathrm{d}s \cdot C_4\right]
$$
$$
+ \frac{\sqrt{2}}{4}\sqrt{d}L_F.
$$

*(2) Furthermore, it holds that*

$$
\left\|Y_{(n+1)h} - \vartheta^{\mathsf{SO}}_{n+1}\right\|_{\mathbb{L}_2} \leqslant r^{\mathsf{SO}}_n \left\|Y_{nh} - \vartheta^{\mathsf{SO}}_n\right\|_{\mathbb{L}_2} + h^2 C^{\mathsf{SO}}_n
$$
$$
+ \left[h\varepsilon_{sc} + \frac{2}{3}\sqrt{d}h^{3/2}\varepsilon^{(L)}_{sc} + \frac{1}{2}h^2\varepsilon^{(M)}_{sc}\right] e^{(L(T-nh)-\frac{1}{2})h},
$$

*where*

$$
r^{\mathsf{SO}}_n = e^{-\int_{nh}^{(n+1)h}(m(T-t)-\frac{1}{2})\,\mathrm{d}t} + h^2 A_{n,1} e^{(L(T-nh)-\frac{1}{2})h},
$$
$$
C^{\mathsf{SO}}_n = A_{n,2} e^{(L(T-nh)-\frac{1}{2})h}.
$$

*Proof.* Recall the expression in display (44), which states that

$$
x_t = \vartheta^{\mathsf{SO}}_n + \int_{nh}^t \left(\frac{1}{2}\vartheta^{\mathsf{SO}}_n + \nabla\log p_{T-nh}(\vartheta^{\mathsf{SO}}_n) + L_n(x_s - \vartheta^{\mathsf{SO}}_n) + M_n(s-nh)\right) \mathrm{d}s + \int_{nh}^t \mathrm{d}W_s
$$

with

$$
L_n = \frac{1}{2}I_d + \nabla^2\log p_{T-nh}(\vartheta^{\mathsf{SO}}_n) \in \mathbb{R}^{d\times d},
$$
$$
M_n = \frac{1}{2}\sum_{j=1}^d \frac{\partial^2}{\partial x_j^2}\nabla\log p_{T-nh}(\vartheta^{\mathsf{SO}}_n) - \frac{\partial}{\partial t}\nabla\log p_{T-nh}(\vartheta^{\mathsf{SO}}_n) \in \mathbb{R}^d.
$$

Plugging the estimates of $\nabla \log p_{T-nh}(\vartheta_n^{\mathsf{SO}})$, $L_n$ and $M_n$ into the previous display yields the following process for $x_t^{\mathsf{SO}}$

$$x_t^{\mathsf{SO}} = \vartheta_n^{\mathsf{SO}} + \int_{nh}^t \frac{1}{2}\vartheta_n^{\mathsf{SO}} + s_*(T-nh, \vartheta_n^{\mathsf{SO}})\,\mathrm{d}s$$

$$+ \int_{nh}^t s_*^{(L)}(T-nh, \vartheta_n^{\mathsf{SO}})(x_s^{\mathsf{SO}} - \vartheta_n^{\mathsf{SO}}) + s_*^{(M)}(T-nh, \vartheta_n^{\mathsf{SO}})(s-nh)\,\mathrm{d}s + \int_{nh}^t \mathrm{d}W_s\,.$$

Then, we obtain

$$x_t^{\mathsf{SO}} - x_t = (t-nh)(s_*(T-nh, \vartheta_n^{\mathsf{SO}}) - \nabla \log p_{T-nh}(\vartheta_n^{\mathsf{SO}}))$$

$$+ (s_*^{(L)}(T-nh, \vartheta_n^{\mathsf{SO}}) - L_n)\int_{nh}^t (x_s^{\mathsf{SO}} - \vartheta_n^{\mathsf{SO}})\,\mathrm{d}s + L_n \int_{nh}^t (x_s^{\mathsf{SO}} - x_s)\,\mathrm{d}s$$

$$+ (s_*^{(M)}(T-nh, \vartheta_n^{\mathsf{SO}}) - M_n) \cdot \frac{1}{2}(t-nh)^2.$$

Notice that

$$\|L_n\|_{\mathbb{L}_2} = \left\|\frac{1}{2}I_d + \nabla^2 \log p_{T-nh}(\vartheta_n^{\mathsf{SO}})\right\|_{\mathbb{L}_2} \leqslant L(T-nh) - \frac{1}{2}\,. \tag{36}$$

Combining this with Assumptions 3 and 5 then provides us with

$$\|x_t^{\mathsf{SO}} - x_t\|_{\mathbb{L}_2}$$

$$\leqslant (t-nh)\varepsilon_{sc} + \varepsilon_{sc}^{(L)}\int_{nh}^t \|x_s^{\mathsf{SO}} - \vartheta_n^{\mathsf{SO}}\|_{\mathbb{L}_2}\,\mathrm{d}s + \|L_n\|_{\mathbb{L}_2}\int_{nh}^t \|x_s^{\mathsf{SO}} - x_s\|\,\mathrm{d}s + \frac{1}{2}(t-nh)^2\varepsilon_{sc}^{(M)}$$

$$\lesssim \left(L(T-nh) - \frac{1}{2}\right)\int_{nh}^t \|x_s^{\mathsf{SO}} - x_s\|\,\mathrm{d}s + (t-nh)\varepsilon_{sc} + \frac{2}{3}\sqrt{d}(t-nh)^{3/2}\varepsilon_{sc}^{(L)} + \frac{1}{2}(t-nh)^2\varepsilon_{sc}^{(M)}.$$

We omit the constant of term $\frac{2}{3}\sqrt{d}(t-nh)^{3/2}\varepsilon_{sc}^{(L)}$ in the last inequality. To handle the resulting integral inequality, we invoke the following Grönwall-type inequality.

**Lemma 16.** *Let $z(t) \geqslant t_0$ satisfy the following inequality:*

$$z(t) \leqslant \alpha(t) + \int_{t_0}^t \beta(s)z(s)\,\mathrm{d}s\,, \quad t \geqslant t_0\,,$$

*where $\beta(s)$ is non-negative, and $t_0$ is the initial time. Then, the solution $z(t)$ satisfies the following bound:*

$$z(t) \leqslant \alpha(t) + \int_{t_0}^t \alpha(s)\beta(s)\exp\left(\int_s^t \beta(r)\,\mathrm{d}r\right)\mathrm{d}s\,, \quad t \geqslant t_0\,.$$

*Additionally, if $\alpha(t)$ is non-decreasing function, then*

$$z(t) \leqslant \alpha(t)\exp\left(\int_{t_0}^t \beta(s)\,\mathrm{d}s\right)\,, \quad t \geqslant t_0\,.$$

Let

$$z(t) = \|x_t^{\mathsf{SO}} - x_t\|_{\mathbb{L}_2}\,,$$

$$\alpha(t) = (t-nh)\varepsilon_{sc} + \frac{2}{3}\sqrt{d}(t-nh)^{3/2}\varepsilon_{sc}^{(L)} + \frac{1}{2}(t-nh)^2\varepsilon_{sc}^{(M)}\,,$$

$$\beta(t) = L(T-nh) - \frac{1}{2}\,,$$

and set $t_0 = nh$. By Lemma 16, we have

$$\|\vartheta_{n+1}^{\mathsf{SO}} - x_{(n+1)h}\|_{\mathbb{L}_2} \leqslant \left(h\varepsilon_{sc} + \frac{2}{3}\sqrt{d}h^{3/2}\varepsilon_{sc}^{(L)} + \frac{1}{2}h^2\varepsilon_{sc}^{(M)}\right)e^{(L(T-nh)-\frac{1}{2})h}\,. \tag{37}$$

The original SDE can be rewritten as follows

$$\tilde{Y}_t = \tilde{Y}_0 + \int_0^t \left( \frac{1}{2}\tilde{Y}_s + \nabla \log p_{T-nh-s}(\tilde{Y}_s) \right) \mathrm{d}s + \int_{nh}^{nh+t} \mathrm{d}W_s \,.$$

Combining this with the definition of $x_t$ in (44), we then have

$$\tilde{Y}_t - x_{nh+t} = \int_0^t \left( \frac{1}{2}\tilde{Y}_s + \nabla \log p_{T-nh-s}(\tilde{Y}_s) - \frac{1}{2}\vartheta_n^{\mathsf{SO}} - \nabla \log p_{T-nh}(\vartheta_n^{\mathsf{SO}}) - L_n(x_{nh+s} - \vartheta_n^{\mathsf{SO}}) - M_n s \right) \mathrm{d}s$$

$$= \int_0^t L_n(\tilde{Y}_s - x_{nh+s}) \,\mathrm{d}s + \int_0^t \left( \frac{1}{2}\tilde{Y}_s + \nabla \log p_{T-nh-s}(\tilde{Y}_s) - \frac{1}{2}\vartheta_n^{\mathsf{SO}} - \nabla \log p_{T-nh}(\vartheta_n^{\mathsf{SO}}) \right) \mathrm{d}s$$

$$- \int_0^t \left( \int_0^s L_n \,\mathrm{d}\tilde{Y}_u \,\mathrm{d}u \right) \mathrm{d}s - \int_0^t \left( \int_0^s M_n \,\mathrm{d}u \right) \mathrm{d}s$$

$$= \int_0^t L_n(\tilde{Y}_s - x_{nh+s}) \,\mathrm{d}s + \int_0^t \left( \int_0^s \mathrm{d}\left( \frac{1}{2}\tilde{Y}_u + \nabla \log p_{T-nh-u}(\tilde{Y}_u) \right) \right) \mathrm{d}s$$

$$- \int_0^t \left( \int_0^s L_n \,\mathrm{d}\tilde{Y}_u \,\mathrm{d}u \right) \mathrm{d}s - \int_0^t \left( \int_0^s M_n \,\mathrm{d}u \right) \mathrm{d}s \,.$$

We then apply the Itô formula to the term $\mathrm{d}\left( \frac{1}{2}\tilde{Y}_u + \nabla \log p_{T-nh-u}(\tilde{Y}_u) \right)$. Recall the definitions of $L_n$ and $M_n$, and after rearranging the expression, we obtain

$$\tilde{Y}_t - x_{nh+t} = \underbrace{\int_0^t L_n(\tilde{Y}_s - x_{nh+s}) \,\mathrm{d}s}_{\mathrm{I}} + \underbrace{\int_0^t \left( \int_0^s \nabla^2 \log p_{T-nh-u}(\tilde{Y}_u) - \nabla^2 \log p_{T-nh}(\tilde{Y}_0) \,\mathrm{d}\tilde{Y}_u \right) \mathrm{d}s}_{\mathrm{II}}$$

$$+ \underbrace{\int_0^t \left[ \int_0^s \left( \frac{1}{2}\sum_{j=1}^d \frac{\partial^2}{\partial x_j^2}\nabla \log p_{T-nh-u}(\tilde{Y}_u) - \partial_t \nabla \log p_{T-nh-u}(\tilde{Y}_u) \right) \right.}_{\mathrm{III}}$$

$$\underbrace{\left. - \left( \frac{1}{2}\sum_{j=1}^d \frac{\partial^2}{\partial x_j^2}\nabla \log p_{T-nh}(\tilde{Y}_0) - \partial_t \nabla \log p_{T-nh}(\tilde{Y}_0) \right) \mathrm{d}u \right] \mathrm{d}s}_{\mathrm{III}} \,.$$

In what follows, we derive the upper bounds for each term on the right-hand side of the previous display.

**Upper bound for term** I: The upper bound of the term I follows directly from the fact that

$$\|L_n\|_{\mathbb{L}_2} \leqslant L(T - nh) - \frac{1}{2} \,.$$

**Upper bound for term** II: To derive the upper bound for the second term, we expand the term $\mathrm{d}\tilde{Y}_u$, yielding

$$\left\| \int_0^s \nabla^2 \log p_{T-nh-u}(\tilde{Y}_u) - \nabla^2 \log p_{T-nh}(\tilde{Y}_0) \,\mathrm{d}\tilde{Y}_u \right\|_{\mathbb{L}_2}$$

$$\leqslant \left\| \int_0^s \left( \nabla^2 \log p_{T-nh-u}(\tilde{Y}_u) - \nabla^2 \log p_{T-nh}(\tilde{Y}_0) \right) \left( \frac{1}{2}\tilde{Y}_u + \nabla \log p_{T-nh-u}(\tilde{Y}_u) \right) \mathrm{d}u \right\|_{\mathbb{L}_2}$$

$$+ \left\| \int_0^s \left( \nabla^2 \log p_{T-nh-u}(\tilde{Y}_u) - \nabla^2 \log p_{T-nh}(\tilde{Y}_0) \right) \mathrm{d}W_u \right\|_{\mathbb{L}_2}$$

$$\leqslant \int_0^s \left\| \nabla^2 \log p_{T-nh-u}(\tilde{Y}_u) - \nabla^2 \log p_{T-nh}(\tilde{Y}_0) \right\|_{\mathbb{L}_2} \cdot \left\| \frac{1}{2}\tilde{Y}_u + \nabla \log p_{T-nh-u}(\tilde{Y}_u) \right\|_{\mathbb{L}_2} \mathrm{d}u$$

$$+ \left( \int_0^s \left\| \nabla^2 \log p_{T-nh-u}(\tilde{Y}_u) - \nabla^2 \log p_{T-nh}(\tilde{Y}_0) \right\|_{\mathbb{L}_2}^2 \mathrm{d}u \right)^{1/2} \,.$$

The second inequality follows from display (28). We note that by Assumptions 6 and 7, it holds that

$$
\left\| \nabla^2 \log p_{T-nh-u}(\tilde{Y}_u) - \nabla^2 \log p_{T-nh}(\tilde{Y}_0) \right\|_{\mathbb{L}_2}
$$

$$
\leqslant \left\| \nabla^2 \log p_{T-nh-u}(\tilde{Y}_u) - \nabla^2 \log p_{T-nh}(\tilde{Y}_u) \right\|_{\mathbb{L}_2} + \left\| \nabla^2 \log p_{T-nh}(\tilde{Y}_u) - \nabla^2 \log p_{T-nh}(\tilde{Y}_0) \right\|_{\mathbb{L}_2}
$$

$$
\leqslant M_2 h (1 + \left\| \tilde{Y}_u \right\|_{\mathbb{L}_2}) + L_F \left\| \tilde{Y}_u - \tilde{Y}_0 \right\|_{\mathbb{L}_2}
$$

$$
\leqslant M_2 h + (L_F + M_2 h) \left\| \tilde{Y}_u - \tilde{Y}_0 \right\|_{\mathbb{L}_2} + M_2 h \left\| \vartheta_n^{\mathsf{SO}} \right\|_{\mathbb{L}_2}
$$

$$
\lesssim L_F \sqrt{du}\,,
$$

Combining this with the previous display provides us with

$$
\left\| \int_0^s \nabla^2 \log p_{T-nh-u}(\tilde{Y}_u) - \nabla^2 \log p_{T-nh}(\tilde{Y}_0)\, \mathrm{d}\tilde{Y}_u \right\|_{\mathbb{L}_2}
$$

$$
\lesssim \int_0^s L_F \sqrt{du} \cdot \left\| \frac{1}{2}\tilde{Y}_u + \nabla \log p_{T-nh-u}(\tilde{Y}_u) \right\|_{\mathbb{L}_2} \mathrm{d}u + \left( \int_0^s L_F^2 du\, \mathrm{d}u \right)^{1/2}.
$$

Hence, we obtain

$$
\| \mathrm{II} \|_{\mathbb{L}_2} = \left\| \int_0^t \left( \int_0^s \nabla^2 \log p_{T-nh-u}(\tilde{Y}_u) - \nabla^2 \log p_{T-nh}(\tilde{Y}_0)\, \mathrm{d}\tilde{Y}_u \right) \mathrm{d}s \right\|_{\mathbb{L}_2}
$$

$$
\leqslant \int_0^t \left\| \int_0^s \nabla^2 \log p_{T-nh-u}(\tilde{Y}_u) - \nabla^2 \log p_{T-nh}(\tilde{Y}_0)\, \mathrm{d}\tilde{Y}_u \right\|_{\mathbb{L}_2} \mathrm{d}s
$$

$$
\leqslant \int_0^t \left[ \int_0^s L_F \sqrt{du} \cdot \left\| \frac{1}{2}\tilde{Y}_u + \nabla \log p_{T-nh-u}(\tilde{Y}_u) \right\|_{\mathbb{L}_2} \mathrm{d}u + \left( \int_0^s L_F^2 du\, \mathrm{d}u \right)^{1/2} \right] \mathrm{d}s
$$

$$
\lesssim \frac{\sqrt{2}}{4} L_F \sqrt{d} t^2 \,.
$$

(38)

**Upper bound for term** III:

In Section 4, it is claimed that the partial derivative of $\nabla \log p_t$ with respect to $t$ can be estimated without requiring additional assumptions. This is achieved by transforming the $t$-derivative into $x$-derivative via the Fokker-Planck equation, as detailed below.

$$
\partial_t p_t(x) = \frac{d}{2} p_t(x) + \frac{1}{2} x^\top \nabla p_t(x) + \frac{1}{2} \sum_{i=1}^d \frac{\partial^2 p_t(x)}{\partial x_i^2}\,. \tag{39}
$$

We need the following auxiliary lemma.

**Lemma 17.** *Let $p_t$ be the probability density function of $X_t$, then*

$$
\sum_{i=1}^d \frac{\partial^2 p_t(x)}{\partial x_i^2} \cdot \frac{1}{p_t(x)} = \mathrm{Tr}\left( \nabla^2 \log p_t(x) \right) + \| \nabla \log p_t(x) \|^2\,,
$$

$$
\nabla \left( \sum_{i=1}^d \frac{\partial^2 p_t(x)}{\partial x_i^2} \right) \cdot \frac{1}{p_t(x)} = \nabla \left( \mathrm{Tr}(\nabla^2 \log p_t(x)) \right) + \nabla(\| \nabla \log p_t(x) \|^2)
$$

$$
+ \left[ \mathrm{Tr}(\nabla^2 \log p_t(x)) + \| \nabla \log p_t(x) \|^2 \right] \cdot \nabla \log p_t(x)\,.
$$

We begin by taking the gradient of $\log p_t$, and then compute the partial derivative of $\nabla \log p_t$ with respect to $t$. This results in

$$
\partial_t \nabla \log p_t(x) = \partial_t \left( \frac{\nabla p_t(x)}{p_t(x)} \right) = \frac{\partial_t \nabla p_t(x)}{p_t(x)} - \frac{\nabla p_t(x)}{p_t(x)} \cdot \frac{\partial_t p_t(x)}{p_t(x)}\,.
$$

Under certain regularity conditions, we can interchange the operators $\partial_t$ and $\nabla$ in the term $\partial_t \nabla p_t(x)$, and substitute $\partial_t p_t(x)$ by (39), it follows that

$$\partial_t \nabla \log p_t(x) = \frac{\nabla \partial_t p_t(x)}{p_t(x)} - \nabla \log p_t(x) \cdot \left( \frac{d}{2} + \frac{1}{2} x^\top \nabla \log p_t(x) + \frac{1}{2} \sum_{i=1}^d \frac{\partial^2 p_t(x)}{\partial x_i^2} \cdot \frac{1}{p_t(x)} \right).$$

and

$$\frac{\nabla \partial_t p_t(x)}{p_t(x)} = \frac{1}{p_t(x)} \cdot \nabla \left( \frac{d}{2} p_t(x) + \frac{1}{2} x^\top \nabla p_t(x) + \frac{1}{2} \sum_{i=1}^d \frac{\partial^2 p_t(x)}{\partial x_i^2} \right)$$

$$= \frac{d}{2} \frac{\nabla p_t(x)}{p_t(x)} + \frac{1}{2} \frac{\nabla p_t(x)}{p_t(x)} + \frac{1}{2} \frac{\nabla^2 p_t(x) x}{p_t(x)} + \frac{1}{2} \nabla \left( \sum_{i=1}^d \frac{\partial^2 p_t(x)}{\partial x_i^2} \right) \cdot \frac{1}{p_t(x)}$$

$$= \frac{d+1}{2} \nabla \log p_t(x) + \frac{1}{2} \frac{\nabla^2 p_t(x) x}{p_t(x)} + \frac{1}{2} \nabla \left( \sum_{i=1}^d \frac{\partial^2 p_t(x)}{\partial x_i^2} \right) \cdot \frac{1}{p_t(x)}.$$

Therefore, we obtain that

$$\partial_t \nabla \log p_t(x) = \frac{d+1}{2} \nabla \log p_t(x) + \frac{1}{2} \frac{\nabla^2 p_t(x) x}{p_t(x)} + \frac{1}{2} \nabla \left( \sum_{i=1}^d \frac{\partial^2 p_t(x)}{\partial x_i^2} \right) \cdot \frac{1}{p_t(x)}$$

$$- \nabla \log p_t(x) \cdot \left( \frac{d}{2} + \frac{1}{2} x^\top \nabla \log p_t(x) + \frac{1}{2} \sum_{i=1}^d \frac{\partial^2 p_t(x)}{\partial x_i^2} \cdot \frac{1}{p_t(x)} \right)$$

$$= \frac{1}{2} \nabla \log p_t(x) + \frac{1}{2} \left( \frac{\nabla^2 p_t(x) x}{p_t(x)} - \nabla \log p_t(x) \nabla \log p_t(x)^\top x \right)$$

$$+ \frac{1}{2} \nabla \left( \sum_{i=1}^d \frac{\partial^2 p_t(x)}{\partial x_i^2} \right) \cdot \frac{1}{p_t(x)} - \frac{1}{2} \nabla \log p_t(x) \sum_{i=1}^d \frac{\partial^2 p_t(x)}{\partial x_i^2} \cdot \frac{1}{p_t(x)}.$$

By Lemma 17, the last two terms above can be calculated. Additionally, it holds that

$$\nabla^2 \log p_t(x) = \frac{\nabla^2 p_t(x)}{p_t(x)} - \frac{\nabla p_t(x) \nabla p_t(x)^\top}{p_t(x)^2}.$$

Thus, $\partial_t \nabla \log p_t(x)$ can be simplified to

$$\partial_t \nabla \log p_t(x) = \frac{1}{2} \nabla \log p_t(x) + \frac{1}{2} \nabla^2 \log p_t(x) x$$

$$+ \frac{1}{2} \left[ \nabla \left( \mathrm{Tr}(\nabla^2 \log p_t(x)) \right) + \nabla(\|\nabla \log p_t(x)\|^2) \right]$$

$$+ \frac{1}{2} \left( \mathrm{Tr}(\nabla^2 \log p_t(x)) + \|\nabla \log p_t(x)\|^2 \right) \cdot \nabla \log p_t(x)$$

$$- \frac{1}{2} \nabla \log p_t(x) \left( \mathrm{Tr}(\nabla^2 \log p_t(x)) + \|\nabla \log p_t(x)\|^2 \right)$$

$$= \frac{1}{2} \nabla \log p_t(x) + \frac{1}{2} \nabla^2 \log p_t(x) x + \frac{1}{2} \nabla(\mathrm{Tr}(\nabla^2 \log p_t(x))) + \frac{1}{2} \nabla(\|\nabla \log p_t(x)\|^2).$$

Notice that

$$\sum_{j=1}^d \frac{\partial^2}{\partial x_j^2} \nabla \log p_t(x) = \frac{1}{2} \nabla(\mathrm{Tr}(\nabla^2 \log p_t(x))).$$

Then, it follows that

$$\frac{1}{2} \sum_{j=1}^d \frac{\partial^2}{\partial x_j^2} \nabla \log p_t(x) - \partial_t \nabla \log p_t(x)$$

$$= - \left( \frac{1}{2} \nabla \log p_t(x) + \frac{1}{2} \nabla^2 \log p_t(x) x + \frac{1}{2} \nabla(\|\nabla \log p_t(x)\|^2) \right)$$

$$= - \frac{1}{2} \left( \nabla \log p_t(x) + \nabla^2 \log p_t(x) x \right) + \nabla^2 \log p_t(x) \cdot \nabla \log p_t(x).$$

Therefore we have

$$
\left\| \frac{1}{2} \sum_{j=1}^{d} \frac{\partial^2}{\partial x_j^2} \nabla \log p_{T-nh-u}(\tilde{Y}_u) - \partial_t \nabla \log p_{T-nh-u}(\tilde{Y}_u) \right\|_{\mathbb{L}_2}
$$

$$
\leqslant \frac{1}{2} \left\| \nabla \log p_{T-nh-u}(\tilde{Y}_u) \right\|_{\mathbb{L}_2} + \frac{1}{2} \left\| \nabla^2 \log p_{T-nh-u}(\tilde{Y}_u) \right\|_{\mathbb{L}_2} \left\| \tilde{Y}_u \right\|_{\mathbb{L}_2}
$$

$$
+ \left\| \nabla^2 \log p_{T-nh-u}(\tilde{Y}_u) \right\|_{\mathbb{L}_2} \left\| \nabla \log p_{T-nh-u}(\tilde{Y}_u) \right\|_{\mathbb{L}_2}
$$

$$
\leqslant \frac{1}{2} \left\| \nabla \log p_{T-nh-u}(\tilde{Y}_u) \right\|_{\mathbb{L}_2} + \frac{1}{2} L(T-nh-u) \left\| \tilde{Y}_u \right\|_{\mathbb{L}_2} + L(T-nh-u) \left\| \nabla \log p_{T-nh-u}(\tilde{Y}_u) \right\|_{\mathbb{L}_2}.
$$

The second inequality follows from Assumption 1. The bounds for $\left\| \tilde{Y}_u \right\|_{\mathbb{L}_2}$ and $\left\| \nabla \log p_{T-nh-u}(\tilde{Y}_u) \right\|_{\mathbb{L}_2}$ can be derived according to the proof of Lemma 10. We then find

$$
\left\| \frac{1}{2} \sum_{j=1}^{d} \frac{\partial^2}{\partial x_j^2} \nabla \log p_{T-nh-u}(\tilde{Y}_u) - \partial_t \nabla \log p_{T-nh-u}(\tilde{Y}_u) \right\|_{\mathbb{L}_2}
$$

$$
\leqslant \frac{1}{2} \left\| \nabla \log p_{T-nh-u}(\tilde{Y}_u) \right\|_{\mathbb{L}_2} + \frac{1}{2} L(T-nh-u) \left\| \tilde{Y}_u \right\|_{\mathbb{L}_2} + L(T-nh-u) \left\| \nabla \log p_{T-nh-u}(\tilde{Y}_u) \right\|_{\mathbb{L}_2}
$$

$$
\leqslant \left( \frac{1}{2} + L(T-nh-u) \right) \left[ L(T-nh-u)(\left\| Y_{nh} - \vartheta_n^{\mathsf{SO}} \right\|_{\mathbb{L}_2} + C_2(n)) + \left( dL(T-nh-u) \right)^{1/2} \right]
$$

$$
+ \frac{1}{2} L(T-nh-u) \left( \left\| Y_{nh} - \vartheta_n^{\mathsf{SO}} \right\|_{\mathbb{L}_2} + C_2(n) + C_4 \right).
$$

Therefore, we obtain

$$
\left\| \mathrm{III} \right\|_{\mathbb{L}_2} \leqslant \int_0^t \int_0^s \left\| \frac{1}{2} \sum_{j=1}^{d} \frac{\partial^2}{\partial x_j^2} \nabla \log p_{T-nh-u}(\tilde{Y}_u) - \partial_t \nabla \log p_{T-nh-u}(\tilde{Y}_u) \right\|_{\mathbb{L}_2} \mathrm{d}u \, \mathrm{d}s
$$

$$
+ \int_0^t \int_0^s \left\| \frac{1}{2} \sum_{j=1}^{d} \frac{\partial^2}{\partial x_j^2} \nabla \log p_{T-nh}(\tilde{Y}_0) - \partial_t \nabla \log p_{T-nh}(\tilde{Y}_0) \right\|_{\mathbb{L}_2} \mathrm{d}u \, \mathrm{d}s
$$

$$
\leqslant \int_0^t \int_0^s \left[ (1 + L(T-nh-u))L(T-nh-u) + (1 + L(T-nh))L(T-nh) \right] \mathrm{d}u \, \mathrm{d}s \cdot \left\| Y_{nh} - \vartheta_n^{\mathsf{SO}} \right\|_{\mathbb{L}_2}
$$

$$
+ \int_0^t \int_0^s \left[ (1 + L(T-nh-u))L(T-nh-u) + (1 + L(T-nh))L(T-nh) \right] \mathrm{d}u \, \mathrm{d}s \cdot C_2(n)
$$

$$
+ \sqrt{d} \int_0^t \int_0^s \left[ (\frac{1}{2} + L(T-nh-u))L(T-nh-u)^{1/2} + (\frac{1}{2} + L(T-nh))L(T-nh)^{1/2} \right] \mathrm{d}u \, \mathrm{d}s
$$

$$
+ \int_0^t \int_0^s \frac{1}{2}(L(T-nh-u) + L(T-nh)) \mathrm{d}u \, \mathrm{d}s \cdot C_4.
$$

$$
(40)
$$

For simplicity, we focus on the lowest-order term. Recall equations (36), (38) and (40), which lead to the following expression

$$
\left\| \tilde{Y}_t - x_{nh+t} \right\|_{\mathbb{L}_2} \leqslant \left( L(T-nh) - \frac{1}{2} \right) \int_0^t \left\| \tilde{Y}_s - x_{nh+s} \right\|_{\mathbb{L}_2} \mathrm{d}s + \left( A_{n,1} \left\| Y_{nh} - \vartheta_n^{\mathsf{SO}} \right\|_{\mathbb{L}_2} + A_{n,2} \right) \cdot t^2,
$$

where

$$
A_{n,1} = \sup_{nh \leqslant t \leqslant (n+1)h} \frac{1}{t^2} \int_0^t \int_0^s \left[ (1 + L(T-nh-u))L(T-nh-u) + (1 + L(T-nh))L(T-nh) \right] \mathrm{d}u \, \mathrm{d}s,
$$

and

$$A_{n,2} = \sup_{nh \leqslant t \leqslant (n+1)h} \frac{1}{t^2} \left[ C_2(n) \int_0^t \int_0^s \left[ (1 + L(T - nh - u))L(T - nh - u) + (1 + L(T - nh))L(T - nh) \right] du\, ds \right.$$

$$+ \sqrt{d} \int_0^t \int_0^s \left[ (\frac{1}{2} + L(T - nh - u))L(T - nh - u)^{1/2} + (\frac{1}{2} + L(T - nh))L(T - nh)^{1/2} \right] du\, ds$$

$$+ \left. \int_0^t \int_0^s \frac{1}{2}(L(T - nh - u) + L(T - nh))\, du\, ds \cdot C_4 \right]$$

$$+ \frac{\sqrt{2}}{4}\sqrt{d}L_F.$$

Using Lemma 16 with

$$z(t) = \left\| \tilde{Y}_t - x_{nh+t} \right\|_{\mathbb{L}_2},$$

$$\alpha(t) = \left( A_{n,1} \left\| Y_{nh} - \vartheta_n^{\mathsf{SO}} \right\|_{\mathbb{L}_2} + A_{n,2} \right) \cdot t^2,$$

$$\beta(t) = L(T - nh) - \frac{1}{2},$$

set $t_0 = nh$, we then obtain

$$\left\| \tilde{Y}_h - x_{(n+1)h} \right\|_{\mathbb{L}_2} \leqslant \left( A_{n,1} \left\| Y_{nh} - \vartheta_n^{\mathsf{SO}} \right\|_{\mathbb{L}_2} + A_{n,2} \right) h^2 \exp\left( (L(T - nh) - \frac{1}{2})h \right)$$

$$= A_{n,1} e^{\left( L(T-nh) - \frac{1}{2} \right)h} h^2 \left\| Y_{nh} - \vartheta_n^{\mathsf{SO}} \right\|_{\mathbb{L}_2} + A_{n,2} e^{(L(T-nh) - \frac{1}{2})h} h^2.$$

Invoking display (37), we arrive at

$$\left\| \tilde{Y}_h - \vartheta_{n+1}^{\mathsf{SO}} \right\|_{\mathbb{L}_2} \leqslant \left\| \tilde{Y}_h - x_{(n+1)h} \right\|_{\mathbb{L}_2} + \left\| \vartheta_{n+1}^{\mathsf{SO}} - x_{(n+1)h} \right\|_{\mathbb{L}_2}$$

$$\leqslant A_{n,1} e^{(L(T-nh) - \frac{1}{2})h} h^2 \left\| Y_{nh} - \vartheta_n^{\mathsf{SO}} \right\|_{\mathbb{L}_2} + A_{n,2} e^{(L(T-nh) - \frac{1}{2})h} h^2$$

$$+ \left[ h\varepsilon_{sc} + \frac{2}{3}h^{3/2}\sqrt{d}\varepsilon_{sc}^{(L)} + \frac{1}{2}h^2\varepsilon_{sc}^{(M)} \right] e^{(L(T-nh) - \frac{1}{2})h}.$$

Furthermore, we can bound the coefficients $A_{n,1}$ and $A_{n,2}$ as follows,

$$A_{n,1} \leqslant \frac{1}{t^2} \int_0^t \left( \int_0^s 2(1 + L_{\max})L_{\max}\, du \right) ds = (1 + L_{\max})L_{\max},$$

$$A_{n,2} \leqslant (1 + L_{\max})L_{\max}C_2(n) + \sqrt{d}(\frac{1}{2} + L_{\max})L_{\max}^{1/2} + \frac{1}{2}L_{\max}C_4 + \frac{\sqrt{2}}{4}\sqrt{d}L_F$$

$$\lesssim \sqrt{d}(\frac{1}{2} + L_{\max})L_{\max}^{1/2} + \frac{\sqrt{2}}{4}\sqrt{d}L_F.$$

Collecting all the pieces then gives

$$\left\| Y_{nh} - \vartheta_{n+1}^{\mathsf{SO}} \right\|_{\mathbb{L}_2} \leqslant r_n^{\mathsf{SO}} \left\| Y_{nh} - \vartheta_n^{\mathsf{SO}} \right\|_{\mathbb{L}_2} + C_n^{\mathsf{SO}} h^2 + \left[ h\varepsilon_{sc} + \frac{2}{3}h^{3/2}\varepsilon_{sc}^{(L)} + \frac{1}{2}h^2\varepsilon_{sc}^{(M)} \right] e^{(L(T-nh) - \frac{1}{2})h},$$

where

$$r_n^{\mathsf{SO}} = e^{-\int_{nh}^{(n+1)h}(m(T-t) - \frac{1}{2})\, dt} + A_{n,1}e^{(L(T-nh) - \frac{1}{2})h}h^2,$$

$$C_n^{\mathsf{SO}} = A_{n,2}e^{(L(T-nh) - \frac{1}{2})h}.$$

$\square$

From the result above, we finally obtain

$$\left\| Y_{Nh} - \vartheta_N^{\mathsf{SO}} \right\|_{\mathbb{L}_2} \lesssim \frac{1}{m_{\min} - 1/2} \left[ h \max_{0 \leqslant k \leqslant N-1} C_k^{\mathsf{SO}} + \left( \varepsilon_{sc} + \frac{2}{3}h^{1/2}\varepsilon_{sc}^{(L)} + \frac{1}{2}h\varepsilon_{sc}^{(M)} \right) e^{(L_{\max} - \frac{1}{2})h} \right]$$

$$\lesssim h \cdot \frac{\sqrt{d}(L_{\max}^{3/2} + \sqrt{2}L_F/4)e^{(L_{\max} - \frac{1}{2})h}}{m_{\min} - 1/2} + \left( \varepsilon_{sc} + \frac{2}{3}\sqrt{hd}\varepsilon_{sc}^{(L)} + \frac{1}{2}h\varepsilon_{sc}^{(M)} \right) e^{(L_{\max} - \frac{1}{2})h}.$$

This completes the proof of Theorem 4.

# D Proof of Auxiliary Lemma

## D.1 Proof of Lemma 8

We have

$$
\begin{aligned}
\frac{\mathrm{d}\,\|H_t - G_t\|^2}{\mathrm{d}t} &= 2\left\langle H_t - G_t, \frac{\mathrm{d}(H_t - G_t)}{\mathrm{d}t}\right\rangle \\
&= 2\left\langle H_t - G_t, \frac{1}{2}(H_t - G_t) + \left(\nabla \log p_{T-t}(H_t) - \nabla \log p_{T-t}(G_t)\right)\right\rangle \quad (41) \\
&= \|H_t - G_t\|^2 + 2\langle H_t - G_t, \nabla \log p_{T-t}(H_t) - \nabla \log p_{T-t}(G_t)\rangle \\
&\leqslant \left(1 - 2m(T-t)\right)\|H_t - G_t\|^2.
\end{aligned}
$$

The last inequality follows from Lemma 6. Then, we take the derivative of $e^{-\int_{t_1}^t (2m(T-s)-1)\,\mathrm{d}s}\|H_t - G_t\|^2$

$$
\begin{aligned}
&\frac{\mathrm{d}}{\mathrm{d}t}\left(e^{\int_{t_1}^t (2m(T-s)-1)\,\mathrm{d}s}\|H_t - G_t\|^2\right) \\
&= (2m(T-t)-1)e^{-\int_{t_1}^t (2m(T-s)-1)\,\mathrm{d}s}\|H_t - G_t\|^2 + e^{-\int_{t_1}^t (2m(T-s)-1)\,\mathrm{d}s}\frac{\mathrm{d}\,\|H_t - G_t\|^2}{\mathrm{d}t} \\
&\leqslant 0.
\end{aligned}
$$

Therefore, we obtain that

$$
e^{\int_{t_1}^t (2m(T-s)-1)\,\mathrm{d}s}\|H_t - G_t\|^2 \leqslant \|H_{t_t} - G_{t_1}\|^2.
$$

taking the expectation of both sides and then applying the square root yields the desired result.

## D.2 Proof of Lemma 10

By the definition of $\tilde{Y}_h$, we have

$$
\begin{aligned}
\left\|\tilde{Y}_t - \tilde{Y}_0\right\|_{\mathbb{L}_2} &= \left\|\int_0^t \left(\frac{1}{2}\tilde{Y}_s + \nabla \log p_{T-nh-s}(\tilde{Y}_s)\right)\mathrm{d}s + \int_{nh}^{nh+t}\mathrm{d}W_s\right\|_{\mathbb{L}_2} \\
&\leqslant \int_0^t \frac{1}{2}\left\|\tilde{Y}_s\right\|_{\mathbb{L}_2}\mathrm{d}t + \int_0^t \left\|\nabla \log p_{T-nh-s}(\tilde{Y}_s)\right\|_{\mathbb{L}_2}\mathrm{d}s + \left\|\int_{nh}^{nh+t}\mathrm{d}W_s\right\|_{\mathbb{L}_2}.
\end{aligned}
$$

To bound the first term, we observe that for any $s \in [0, h]$, the following holds

$$
\begin{aligned}
\left\|\tilde{Y}_s\right\|_{\mathbb{L}_2} &\leqslant \|Y_{nh+s}\|_{\mathbb{L}_2} + \left\|\tilde{Y}_s - Y_{nh+s}\right\|_{\mathbb{L}_2} \\
&\leqslant \left\|Y_{nh+s} - X_{nh+s}^{\leftarrow}\right\|_{\mathbb{L}_2} + \left\|X_{nh+s}^{\leftarrow}\right\|_{\mathbb{L}_2} + \left\|\tilde{Y}_s - Y_{nh+s}\right\|_{\mathbb{L}_2} \\
&\leqslant e^{-\int_0^{nh+s}(m(T-t)-\frac{1}{2})\,\mathrm{d}t}\left\|Y_0 - X_0^{\leftarrow}\right\|_{\mathbb{L}_2} + \left\|X_{T-(nh+s)}\right\|_{\mathbb{L}_2} + e^{-\int_{nh}^s (m(T-u)-\frac{1}{2})\,\mathrm{d}u}\left\|Y_{nh} - \vartheta_n^{\mathsf{EM}}\right\|_{\mathbb{L}_2} \\
&\leqslant e^{-\int_0^{nh}(m(T-t)-\frac{1}{2})\,\mathrm{d}t}\left\|Y_0 - X_T\right\|_{\mathbb{L}_2} + \sup_{0\leqslant t\leqslant T}\|X_t\|_{\mathbb{L}_2} + \left\|Y_{nh} - \vartheta_n^{\mathsf{EM}}\right\|_{\mathbb{L}_2}.
\end{aligned}
$$

Here, the second inequality follows from the Grönwall inequality applied on $\left\|Y_{nh+s} - X_{nh+s}^{\leftarrow}\right\|_{\mathbb{L}_2}$ and $\left\|\tilde{Y}_s - Y_{nh+s}\right\|_{\mathbb{L}_2}$, and the fact that $\|X_t\|_{\mathbb{L}_2} = \left\|X_{T-t}^{\leftarrow}\right\|_{\mathbb{L}_2}$. To bound the second term, we need the following lemma.

**Lemma 18.** *If the target distribution $p_0$ satisfies Assumption 1, it holds that*

$$
\|\nabla \log p_t(X_t)\|_{\mathbb{L}_2} \leqslant (dL(t))^{1/2}.
$$

According to Lemma 18, it follows that

$$\left\| \nabla \log p_{T-nh-s}(\tilde{Y}_s) \right\|_{\mathbb{L}_2}$$

$$\leqslant \left\| \nabla \log p_{T-nh-s}(\tilde{Y}_s) - \nabla \log p_{T-nh-s}(X_{nh+s}^{\leftarrow}) \right\|_{\mathbb{L}_2} + \left\| \nabla \log p_{T-nh-s}(X_{nh+s}^{\leftarrow}) \right\|_{\mathbb{L}_2}$$

$$\leqslant L(T-nh-s) \left\| \tilde{Y}_s - X_{nh+s}^{\leftarrow} \right\|_{\mathbb{L}_2} + (dL(T-nh-s))^{1/2}$$

$$\leqslant L(T-nh-s) \left\| \tilde{Y}_0 - X_{nh}^{\leftarrow} \right\|_{\mathbb{L}_2} + (dL(T-nh-s))^{1/2}$$

$$\leqslant L(T-nh-s) \left( \left\| \tilde{Y}_0 - Y_{nh} \right\|_{\mathbb{L}_2} + \left\| Y_{nh} - X_{nh}^{\leftarrow} \right\|_{\mathbb{L}_2} \right) + (dL(T-nh-s))^{1/2}$$

$$\leqslant L(T-nh-s) \left( \left\| Y_{nh} - \vartheta_n^{\mathsf{EM}} \right\|_{\mathbb{L}_2} + e^{-\int_0^{nh}(m(T-t)-\frac{1}{2})\,dt} \left\| Y_0 - X_T \right\|_{\mathbb{L}_2} \right) + (dL(T-nh-s))^{1/2}.$$

Here, we use the fact that $\tilde{Y}_0 = \vartheta_n^{\mathsf{EM}}$, and Grönwall inequality are used in the third inequality and the last one. This completes the proof.

### D.3 Proof of Lemma 13

For the stochastic integral of process $X$, we have

$$\mathbb{E}(I_t(X))^2 = \mathbb{E} \int_0^t X_u^2 d\langle M \rangle_u \,.$$

Then, we obtain

$$\left\| \int_{nh}^{(n+U_n)h} dW_t - \int_0^1 \left( \int_{nh}^{(n+u)h} dW_t \right) du \right\|_{\mathbb{L}_2}^2$$

$$= \mathbb{E} \left( \int_0^1 \int_{nh}^{(n+1)h} -\mathbf{1}_{\{U_n \leqslant u\}} \mathbf{1}_{\{(n+U_n)h \leqslant t \leqslant (n+u)h\}} + \mathbf{1}_{\{U_n > u\}} \mathbf{1}_{\{(n+u)h \leqslant t \leqslant (n+U_n)h\}} \, dW_t \, du \right)^2$$

$$\leqslant \int_0^1 \mathbb{E} \left( \int_{nh}^{(n+1)h} -\mathbf{1}_{\{U_n \leqslant u\}} \mathbf{1}_{\{(n+U_n)h \leqslant t \leqslant (n+u)h\}} + \mathbf{1}_{\{U_n > u\}} \mathbf{1}_{\{(n+u)h \leqslant t \leqslant (n+U_n)h\}} \, dW_t \right)^2 du$$

$$= \int_0^1 \left( \mathbb{E} \int_{nh}^{(n+1)h} \mathbf{1}_{\{U_n \leqslant u\}} \mathbf{1}_{\{(n+U_n)h \leqslant t \leqslant (n+u)h\}} + \mathbf{1}_{\{U_n > u\}} \mathbf{1}_{\{(n+u)h \leqslant t \leqslant (n+U_n)h\}} \, dt \right) du$$

$$= \int_0^1 \left( \mathbb{E} \left( \mathbf{1}_{\{U_n \leqslant u\}} (u - U_n)h + \mathbf{1}_{\{U_n > u\}} (U_n - u)h \right) \right) du$$

$$= h \int_0^1 \left( u^2 - u + \frac{1}{2} \right) du$$

$$= \frac{1}{3} h \,.$$

### D.4 Proof of Lemma 16

Define the function $w(s)$ via

$$w(s) = \exp\left( -\int_{t_0}^s \beta(r)dr \right) \int_{t_0}^s \beta(r)z(r) \, dr \,, \quad \forall s \geqslant t_0 \,.$$

Differentiating this function gives

$$w'(s) = \left( z(s) - \int_{t_0}^s \beta(r)z(r) \, dr \right) \beta(s) \exp\left( -\int_{t_0}^s \beta(r) \, dr \right) \leqslant \alpha(s)\beta(s) \exp\left( -\int_{t_0}^s \beta(r) \, dr \right).$$

Note that $w(t_0) = 0$. Integrating the function $w$ from $t_0$ to $t$ yields

$$w(t) \leqslant \int_{t_0}^{t} \alpha(s)\beta(s) \exp\left(-\int_{t_0}^{s} \beta(r)\,\mathrm{d}r\right)\,\mathrm{d}s\,.$$

By the definition of $w(s)$, we also have

$$\int_{t_0}^{t} \beta(s)z(s)\,\mathrm{d}s = \exp\left(\int_{t_0}^{t} \beta(r)\,\mathrm{d}r\right) w(t)\,.$$

Combining the previous two displays provides us with

$$\int_{t_0}^{t} \beta(s)z(s)\,\mathrm{d}s \leqslant \int_{t_0}^{t} \alpha(s)\beta(s) \exp\left(\int_{s}^{t} \beta(r)\,\mathrm{d}r\right)\,\mathrm{d}s\,.$$

By substituting this estimate into the inequality, we can obtain the first desired result. Furthermore, if $\alpha$ is non-decreasing, then for any $s \leqslant t$, it holds that $\alpha(s) \leqslant \alpha(t)$. This leads to

$$z(t) \leqslant \alpha(t) + \alpha(t) \int_{t_0}^{t} \beta(s) \exp\left(\int_{s}^{t} \beta(r)\,\mathrm{d}r\right)\,\mathrm{d}s\,.$$

which can be simplified to

$$z(t) \leqslant \alpha(t) \exp\left(\int_{t_0}^{t} \beta(r)\,\mathrm{d}r\right), \quad t \geqslant t_0\,.$$

This completes the proof.

### D.5  Proof of Lemma 17

Notice that

$$\nabla^2 \log p_t(x) = -\frac{1}{p_t(x)^2}\nabla p_t(x)\nabla p_t(x)^\top + \frac{1}{p_t(x)}\nabla^2 p_t(x)$$

$$= -\nabla \log p_t(x)\nabla \log p_t(x)^\top + \frac{1}{p_t(x)}\nabla^2 p_t(x)\,,$$

which indicates

$$\frac{1}{2}\sum_{i=1}^{d} \frac{\partial^2 p_t(x)}{\partial x_i^2} \cdot \frac{1}{p_t(x)} = \frac{1}{2}\operatorname{Tr}\left(\frac{1}{p_t(x)}\nabla^2 p_t(x)\right)$$

$$= \frac{1}{2}\operatorname{Tr}\left(\nabla^2 \log p_t(x) + \nabla \log p_t(x)\nabla \log p_t(x)^\top\right)$$

$$= \frac{1}{2}\operatorname{Tr}\left(\nabla^2 \log p_t(x)\right) + \frac{1}{2}\|\nabla \log p_t(x)\|^2\,,$$

Additionally, we have

$$\nabla\left(\frac{\partial^2 \log p_t(x)}{\partial x_i^2}\right) = \nabla\left(\frac{\partial^2 p_t(x)}{\partial x_i^2} \cdot \frac{1}{p_t(x)} - \left(\frac{\partial p_t(x)}{\partial x_i} \cdot \frac{1}{p_t(x)}\right)^2\right)$$

$$= \nabla\left(\frac{\partial^2 p_t(x)}{\partial x_i^2}\right) \cdot \frac{1}{p_t(x)} - \frac{\partial^2 p_t(x)}{\partial x_i^2} \cdot \frac{1}{p_t(x)} \cdot \nabla \log p_t(x) - \nabla\left(\left(\frac{\partial \log p_t(x)}{\partial x_i}\right)^2\right)\,.$$

Then, we obtain

$$\nabla\left(\sum_{i=1}^{d} \frac{\partial^2 p_t(x)}{\partial x_i^2}\right) \cdot \frac{1}{p_t(x)}$$

$$= \nabla\left(\operatorname{Tr}(\nabla^2 \log p_t(x))\right) + \left[\operatorname{Tr}(\nabla^2 \log p_t(x)) + \|\nabla \log p_t(x)\|^2\right] \cdot \nabla \log p_t(x) + \nabla(\|\nabla \log p_t(x)\|^2)\,.$$

### D.6 Proof of Lemma 18

Note that

$$\mathbb{E}(\|\nabla \log p_t(X_t)\|^2) = \int_{\mathbb{R}^d} \|\nabla \log p_t(x)\|^2 \, p_t(x) \, \mathrm{d}x$$

$$= \lim_{R \to \infty} \int_{B(0,R)} \langle \nabla \log p_t(x), \nabla \log p_t(x) \rangle p_t(x) \, \mathrm{d}x$$

$$= \lim_{R \to \infty} \int_{B(0,R)} \langle \nabla \log p_t(x), \nabla p_t(x) \rangle \, \mathrm{d}x \, ,$$

where $B(0, R)$ denotes the Euclidean ball with radius $R > 0$ centered at the origin. Using integration by parts, we then obtain

$$\mathbb{E}(\|\nabla \log p_t(X_t)\|^2) = \lim_{R \to \infty} \int_{B(0,R)} -p_t(x)\Delta \log p_t(x) \, \mathrm{d}x + \int_{\partial B(0,R)} p_t(x) \frac{\partial \log p_t(x)}{\partial \vec{n}} \, \mathrm{d}S$$

$$= \int_{\mathbb{R}^d} p_t(x) \cdot (-\Delta \log p_t(x)) \, \mathrm{d}x$$

$$\leqslant dL(t) \, ,$$

where $\frac{\partial f}{\partial \vec{n}} = \nabla f \cdot \vec{n}$ represents the directional derivative along the normal vector $\vec{n}$ and $\mathrm{d}S$ denotes the surface integral over the spherical surface. Here we use the fact that $p_t(x)$ converges to 0 at an exponential rate as $\|x\|$ approaches infinity, and the fact that

$$-\Delta \log p_t(x) = -\operatorname{Tr}(\nabla^2 \log p_t(x)) \in [0, dL(t)] \, ,$$

which follows from Lemma 6.

## E  Details for second-order acceleration

In this section, we present a complete derivation of the second-order acceleration scheme, detailing the implementation of Itô-Taylor expansions and Itô's formula. Building upon the general backward process framework

$$dx_t = \gamma(T - t, x_t) \, \mathrm{d}t + \sigma \, \mathrm{d}W_t \, , \tag{42}$$

where $\sigma > 0$ and $W_t$ is the $d$-dimensional Brownian motion. We apply Itô's formula to $\gamma(T - t, x)$. This procedure generates an approximated structure of SDE (42),

$$dx_t = [\gamma(T - s, x_s) + L_s(x_t - x_s) + M_s(t - s)] \, \mathrm{d}t + \sigma \, \mathrm{d}W_t \, . \tag{43}$$

with

$$L_s = \frac{\partial \gamma}{\partial x}(T - s, x_s) \quad \text{and} \quad M_s = \frac{\sigma^2}{2} \frac{\partial^2 \gamma}{\partial x^2}(T - s, x_s) - \frac{\partial \gamma}{\partial t}(T - s, x_s)$$

which serves as the foundation for our subsequent second-order discretization. In Section 4, we demonstrate that this approximation preserves the core dynamical structure of the original SDE (42) while admitting a closed-form solution. This is achieved by replacing the intractable drift term $\gamma(T - t, x_s)$ with its Itô-expanded counterpart, which remains analytical tractable through explicit integration.

Applying Itô's formula to $e^{-L_s t} x_t$ yields

$$\mathrm{d}(e^{-L_s t} x_t) = e^{-L_s t}(\gamma(T - s, x_s) - L_s x_s + M_s(t - s)) \, \mathrm{d}t + e^{-L_s t} \sigma \, \mathrm{d}W_t \, .$$

For fixed $s$, both sides of the equation permit closed-form integration. While the Brownian integral $\int_s^{s+\Delta t} e^{-L_s t} \sigma \, \mathrm{d}W_t$ formally appears non-analytic, it equivalently manifests as a Gaussian random variable with explicitly computable variance. This enables full analytical representation for $x_{s+\Delta t}$

when integrating over $[s, s + \Delta t]$,

$$x_{s+\Delta t} = e^{L_s \Delta t} x_s + \int_s^{s+\Delta t} e^{L_s(s+\Delta t - t)} \, dt (\gamma(T-s, x_s) - L_s x_s)$$

$$+ \int_s^{s+\Delta t} e^{L_s(s+\Delta t - t)}(t-s) \, dt \, M_s + \sigma \int_s^{s+\Delta t} e^{L_s(s+\Delta t - t)} \, dW_t$$

$$= x_s + L_s^{-1}(e^{L_s \Delta t} - 1)\gamma(T-s, x_s) + L_s^{-2}\big[(e^{L_s \Delta t} - 1) - L_s \Delta t\big] M_s$$

$$+ \sigma \int_s^{s+\Delta t} e^{L_s(s+\Delta t - u)} \, dW_t \,.$$

Having established the general framework, we now specialize to our core case through the parameterization: set $\gamma(T-t, x) = \dfrac{1}{2}x + \nabla \log p_{T-t}(x)$, $\sigma = 1$, let $\Delta t \in [0, h]$ and $s = nh$.

In the resulting expression, we denote $x_s$ by $\vartheta_n^{\mathsf{SO}}$. Then for any $t \in [nh, (n+1)h]$, the solution admits the semi-analytic representation

$$x_t = \vartheta_n^{\mathsf{SO}} + \int_{nh}^t \big(\frac{1}{2}\vartheta_n^{\mathsf{SO}} + \nabla \log p_{T-nh}(\vartheta_n^{\mathsf{SO}}) \tag{44}$$

$$+ L_n(x_u - \vartheta_n^{\mathsf{SO}}) + M_n(u - nh)\big) \, du + \int_{nh}^t dW_u$$

where

$$L_n = \frac{1}{2}I_d + \nabla^2 \log p_{T-nh}(\vartheta_n^{\mathsf{SO}})$$

$$M_n = \frac{1}{2}\sum_{j=1}^d \frac{\partial^2}{\partial x_j^2} \nabla \log p_{T-nh}(\vartheta_n^{\mathsf{SO}}) - \frac{\partial}{\partial t} \nabla \log p_{T-nh}(\vartheta_n^{\mathsf{SO}}) \,.$$

Though $L_n$ and $M_n$ are theoretically defined through exact derivatives in SDE (43), their practical evaluation requires approximations due to the score function's computational intractability. We implement these approximations via numerical methods or neural networks, with concrete techniques for $L_n$ and $M_n$ estimation provided separately in Appendix F and G. Substituting these approximations into the SDE (44) yields

$$x_t = \vartheta_n^{\mathsf{SO}} + \int_{nh}^t \Big(\gamma(T-nh, \vartheta_n^{\mathsf{SO}}) + s_*^{(L)}(T-nh, \vartheta_n^{\mathsf{SO}})(x_u - \vartheta_n^{\mathsf{SO}})$$

$$+ s_*^{(M)}(T-nh, \vartheta_n^{\mathsf{SO}})(u-nh)\Big) \, du + \int_{nh}^t dW_u \,.$$

Crucially, this substitution preserves the closed-form integrability of the original framework. Adopting the same exponential integration strategy as above, we derive a closed form of $x_t$. Let $\vartheta_{n+1}^{\mathsf{SO}}$ denote $x_{(n+1)h}$, the second-order discretization scheme is given by

$$\vartheta_{n+1}^{\mathsf{SO}} = \vartheta_n^{\mathsf{SO}} + s_*^{(L)}(T-nh, \vartheta_n^{\mathsf{SO}})^{-1} \left(e^{s_*^{(L)}(T-nh, \vartheta_n^{\mathsf{SO}})h} - I_d\right) \left(\frac{1}{2}\vartheta_n^{\mathsf{SO}} + s_*(T-nh, \vartheta_n^{\mathsf{SO}})\right)$$

$$+ s_*^{(L)}(T-nh, \vartheta_n^{\mathsf{SO}})^{-2} \left(e^{s_*^{(L)}(T-nh, \vartheta_n^{\mathsf{SO}})h} - s_*^{(L)}(T-nh, \vartheta_n^{\mathsf{SO}})h - I_d\right) s_*^{(M)}(T-nh, \vartheta_n^{\mathsf{SO}})$$

$$+ \int_{nh}^{(n+1)h} e^{s_*^{(L)}(T-nh, \vartheta_n^{\mathsf{SO}})[(n+1)h - t]} \, dW_t \,.$$

Implementation specifics for handling the matrix exponentials and stochastic integral are addressed in Appendix G.

## F  Numerical Studies on Synthetic Data

We apply the five schemes to the posterior density of penalized logistic regression, defined by $p_0(\theta) \propto \exp(-f(\theta))$ with the potential function

$$f(\theta) = \frac{\lambda}{2}\|\theta\|^2 + \frac{1}{n_{\mathsf{data}}} \sum_{i=1}^{n_{\mathsf{data}}} \log(1 + \exp(-y_i x_i^\top \theta)) \,,$$

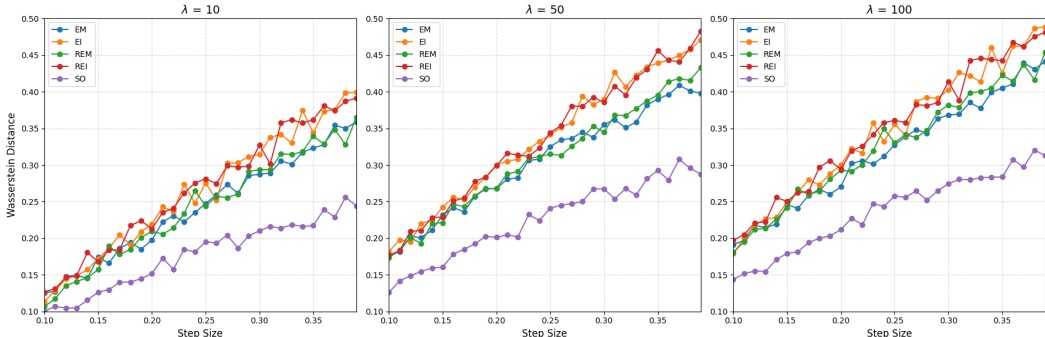

Figure 2: Error of various discretization schemes and second-order sampler with different choice of step size.

where $\lambda > 0$ denotes the tuning parameter. The data $\{x_i, y_i\}_{i=1}^{n_{\text{data}}}$, composed of binary labels $y_i \in \{-1, 1\}$ and features $x_i = (x_{i,1}, \cdots, x_{i,d})^\top \in \mathbb{R}^d$ generated from $x_{i,j} \overset{\text{i.i.d.}}{\sim} \mathcal{N}(0, 100)$.

### F.1 Implementation Details

In the numerical studies, we set $T = 10$, and the number of Monte Carlo iterations is chosen as the floor of $T/h$, where $h$ varies according to the step size indicated in the figure. Figure 2 shows the Wasserstein distance measured along the first dimension between the empirical distributions of the $N$-th outputs from SGMs and the target distribution, with different choices of the step size $h$. In this simulation, we use the Monte-Carlo method to estimate the score function and the Hessian matrix.

### F.2 Calculation

In this part, we derive explicit formulas for each coefficient term we need. First, the score function can be computed as

$$\nabla \log p_0(\theta) = -\left(\lambda\theta + \frac{1}{n_{\text{data}}} \sum_{i=1}^{n_{\text{data}}} \frac{-y_i x_i \exp(-y_i x_i^\top \theta)}{1 + \exp(-y_i x_i^\top \theta)}\right)$$

$$= -\left(\lambda\theta + \frac{1}{n_{\text{data}}} \sum_{i=1}^{n_{\text{data}}} \frac{-y_i x_i}{1 + \exp(y_i x_i^\top \theta)}\right).$$

For simplicity, we denote the logistic sigmoid function $\sigma(u) = \dfrac{1}{1 + e^{-u}}$, then

$$\nabla \log p_0(\theta) = -\left(\lambda\theta + \frac{1}{n_{\text{data}}} \sum_{i=1}^{n_{\text{data}}} -y_i x_i \sigma(-y_i x_i^\top \theta)\right).$$

Since $\sigma'(u) = \sigma(u)[1 - \sigma(u)]$, we have

$$\nabla^2 \log p_0(\theta) = -\left(\lambda I_d + \frac{1}{n_{\text{data}}} \sum_{i=1}^{n_{\text{data}}} y_i^2 \sigma(-y_i x_i^\top \theta)\left[1 - \sigma(-y_i x_i^\top \theta)\right] x_i x_i^\top\right)$$

$$= -\lambda I_d - \frac{1}{n_{\text{data}}} \sum_{i=1}^{n_{\text{data}}} \sigma(-y_i x_i^\top \theta)\left[1 - \sigma(-y_i x_i^\top \theta)\right] x_i x_i^\top.$$

As $x_i x_i^\top \succcurlyeq 0$, $\nabla^2 \log p_0(\theta) \preccurlyeq -\lambda I_d$. We also have that $\sigma(1 - y_i x_i^\top \theta) \in (0, 1)$, then

$$\nabla^2 \log p_0(\theta) \succcurlyeq -\lambda I_d - \frac{1}{4n_{\text{data}}} \sum_{i=1}^{n_{\text{data}}} x_i x_i^\top$$

$$\succcurlyeq -\left(\lambda + \frac{1}{n_{\text{data}}} \lambda_{\max}(\sum_{i=1}^{n_{\text{data}}} x_i x_i^\top)\right) I_d.$$

Therefore,

$$m_0 = \lambda, \quad L_0 = \lambda + \frac{1}{n_{\text{data}}}\lambda_{\max}\left(\sum_i^{n_{\text{data}}} x_i x_i^\top\right).$$

Recall that the transition probability $p_{t|0}(\theta_t|\theta_0) = \phi(\theta_t; \mu_t, \Sigma_t)$, where $\mu_t = e^{-\frac{1}{2}t}\theta_0, \Sigma_t = (1 - e^{-t})I_d$, and $\phi(\theta, \mu, \Sigma)$ denotes the probability density function of $\mathcal{N}(\mu, \Sigma)$, then we have

$$
\begin{aligned}
p_t(\theta_t) &= \int_{\mathbb{R}^d} p_{t|0}(\theta_t|\theta_0)p_0(\theta_0)\, \mathrm{d}\theta_0 \\
&= \int_{\mathbb{R}^d} \frac{1}{\sqrt{(2\pi)^d|\Sigma_t|}} \exp(-\frac{1}{2}(\theta_t - \mu_t)^\top\Sigma_t^{-1}(\theta_t - \mu_t))p_0(\theta_0)\, \mathrm{d}\theta_0 \\
&= \int_{\mathbb{R}^d} \frac{1}{[2\pi(1 - e^{-t})]^{d/2}} \exp(-\frac{1}{2(1 - e^{-t})}\|\theta_t - e^{-\frac{1}{2}t}\theta_0\|^2)p_0(\theta_0)\, \mathrm{d}\theta_0 \\
&= \frac{1}{[2\pi(1 - e^{-t})]^{d/2}}\mathbb{E}_{\theta_0 \sim p_0}\left[\exp(-\frac{1}{2(1 - e^{-t})}\|\theta_t - e^{-\frac{1}{2}t}\theta_0\|^2)\right].
\end{aligned}
$$

Hence,

$$
\begin{aligned}
\nabla p_t(\theta_t) &= \frac{1}{[2\pi(1 - e^{-t})]^{d/2}}\mathbb{E}_{\theta_0 \sim p_0}\left[\nabla\left(\exp(-\frac{1}{2(1 - e^{-t})}\|\theta_t - e^{-\frac{1}{2}t}\theta_0\|^2)\right)\right] \\
&= \frac{1}{[2\pi(1 - e^{-t})]^{d/2}}\mathbb{E}_{\theta_0 \sim p_0}\left[\exp(-\frac{1}{2(1 - e^{-t})}\|\theta_t - e^{-\frac{1}{2}t}\theta_0\|^2)\cdot\frac{-(\theta_t - e^{-\frac{1}{2}t}\theta_0)}{1 - e^{-t}}\right], \\
\nabla^2 p_t(\theta_t) &= \frac{1}{[2\pi(1 - e^{-t})]^{d/2}}\mathbb{E}_{\theta_0 \sim p_0}\left[\exp(-\frac{1}{2(1 - e^{-t})}\|\theta_t - e^{-\frac{1}{2}t}\theta_0\|^2)\right. \\
&\qquad\qquad\left.\cdot\left(\frac{(\theta_t - e^{-\frac{1}{2}t}\theta_0)(\theta_t - e^{-\frac{1}{2}t}\theta_0)^\top}{(1 - e^{-t})^2} - \frac{1}{1 - e^{-t}}I_d\right)\right].
\end{aligned}
$$

We can approximate $p_t(\theta_t), \nabla p_t(\theta_t)$ and $\nabla^2 p_t(\theta_t)$ or even higher order derivative tensor of $p_t(\theta_t)$ by Monte Carlo method, therefore, we can compute score function and its high order derivative by

$$\nabla \log p_t(\theta_t) = \frac{\nabla p_t(\theta_t)}{p_t(\theta_t)}, \quad \nabla^2 \log p_t(\theta_t) = \frac{\nabla^2 p_t(\theta_t)}{p_t(\theta_t)} - \frac{\nabla p_t(\theta_t)\nabla p_t(\theta_t)^\top}{p_t(\theta_t)^2}.$$

## G   Real Data Analysis

### G.1   Implementation Details

We set the step size $h = 0.2$ and $N = 2/h$. We conducted experiments on an NVIDIA RTX 4060 GPU (16GB VRAM). The training process required 2 GPU hours over 100 epochs with a batch size of 32, using CUDA 12.4, PyTorch 2.4, and torchvision 0.20.0. Figure 3 shows the digits generated by five algorithms, using the same score functions. The execution times for the algorithms are as follows: EM method 2 hour 12 min 49 s, EI method 2 hour 12 min 50 s, REM method 2 hour 13 min 30 s, REI method 2 hour 13 min 47 s, SO method 2 hour 14 min 05 s.

### G.2   Score matching function for second order acceleration

For the MNIST dataset, we have demonstrated in the proof of Proposition 15 that computing third-order derivatives is unnecessary. Unlike existing high-order methods for estimating second-order scores [28], which require the joint training of score functions and Hessian matrices and consequently incur substantial computational overhead, our second-order algorithm avoids explicit computation of the Jacobian matrix. Furthermore, by employing Hessian-vector products (HVPs), we efficiently capture higher-order information, enabling our second-order acceleration method to achieve improved performance with reduced iteration complexity and manageable computational cost.

More specifically, in the experiments of the MNIST dataset, we construct a U-Net architecture incorporating time and label embeddings to train the score function, where the time embedding operates

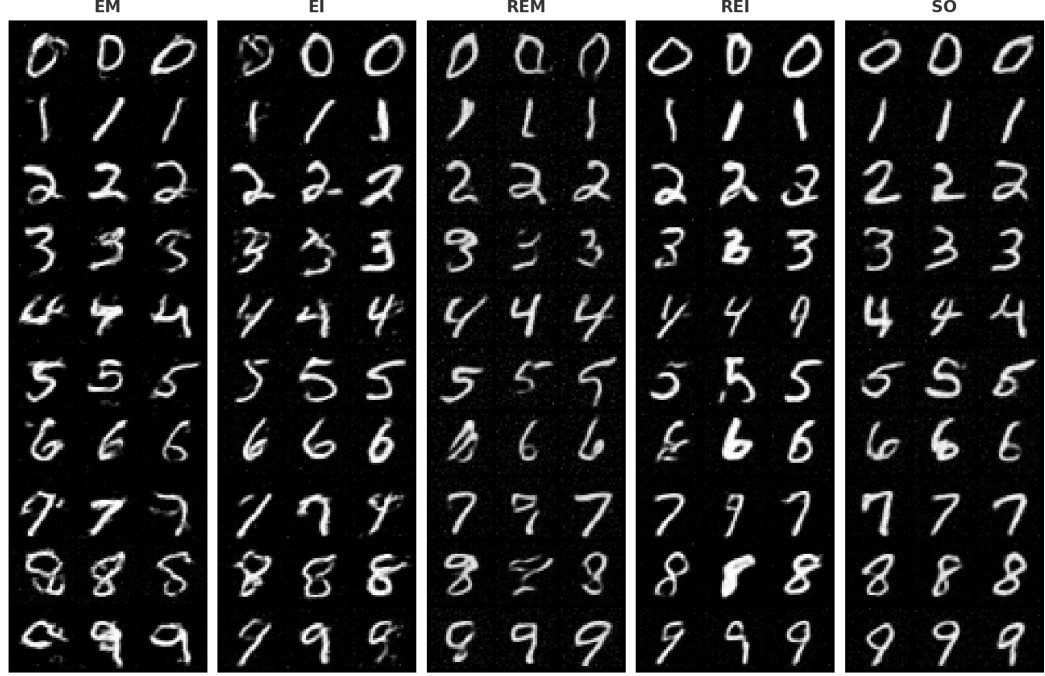

Figure 3: Comparative visualization of generated MNIST digits under various discretization schemes.

on the temporal variable $t$ of the score function, while the label embedding leverages MNIST's categorical digit labels. This conditional formulation expresses the score function as $\nabla \log p(t, x|\text{label})$, enabling per-class score estimation through discriminative embedding propagation.

Recall iteration rule of the SO algorithm, we have

$$\vartheta_{n+1}^{\text{SO}} = \vartheta_n^{\text{SO}} + s_*^{(L)}(T - nh, \vartheta_n^{\text{SO}})^{-1} \left( e^{s_*^{(L)}(T-nh, \vartheta_n^{\text{SO}})h} - I_d \right) \left( \frac{1}{2}\vartheta_n^{\text{SO}} + s_*(T - nh, \vartheta_n^{\text{SO}}) \right)$$

$$+ s_*^{(L)}(T - nh, \vartheta_n^{\text{SO}})^{-2} \left( e^{s_*^{(L)}(T-nh, \vartheta_n^{\text{SO}})h} - s_*^{(L)}(T - nh, \vartheta_n^{\text{SO}})h - I_d \right) s_*^{(M)}(T - nh, \vartheta_n^{\text{SO}})$$

$$+ \int_{nh}^{(n+1)h} e^{s_*^{(L)}(T-nh, \vartheta_n^{\text{SO}})[(n+1)h-t]} \, \mathrm{d}W_t \,.$$

Note that

$$\int_{nh}^{(n+1)h} e^{s_*^{(L)}(T-nh, \vartheta_n^{\text{SO}})[(n+1)h-t]} \, \mathrm{d}W_t \sim \mathcal{N}\left( 0, \frac{1}{2}s_*^{(L)}(T - nh, \vartheta_n^{\text{SO}})^{-1} \left( e^{2s_*^{(L)}(T-nh, \vartheta_n^{\text{SO}})h} - I_d \right) \right).$$

Let $g_n(\cdot) := s_*(T - nh, \cdot)$ denote the score matching function at time $T - nh$. Although the approximation of the Hessian matrix $\nabla^2 \log p_{T-nh}(\cdot)$ will not explicitly appear in the algorithmic implementation, we formally designate it as $H_n(\cdot)$ for notational clarity. Consequently, the estimators of $L_n$ and $M_n$ are chosen to be

$$s_*^{(L)}(T - nh, \vartheta_n^{\text{SO}}) := \frac{1}{2}I_d + H_n(\vartheta_n^{\text{SO}})$$

$$s_*^{(M)}(T - nh, \vartheta_n^{\text{SO}}) := -\frac{1}{2}g_n(\vartheta_n^{\text{SO}}) - H_n(\vartheta_n^{\text{SO}})(\frac{1}{2}\vartheta_n^{\text{SO}} + g_n(\vartheta_n^{\text{SO}}))\,.$$

Employing the Taylor expansion, we have

$$(s_*^{(L)})^{-1} \left( e^{hs_*^{(L)}} - I_d \right) = \sum_{k=1}^{\infty} \frac{h^k (s_*^{(L)})^{k-1}}{k!}\,,$$

$$(s_*^{(L)})^{-2} \left( e^{hs_*^{(L)}} - hs_*^{(L)} - I_d \right) = \sum_{k=2}^{\infty} \frac{h^k (s_*^{(L)})^{k-2}}{k!}\,,$$

$$\left[\frac{1}{2}(s_*^{(L)})^{-1}\left(e^{2hs_*^{(L)}} - I_d\right)\right]^{1/2} = \sqrt{h}\sum_{k=0}^{\infty} a_k(hs_*^{(L)})^k\,,$$

where $(a_0, a_1, a_2, a_3, a_4, a_5, a_6 \cdots) = (1, \frac{1}{2}, \frac{5}{24}, \frac{1}{16}, \frac{79}{5760}, \frac{3}{1280}, \frac{71}{193536}, \cdots)$. We thus reformulate all operators in the discretization scheme using matrix multiplications, which is a crucial step that avoids the explicit computation and storage of the full Hessian matrix $H_t$. This is achieved by leveraging Hessian-vector products (HVPs) via automatic differentiation, which reduces memory complexity to $\mathcal{O}(d)$ while retaining second-order curvature information. Specifically, given that $g_n$ corresponds to the neural network's output and $H_n$ represents its Jacobian matrix, we compute $H_n v$ for any vector $v$ through PyTorch's reverse-mode differentiation (`torch.autograd.grad`). By iteratively applying this HVP procedure $k$ times, we efficiently construct $H_n^k v$ for any $k \geqslant 0$. Through Taylor series expansion, these HVP-powerd computations enable precise evaluation of each term in the discretization scheme.

