# OpenReview forum: "Advancing Wasserstein Convergence Analysis of Score-Based Models: Insights from Discretization and Second-Order Acceleration"
_NeurIPS.cc/2025/Conference — NeurIPS 2025 poster_

### Official Review · Reviewer_mBSW · 2025-07-01

**Clarity:** 2
**Significance:** 3
**Originality:** 2
**Rating:** 5
**Confidence:** 3

**Summary:**

The paper studies the Wasserstein convergence of four different schemes for score-based diffusion models: Euler-Maruyama, exponential integrator, vanilla midpoint randomization, and exponential integrator with midpoint randomization. The paper observes that those two randomized midpoint methods do not give an improved convergence result because of the variance. The paper also proposed an accelerated sampler based on the Local Linearization Method. An approximation to the drift is designed according to Ito's formula, and Hessian estimation of the log density is employed. This leads to an improved dependence on the discretization $h$, so that only $O(\frac{1}{\varepsilon}\log(\frac{1}{\varepsilon}))$ is required to be $\varepsilon$ close to the target distribution.

**Questions:**

Questions:
- The theorems involve $m_{\min} - 1/2$, which is supposed to be positive number. I understand that $m_{\min} > 1/2$ is a simple condition for one-step contraction in the coupling argument. But I want to ask what if $m_{\min} > 1/2$ is not satisfied. Do you need to consider other reverse SDEs by changing the coefficient in front of the score in eq.(3) so that the drift in (3) will lead to contraction during coupling? Currently there is no remark on the fact that $m_{\min} > 1/2$ and I think maybe comments on this in the paper will be helpful to readers.
- Currently the complexity result (eg. Corollary 2 and Corollary 5) only shows the dependence on $\varepsilon$. Maybe the dependence on the dimension $d$ is also important especially for high dimensions. Ideally dependence on the condition number $L/m$ (or other counterparts such as $L(t)/m(t)$) is also interesting to people.
- For $\lVert X_0\rVert_{\mathbb{L}_2}$ in the bounds, if we assume $\log p_0$ is minimized at $x=0$, then it has the upper bound $\sqrt{d/m_0}$ (see for example, Lemma 4.0.1 in Sinho Chewi's book draft https://chewisinho.github.io/main.pdf). I want to ask why the authors left the term in the theorem rather than gave it a bound. I am also not sure whether such considerations could simplify/improve some bounds in the proof. (will the minimizer stays at $x=0$ for $p_t(x)$?)
- How tight are the bounds? For example, the authors can illustrate this when the target is a Gaussian.
- In Appendix G.2, implementation details are illustrated for the proposed scheme. How do you choose $k$ for the truncation of matrix exponentials?


Typos:
- page 8 footnote 2: the derivation is in Appendix E rather than F. Also, at the end of Appendix E, the matrix exponentials are addressed in Appendix G rather than F.
- Appendix D.1 is the proof of Lemma 8 rather than Lemma 7.
- References [13] and [14] are the same.

**Ethical Concerns:**

["NO or VERY MINOR ethics concerns only"]

**Final Justification:**

The authors have clearly answered all my questions. And the connection to existing works mentioned in other reviews and rebuttals helps me to better understand the current situation for this research direction. So I think the paper can be interesting in this field for future research.

**Limitations:**

yes

**Quality:**

3

**Strengths And Weaknesses:**

Strengths:
1. The paper uses a clear coupling framework to study the convergence of four different practical schemes in a Wasserstein distance.
2. An improved scheme using Hessian information is proposed, and the experiment on MNIST shows the effectiveness of the new scheme.

Weaknesses:
1. The analysis is kind of standard: using coupling technique and analyzing the contraction rate and the one-step discretization error. It is kind of incremental, but such a paper to illustrate and compare different discretization schemes could be helpful for people to refer to.
2. I am not sure how practical the newly proposed scheme is. For example, there is no experiment on CIFAR-10. I understand that this is a theory-focused paper, but I just want to know whether there is some implementation difficulty on CIFAR-10.

---

> ### Author Rebuttal · Authors · 2025-07-30
>
> We're grateful for the reviewer's thoughtful comments. Please find our detailed responses below.
>
>
> 1. >`The analysis is kind of standard: using coupling technique and analyzing the contraction rate and the one-step discretization error. It is kind of incremental, but such a paper to illustrate and compare different discretization schemes could be helpful for people to refer to.`
>
> We agree that the coupling-based proof technique used in our analysis of first-order samplers follows a well established structure. However, to our knowledge, a systematic comparison of different discretization schemes under the Wasserstein metric in constrained settings has not been previously studied. Applying these tools in this context provides new insights and clarifies the trade-offs between different methods.
>
> Moreover, the second-order sampler we propose, which is based on a local linearization approach, is new within the diffusion model literature. Although some elements of the analysis share similarities with first-order methods, the algorithm itself and the convergence guarantees we establish are original.
>
> 2. >`I am not sure how practical the newly proposed scheme is. For example, there is no experiment on CIFAR-10. I understand that this is a theory-focused paper, but I just want to know whether there is some implementation difficulty on CIFAR-10.`
>
>  The proposed scheme does not present any fundamental implementation difficulty on datasets such as CIFAR-10. The main challenge lies in computational overhead, primarily due to the increased cost of evaluating second-order information. However, with GPU acceleration and efficient techniques such as Hessian vector products, we believe the method remains feasible and can be implemented within a reasonable runtime. Exploring such high dimensional applications is an interesting direction for future empirical validation, building on the current theoretical foundation.
>
> 3. >`The theorems involve $m_min-1/2$, which is supposed to be positive number. I understand that
>  is a simple condition for one-step contraction in the coupling argument. But I want to ask what if $m_min>1/2$ is not satisfied. Do you need to consider other reverse SDEs by changing the coefficient in front of the score in eq.(3) so that the drift in (3) will lead to contraction during coupling? Currently there is no remark on the fact that $m_min>1/2$ and I think maybe comments on this in the paper will be helpful to readers.`
>
> We thank the reviewer for the helpful suggestion. The $-0.5$ term arises from the drift of the forward SDE, as shown in Lemma 8. In particular, the expression $m(T - s) - \frac{1}{2}$ follows from Line 723, where we use the identity
>
> $$
> \begin{aligned}
> \left\langle X_t, \frac{{\rm d} X_t}{{\rm d} t} \right\rangle = \left\langle X_t, \frac{1}{2} X_t + \nabla \log p_{T - t}(X_t) \right\rangle
> = \frac{1}{2} \\|X_t\\|^2 + \left\langle X_t, \nabla \log p_{T - t}(X_t) \right\rangle
> \leq \left(\frac{1}{2} - m(T - t)\right) \\|X_t\\|^2,
> \end{aligned}
> $$
>
> which reflects the influence of the drift term $-\frac{1}{2} X_t$ in the forward process.
>
> More generally, if the forward SDE is written as
>
> $$
> {\rm d} X_t = -\beta X_t {\rm d} t + {\rm d} B_t,
> $$
>
> then the bound becomes $m(t) - \beta$, showing explicit dependence on the coefficient $\beta$. While this still requires $p_{\textsf{data}}$ to be strongly log-concave, the condition $m_0 > \frac{1}{2}$ can be generalized to $m_0 > \beta$ for any $\beta > 0$.
>
> If the condition $m_{\min} > 1/2$ is not met, one can adjust the drift accordingly (e.g., using a smaller $\beta$), which does not affect the main results of the paper. We will add a remark clarifying this point in the revised manuscript.
>
> 4. >`Currently the complexity result (eg. Corollary 2 and Corollary 5) only shows the dependence on $\epsilon$. Maybe the dependence on the dimension $d$
>  is also important especially for high dimensions. Ideally dependence on the condition number $L/M$ (or other counterparts such as $L(t)/m(t)$) is also interesting to people.`
>
> We thank the reviewer for raising this point. We have explicitly quantified the dimensional dependence in the sample complexity for Theorems 1, 3, and 4 in our revision. For clarity, we summarize the results below:
>
> | | EM / EI / REM / REI | SO |
> |----| -------- | -------- |
> |Sample Complexity| $\tilde{\mathcal{O}}(d/\varepsilon^2)$ | $\tilde{\mathcal O}(d/\varepsilon)$  |
>
> Regarding the condition number, Theorems 1, 3, and 4 involve constants that depend on $L_{\max}$ and $m_{\min}$. In particular, the term $\frac{L_{\max} + 1/2}{m_{\min} - 1/2}$ can be interpreted as an analogue of the condition number when $m_{\min}$ is not too close to $1/2$. We will clarify this interpretation in the revised manuscript.
>
> 5. >`For $\|X_0\|_{\mathbb{L}_2}$ in the bounds, if we assume $\log p_0$ is minimized at $x=0$, then it has the upper bound $\sqrt{d/m_0}$. I want to ask why the authors left the term in the theorem rather than gave it a bound. I am also not sure whether such considerations could simplify/improve some bounds in the proof. (will the minimizer stays at $x=0$ for $p_t(x)$?)`
>
> We thank the reviewer for the thoughtful question. The term $\lVert X_0 \rVert_{\mathbb{L}_2}$ appears in our bounds as it reflects the dependence on the initialization and arises naturally in the standard decomposition of the Wasserstein error (see, for example, Theorem 5 in [13] and Theorem 2 in [12]).
> We chose to keep the term explicit in our theorems to maintain generality and avoid imposing potentially unnecessary assumptions on the form of $p_0$.
>
> Although assuming that $\log p_0$ is minimized at $x = 0$ enables the upper bound $\sqrt{d/m_0}$, this condition does not generally hold for $\log p_t(x)$ as time $t$ evolves. The forward SDE can shift the distribution, and without strong symmetry or centering assumptions, the mode of $p_t$ may move away from the origin. As a result, relying on such assumptions would reduce the generality of our analysis and may not yield significant simplifications in the proof. Therefore, we opt to make the dependence on $\lVert X_0 \rVert_{\mathbb{L}_2}$ explicit in our bounds.
>
>
> 6. >`How tight are the bounds? For example, the authors can illustrate this when the target is a Gaussian.`
>
> We thank the reviewer for the insightful question regarding the tightness of our bounds. To our knowledge, the only existing lower bound result in the Wasserstein distance is provided in [14], where the target distribution is assumed to be Gaussian. In this setting, the lower bound for the iteration complexity is shown to be of order $\Omega(\sqrt{d}/\varepsilon)$. As noted in [14], the same EM scheme is analyzed, and an upper bound with a convergence rate comparable to ours is established. This bound is also believed to be tight under the same assumptions (see [14] for further discussion). This suggests that the current upper bound may not be further improved without imposing additional structural conditions on the data distribution. Investigating whether such assumptions on $p_0$ could lead to improved upper bounds that match the lower bound in [14] remains an interesting direction for future research.
>
> 7. >`In Appendix G.2, implementation details are illustrated for the proposed scheme. How do you choose $k$ for the truncation of matrix exponentials?`
>
> We thank the reviewer for this helpful question. The choice of $k$ reflects a trade-off between computational cost and approximation accuracy. In our implementation, we use the Taylor expansion to approximate functions of $s_*^{(L)}$, but selecting an optimal $k$ is nontrivial, as we have limited knowledge of the error introduced by estimating $\nabla^2 \log p_{T - t}(x)$ using PyTorch’s autograd.
>
> However, we can still provide some guidance based on the Taylor expansion residual:
>
> $$
> \\|R_n(A)\\| \leq \frac{\|A\|^{n+1}}{(n+1)!} \sup_{t \in [0,1]} \\|f^{(n+1)}(A)\\|.
> $$
>
> For sufficiently smooth functions $f$, the residual decays roughly as $1 / (n+1)!$, indicating that higher-order terms contribute less.
>
> In practice, however, the overall score matching error $\varepsilon_{\text{sc}}$ is not arbitrarily small. It is lower bounded by the expressiveness of the neural network. In our U-Net training, the DSM loss plateaus around 0.01 after 50 epochs. This suggests that the total approximation error is dominated by training error rather than the truncation error from the matrix exponential.
>
> To balance efficiency and performance, we tested three values of $k$: 3, 5, and 10. We ultimately chose $k = 5$, as it performed similarly to $k = 10$ but significantly better than $k = 3$, offering a trade-off between accuracy and runtime.
>
> 8. Typos: We have corrected the typos in the revised version.

---

> > ### Comment · Reviewer_mBSW · 2025-08-05
> >
> > I want to thank the authors for the clear response. I do not have further questions for now. I also have a better understanding of the connection to related works by reading rebuttals for other reviewers. Therefore, I decide to support the paper by raising my score to 5.

---

> > > ### Author Response · Authors · 2025-08-09
> > >
> > > We are delighted that all your questions have been addressed, and we truly appreciate your positive feedback.

---

### Official Review · Reviewer_mdah · 2025-07-02

**Clarity:** 3
**Significance:** 2
**Originality:** 2
**Rating:** 4
**Confidence:** 3

**Summary:**

This work first studies the convergence of various discretization methods for the diffusion backward process, measured by the Wasserstein-2 distance. This work also derives a new discretization method based on the higher-order derivatives.

**Questions:**

1. Continuing Weakness 2b, What is special about the Wasserstein-2 distance?
2. In all theoretical results, a lot of coefficients are related to $m\_{\min}-0.5$. What is special about the $-0.5$ term? Does it pose an implicit restriction on $m_{\min}$ that $m_{\min}>0.5$?

**Ethical Concerns:**

["NO or VERY MINOR ethics concerns only"]

**Final Justification:**

I find the contributions of this work above the threshold of acceptance.

1. The first part about the characterization of the sampling error measured by W2 distance is new. Although the assumptions are quite strong, it is a minor drawback given that there have been few other works studying the W2 distance.
2. The proposed new method is an effective method that leverages the information of the hessian of $\log p$. The hessian-vector product avoids introducing significant computational overhead, and the empirical results are good.
3. The main drawback of the current version is that some discussions that I find important, including why the study of W2 distance is important, comparison with previous theoretical works (Weakness 2a), and comparison with other methods that use second-order information is inadequate. I am looking forward to improvements in the revision.

Given the strengths that overweigh weaknesses, I would like to recommend accepting this paper.

**Limitations:**

As is discussed in "Weaknesses", the experiments do not seem to be adequate. The paper does not seem to have any negative societal impact.

**Quality:**

2

**Strengths And Weaknesses:**

## Strengths

1. The theoretical results are presented in good logic, and the new discretization method is also derived with sound reasoning.
2. The assumptions are clearly stated and moderate.

## Weaknesses

1. As a whole, this work looks like a simple combination of two parts: one about the analysis of Wasserstein-2 distance, and the derivation of the new discretization. These two parts seem to be only loosely connected.
2. For the first part, the work seems to lack **detailed** comparison against previous works.
- 2a. Comparison against previous works on Wasserstein-2 distance: In the last paragraph of Section 3.1, results in [14] are mentioned. How do the results in this work compare against those in [14]?
- 2b. The comparison against results based on KL and TV are mentioned but not explained in detail. It is unclear whether the results about Wasserstein-2 distance imply different insights.
- 2c. The authors could also discuss the results in Li et al. (2024).

Li, Runjia, Di, Qiwei, and Gu, Quanquan. Unified Convergence Analysis for Score-Based Diffusion Models with Deterministic Samplers. 2024

3. For the second part, the experiments do not seem to be adequate.
- 3a. The experiments are only conducted on MNIST, which is a relatively easy task. I would suggest that the authors include more challenging tasks so that the possible instability in the estimation of higher-order derivatives can be evaluated.
- 3b. I would suggest that the authors include baselines based on ODE sampling, especially the Runge-Kutta discretization.
- 3c. A possible drawback of the algorithm is that the estimation of high-order derivatives may cause higher computational complexity. However, the results does

---

> ### Author Rebuttal · Authors · 2025-07-30
>
> We thank the reviewer for the feedback. Our responses to each point are provided below.
>
> 1. >`As a whole, this work looks like a simple combination of two parts: one about the analysis of Wasserstein-2 distance, and the derivation of the new discretization. These two parts seem to be only loosely connected.`
>
> While the paper may appear to have two separate components, both sections are closely connected through the shared goal of analyzing score-based diffusion models under the Wasserstein-2 ($W_2$) distance. The first part studies first-order discretization schemes and provides convergence guarantees in $W_2$. The second part builds on this by analyzing second-order methods, which to our knowledge have not been well explored in the context of Wasserstein convergence. The development of the second-order scheme leverages techniques and insights from the first-order analysis. Together, both parts contribute to a unified theoretical understanding of the Wasserstein behavior of discretized diffusion samplers.
>
> 2. >`Comparison against previous works on Wasserstein-2 distance: In the last paragraph of Section 3.1, results in [14] are mentioned. How do the results in this work compare against those in [14]?`
>
> The results in [14] focus on score-based diffusion models using the Euler discretization scheme and provide $W_2$ convergence guarantees under that setting. In our work, we compare our results with those of [14] immediately following Corollary 2 (line 176), highlighting the consistency between the results.
>
> 3. >`The comparison against results based on KL and TV are mentioned but not explained in detail. It is unclear whether the results about Wasserstein-2 distance imply different insights...What is special about the Wasserstein-2 distance?`
>
> Our work specifically focuses on convergence analysis under the $W_2$ distance, which is fundamentally different from KL and TV divergence in both its geometric structure and the techniques required for analysis. While KL and TV are widely used in prior work, they are not directly comparable to $W_2$, and convergence in one does not generally imply convergence in the others. For this reason, our analysis is tailored to the Wasserstein setting, and a direct comparison with KL or TV-based results may not be appropriate due to the fundamental differences between these metrics.
>
> 4. >`The authors could also discuss the results in Li et al. (2024). Li, Runjia, Di, Qiwei, and Gu, Quanquan. Unified Convergence Analysis for Score-Based Diffusion Models with Deterministic Samplers. 2024`
>
> We thank the reviewer for pointing out the recent work by Li et al. (2024), which provides a unified convergence analysis for deterministic samplers under the total variation distance. In contrast, our work studies stochastic samplers based on discretized SDEs and provides convergence guarantees under the Wasserstein-2 metric. The theoretical settings, assumptions, and proof techniques are thus quite different. While both works share the broader goal of understanding diffusion-based generative models, a direct comparison may not be appropriate due to these fundamental differences. We will add a brief mention of this work in the related literature to clarify the distinction.
>
> 5. >`The experiments are only conducted on MNIST, which is a relatively easy task. I would suggest that the authors include more challenging tasks so that the possible instability in the estimation of higher-order derivatives can be evaluated.`
>
> The numerical studies and MNIST-based experiments in our current submission are designed to illustrate the theoretical findings and validate the convergence behavior provided by our analysis. Since this work is submitted under the theory track, our primary goal is to establish and support theoretical guarantees rather than to optimize empirical performance on complex datasets. While we acknowledge that evaluating higher-order methods on more challenging datasets may be of practical interest, we believe such extensions are better suited for follow-up work focused on empirical exploration.
>
> 6. >`I would suggest that the authors include baselines based on ODE sampling, especially the Runge-Kutta discretization.`
>
> Our work focuses on stochastic samplers derived from SDE discretizations, with theoretical guarantees established in the Wasserstein-2 distance. In contrast, ODE-based methods correspond to deterministic samplers and are typically analyzed under different metrics, such as total variation or KL divergence. Since their sampling dynamics and theoretical frameworks differ fundamentally from ours, including them as baselines would fall outside the scope of our current analysis.
>
>
> 7. >`A possible drawback of the algorithm is that the estimation of high-order derivatives may cause higher computational complexity.`
>
> We thank the reviewer for raising this point. While high-order derivative computations can be computationally intensive in general, our implementation alleviates this issue by using Hessian–vector products (HVPs), which avoid explicitly forming or storing the full Hessian matrix. This strategy substantially reduces both time and memory costs, making the method practical in our experimental setting. Nevertheless, developing more efficient and accurate techniques for estimating higher-order derivatives of the score function remains an interesting and valuable direction for future work.
>
>
> 8. >`In all theoretical results, a lot of coefficients are related to $m_{min}-0.5$. What is special about the $-0.5$ term? Does it pose an implicit restriction on $m_{min}$ that $m_{min}>0.5$?`
>
> The $-0.5$ term arises from the drift term of the forward SDE, as demonstrated in Lemma 8. Specifically, the expression $m(T - s) - \frac{1}{2}$ follows from Line 723, where we use the fact that
>
> $$
> \begin{aligned}
> \left\langle X_t, \frac{{\rm d} X_t}{{\rm d} t} \right\rangle = \left\langle X_t, \frac{1}{2} X_t + \nabla \log p_{T - t}(X_t) \right\rangle
> = \frac{1}{2} \\|X_t\\|^2 + \left\langle X_t, \nabla \log p_{T - t}(X_t) \right\rangle
> \leq \left(\frac{1}{2} - m(T - t)\right) \\|X_t\\|^2,
> \end{aligned}
> $$
>
> which reflects the contribution of the drift term $-\frac{1}{2} X_t$ in the forward SDE.
>
> More generally, if the forward SDE takes the form
>
> $$
> {\rm d} X_t = -\beta X_t  {\rm d} t + {\rm d} B_t,
> $$
>
> then the bound becomes $m(t) - \beta$, reflecting a dependence on the coefficient $\beta$. While this setup still requires the data distribution $p_{\textsf{data}}$ to be strongly log-concave, the condition $m_0 > \frac{1}{2}$ can be generalized to $m_0 > \beta$ for any $\beta > 0$.

---

> > ### Comment · Reviewer_mdah · 2025-08-02
> >
> > Many thanks for the rebuttal! However, I still think some of the explanations are inadequate.
> >
> > 2. Comparisons with [14] do not contain enough details. I am expecting to see the mathematical formulation of the results of [14] and how each term corresponds to the result of the manuscript.
> >
> > 3. I am expecting a simple explanation of how W2 distance differs from KL and TV. An intuition/heuristic, or an innovative example is good.
> >
> > 6. I was suggesting a comparison against higher-order ODE methods because if the model has access to higher-order derivatives, then it should naturally outperform those without such access. Therefore, it could be more fair to compare your algorithm to compare against “higher-order” methods. I don’t recall any other SDE-based methods that leverage higher-order information, but there are plenty of ODE-based methods that do, and Runge-Kutta is the most important one. I wasn’t expecting a theoretical comparison against Runge-Kutta, but I think empirical comparisons should be possible.

---

> ### Author Response · Authors · 2025-08-03
>
> We thank the reviewer for the clarification of the questions. Please find our point-by-point response below.
>
> 2. >`Comparisons with [14] do not contain enough details. I am expecting to see the mathematical formulation of the results of [14] and how each term corresponds to the result of the manuscript.`
>
> Below we provide a detailed comparison between Theorem 1 in our paper and Theorem 5 of \[14], including a term by term correspondence.
>
> Since \[14] focuses on the Euler–Maruyama method, we specialize their Theorem 5 to match our setting by setting $f(t) = \frac{1}{2}$ and $g(t) = 1$ in their equations (3.8)–(3.14). This yields the following bound:
>
> $$
> \begin{aligned}
> \mathcal{W}\_2(\mathcal{L}(y_N), p_0) &\leq e^{-\int_0^{N\eta} m(t)\\, \mathrm{d}t} \\|x_0\\|\_{\mathbb{L}\_2} + \sum_{k=1}^N \prod_{j=k+1}^N \left(1 - \int_{(j-1)\eta}^{j\eta} (m(T - t) - \tfrac{1}{2})\\, \mathrm{d}t + M_1 \eta^2 \right) \\\\
> &\quad \cdot \Bigg( M_1 \eta^2 \left(1 + \\|x_0\\|\_{\mathbb{L}\_2} + \sup_{0 \leq t \leq T} \left( e^{-t} \\|x_0\\|\_{\mathbb{L}\_2}^2 + d(1 - e^{-t}) \right)^{1/2} \right) \\\\
> &\qquad + \varepsilon_{sc} \eta + \sqrt{\eta} h_{k,\eta} \left( \int_{(k-1)\eta}^{k\eta} \left(\tfrac{1}{2} + L(T - t)\right)^2\\, \mathrm{d}t \right)^{1/2} \Bigg),
> \end{aligned}
> $$
>
> where
>
> $$
> \begin{aligned}
> h_{k,\eta} &= e^{-T/2} \cdot e^{-\int_0^T (m(t) - \tfrac{1}{2})\\, \mathrm{d}t} \\|x_0\\|\_{\mathbb{L}\_2} \cdot \int_{(k-1)\eta}^{k\eta} \left(\tfrac{1}{2} + L(T - s)\right)\\, \mathrm{d}s \\\\
> &\quad + \tfrac{1}{2} \eta \cdot \sup_{0 \leq t \leq T} \left( e^{-t} \\|x_0\\|\_{\mathbb{L}\_2}^2 + d(1 - e^{-t}) \right)^{1/2} + \sqrt{\eta d}.
> \end{aligned}
> $$
>
> This expression aligns with equations (20) and (21) in our paper. For simplicity and clarity, we simplify the coefficients and retain only the dominant terms. This yields the following cleaner version, consistent with both our Theorem 1 and the formulation in \[14]:
>
> $$
> \begin{aligned}
> \mathcal{W}\_2(\mathcal{L}(y_N), p_0) \leq e^{-\int_0^{N\eta} m(t)\\, \mathrm{d}t} \\|x_0\\|\_{\mathbb{L}\_2} + \sum_{k=1}^N \left( \prod_{j=k+1}^N e^{-\int_{(j-1)\eta}^{j\eta} (m(T - t) - \tfrac{1}{2})\\, \mathrm{d}t} \right) \\
> \cdot \left( \varepsilon_{sc} \eta + \eta \sqrt{d} \left( \int_{(k-1)\eta}^{k\eta} \left( \tfrac{1}{2} + L(T - t) \right)^2\\, \mathrm{d}t \right)^{1/2} \right).
> \end{aligned}
> $$
>
> Now recall equations (20) and (21) in our paper, which state that
>
> $$
> \mathcal{W}\_2(\mathcal{L}(y_N), p_0) \leq e^{-\int_0^{Nh} m(t)\\, \mathrm{d}t} \\|x_0\\|\_{\mathbb{L}\_2} + \sum_{k=0}^{N-1} \left( \prod_{j=k+1}^{N-1} r_j^{\sf EM} \right) \left( h^2 C_k^{\sf EM} + h^{3/2} \sqrt{d} C_1(k) + h \varepsilon_{sc} \right),
> $$
>
> where
>
> $$
> r_j^{\sf EM} = e^{-\int_{jh}^{(j+1)h}(m(T - t) - \tfrac{1}{2})\\, \mathrm{d}t} + h^2 (C_1(k)^2 + M_1).
> $$
>
> Since $h$ is small, the term containing $h^2$ is dominated by the other leading terms. By ignoring it and applying the inequality
>
> $$
> \sqrt{h} C_1(k) \leq \left( \int_{k\eta}^{(k+1)\eta} \left( \tfrac{1}{2} + L(T - t) \right)^2\\, \mathrm{d}t \right)^{1/2},
> $$
>
> then our result simplifies to
>
> $$
> \begin{aligned}
> \mathcal{W}\_2(\mathcal{L}(y_N), p_0) &\leq e^{-\int_0^{Nh} m(t)\\, \mathrm{d}t} \\|x_0\\|\_{\mathbb{L}\_2} \\\\
> &\quad + \sum_{k=1}^N e^{-\int_{kh}^{Nh} (m(T - t) - \tfrac{1}{2})\\, \mathrm{d}t} \left( h \varepsilon_{sc} + h \sqrt{d} \left( \int_{(k-1)h}^{kh} \left( \tfrac{1}{2} + L(T - t) \right)^2 \\, \mathrm{d}t \right)^{1/2} \right),
> \end{aligned}
> $$
>
> which matches the form of the bound in \[14].
>
> **Explanation of Each Term:**
>
> * The product term $e^{-\int_{jh}^{(j+1)h} (m(T - t) - \tfrac{1}{2})\\, \mathrm{d}t}$ reflects the contraction induced by the strong concavity of $\nabla \log p_{T - t}(x)$.
> * The term $h \varepsilon_{sc}$ accounts for the score matching error, which accumulates across discretization steps.
> * The $h^{3/2} \sqrt{d} C_1(k)$ term represents the discretization error from the Euler method.
>
> We hope this detailed comparison addresses the reviewer's concern and clarifies how our result connects to and improves upon the analysis in \[14].

---

> ### Author Response · Authors · 2025-08-03
>
> 3. >`I am expecting a simple explanation of how W2 distance differs from KL and TV. An intuition/heuristic, or an innovative example is good.`
>
> We thank the reviewer for the suggestion to provide an intuitive explanation of how the $W_2$ distance differs from KL divergence and total variation (TV) distance. Below, we include a concrete example adapted from [Arsenyan et al., 2025] that illustrates this distinction clearly.
>
> Let $P$ be the exponential distribution with parameter 1. Define a mixture distribution $P_n = (1 - \delta_n) P + \delta_n Q_n$, where $Q_n$ is the uniform distribution on $[n, n+2]$ and $\delta_n = 1/\sqrt{n}$. This construction ensures that $P_n$ is very close to $P$ in both TV and KL divergence for $n\geqslant 2$:
>
> $$
> d_{\mathrm{TV}}(P_n, P) \leqslant \delta_n = n^{-1/2},
> $$
>
> $$
> d_{\mathrm{KL}}(P \| P_n) = -\log(1 - \delta_n) \leqslant 2n^{-1/2}.
> $$
> Therefore, one could expect that $P_n$ serves as an excellent generative model for the target.
> However, despite this closeness in KL and TV, the mean and the varaince of $P_n$ grows rapidly:
>
> $$
> \mathbb{E}_{P_n}[X] = 1+ n\delta_n \geqslant n^{1/2}.
> $$
>
> $$
> \mathbb{E}_{P_n}[X^2] \geqslant 2(1 - \delta_n) + \delta_n n^2 \geqslant n^{3/2}.
> $$
>
> Thus, samples from $P_n$ will have means and variances that diverge from those of $P$ as $n\to\infty$, showing that small KL or TV distance does not imply closeness in moments.
>
> In contrast, the Wasserstein-2 distance does control differences in moments. In particular, for any pair of distributions $P$ and $Q$ defined on the same space, it holds that
>
> $$
> \\|\mathbb{E}\_P[X] - \mathbb{E}\_{Q}[X]\\| \leq W_2(P, Q)
> $$
>
> and
>
> $$
> \left\| \left(\mathbb{E}\_P[X^\top A X]\right)^{1/2} - \left(\mathbb{E}\_{Q}[X^\top A X]\right)^{1/2} \right\| \leq W_2(P, Q),
> $$
>
> for any positive semidefinite matrix $0 \preceq A \preceq I$. This illustrates why $W_2$ is better suited for assessing the quality of generative models in practice for statistical purposes, as it captures differences in the geometric structure of the distributions.
>
> **Reference:**
> Arsenyan, Vahan and Vardanyan, Elen and Dalalyan, Arnak. *Assessing the Quality of Denoising Diffusion Models in Wasserstein Distance: Noisy Score and Optimal Bounds*.  ArXiv 2506.09681.

---

> ### Author Response · Authors · 2025-08-03
>
> 4. >`I was suggesting a comparison against higher-order ODE methods because if the model has access to higher-order derivatives, then it should naturally outperform those without such access. Therefore, it could be more fair to compare your algorithm to compare against “higher-order” methods. I don’t recall any other SDE-based methods that leverage higher-order information, but there are plenty of ODE-based methods that do, and Runge-Kutta is the most important one. I wasn’t expecting a theoretical comparison against Runge-Kutta, but I think empirical comparisons should be possible.`
>
> We thank the reviewer for the suggestion to compare our method with higher-order ODE-based approaches such as those using Runge-Kutta integrators. However, we believe such a comparison is not meaningful in our context, either theoretically or empirically.
>
> As we mentioned in our previous response, our work focuses on second-order SDE-based sampling algorithms and provides convergence guarantees in Wasserstein distance under stochastic dynamics. While diffusion models can be formulated using either SDEs or their deterministic ODE counterparts, the two frameworks differ fundamentally in algorithmic structure, assumptions, and analytical tools. ODE-based samplers, including those using Runge-Kutta methods, simulate deterministic flows and are typically analyzed using metrics such as KL divergence or total variation distance. In contrast, our analysis is designed for stochastic processes and is not applicable to deterministic methods.
>
> Furthermore, although Runge-Kutta methods are widely used for solving ODEs due to their high-order accuracy, they are fundamentally designed for deterministic dynamics. While there are stochastic adaptations of these methods, such as stochastic Runge-Kutta methods, they differ significantly in structure and analysis from the deterministic versions (see discussion in [42]) and from our method. As such, Runge-Kutta integrators for ODEs are not suitable as baselines for evaluating an SDE-based sampler analyzed in Wasserstein distance.
>
> Regarding the suggestion to compare with other higher-order methods, we note that our paper already discusses relevant prior work. For example, [42] applies a stochastic Runge-Kutta method to accelerate SDE sampling, and [28] accelerates the stochastic DDPM sampler by leveraging precise score  and Hessian estimations of the log density, even for possibly non-smooth target distribution. However, these works consider different classes of samplers, adopt different assumptions, and analyze convergence in KL divergence or total variation distance rather than Wasserstein distance. In particular, both [42] and [28] focus on DDPM-type samplers, which differ from our setting both algorithmically and analytically. To our knowledge, no existing method provides a directly comparable baseline under the same theoretical setting and convergence criteria.
>
> We also agree with the reviewer’s point that access to higher-order information should lead to better performance. In our experiments, we compared our second-order method, which uses Hessian information, with a first-order baseline that does not. The results show a clear improvement from using second-order information, directly confirming the reviewer’s observation and supporting our theoretical findings.
>
> We hope this response clarifies why comparison with ODE-based methods is not appropriate for the scope and focus of our work.

---

> > ### Comment · Reviewer_mdah · 2025-08-05
> >
> > Many thanks for the reply! I am very satisfied with the comparison with previous works, and the example that shows the separation of Wassertein-2 distance from TV and KL. I have decided to raise my score.

---

> > > ### Author Response · Authors · 2025-08-09
> > >
> > > We are pleased to have addressed all your questions and greatly appreciate your positive feedback.

---

### Official Review · Reviewer_cN6S · 2025-07-02

**Clarity:** 4
**Significance:** 2
**Originality:** 3
**Rating:** 4
**Confidence:** 3

**Summary:**

This paper provides a theoretical convergence analysis of diffusion models in the Wasserstein-2 distance.
Specifically, the authors first conduct a comprehensive comparison of different discretization schemes, including EM method, exponential integrator (EI) and randomized midpoint methods.
These schemes are shown to have a similar convergence rate of $\mathcal{O}(\frac{1}{\epsilon^2} \log (\frac{1}{\epsilon}))$ in $W_2$ distance.
In addition to these first first-order discretization results, the authors introduce an accelerated sampler that leverages the Hessian estimation.
Theoretically, they prove that this second-order discretization scheme achieves a significantly improved convergence rate of $\mathcal{O}((\frac{1}{\epsilon} \log (\frac{1}{\epsilon}))$.

In the experiments, the proposed second-order accelerated method is compared with the above first-order discretization schemes on both a synthetic posterior of a penalized logistic regression and the MNIST data distribution.
The numerical results in Figure 1 confirm that the proposed second-order method converges faster, which aligns with the theoretical analysis.

**Questions:**

- My major question is about the relationship between the convergence rate and the dimension of the target distribution.
- In the MNIST experiments, the running times for all five methods are reported to be close (within a 2-minute range for a >2 hour run) in Appendix G.1.
Could the authors explain why the additional HVP computations in the second-order method did not lead to a noticeable increase in running time.

**Ethical Concerns:**

["NO or VERY MINOR ethics concerns only"]

**Final Justification:**

I agree with the authors' response regarding the stronger assumptions. Overall, they have also addressed my other concerns. I recommend acceptance of the manuscript.

**Limitations:**

yes

**Quality:**

2

**Strengths And Weaknesses:**

### Strength
- The paper provides a systematic comparison of multiple popular discretization schemes (EM, EI, REM, REI) in terms of $W_2$ distance.
- The use of Hessian-vector products in the real-data experiment like MNIST makes the second-order approach computationally feasible.

### Weaknesses
The theoretical results are comprehensive, but further analysis could help clarify their properties and underlying assumptions.
- In contrast to prior work [1], which relies on the relatively mild assumption of a bounded second moment for total variation (TV) divergence analysis, Assumption 1 imposes a stronger condition—strongly log-concavity. This assumption implies the Log-Sobolev inequality, an assumption that is generally not adopted in most recent theoretical results of diffusion models.
- For the data generation task, the paper would benefit from the inclusion of quantitative metrics.
To better align the experiments with the theoretical results, the authors could report the FID or $W_2$ distance on toy 2D tasks, adopting an approach similar to that shown in Table 1 and Table 2 of paper [2].

[1] Fan, Ying, and Kangwook Lee. "Optimizing ddpm sampling with shortcut fine-tuning." ICML 2023.

[2] Gupta, Shivam, Linda Cai, and Sitan Chen. "Faster Diffusion Sampling with Randomized Midpoints: Sequential and Parallel." ICLR 2025.

---

> ### Author Rebuttal · Authors · 2025-07-30
>
> We thank the reviewer for the constructive remarks. Our responses to each point are provided below.
>
> 1. >`In contrast to prior work [1], which relies on the relatively mild assumption of a bounded second moment for total variation (TV) divergence analysis, Assumption 1 imposes a stronger condition—strongly log-concavity. This assumption implies the Log-Sobolev inequality, an assumption that is generally not adopted in most recent theoretical results of diffusion models.`
>
> We thank the reviewer for this thoughtful observation. It is true that Assumption 1 (strong log-concavity) is stronger than the bounded second-moment assumption used in prior TV-based analyses such as [1]. However, our analysis focuses on Wasserstein convergence for score-based diffusion models on non-compact domains, which requires stronger assumptions to ensure stability.
>
> In particular, our analysis relies on the Wasserstein contraction properties of the reverse SDE to control discretization and score-matching errors. These contraction results critically depend on time-uniform strong convexity and smoothness of the score function $\nabla \log p_t$, which we derive from Assumption 1, as also demonstrated in [13].
>
> Some existing works avoid strong convexity by assuming compactly supported data and introducing early stopping with projection onto the support, e.g. [5]. In contrast, our setting allows for unbounded data distributions and thus necessitates a different technical framework.
>
> We agree that relaxing Assumption 1 is an important and challenging direction for future work. We will revise the manuscript to clarify the role of this assumption and contrast it more explicitly with assumptions used in TV-based analyses.
>
> 2. >`For the data generation task, the paper would benefit from the inclusion of quantitative metrics. To better align the experiments with the theoretical results, the authors could report the FID or $W_2$
>  distance on toy 2D tasks, adopting an approach similar to that shown in Table 1 and Table 2 of paper [2].`
>
> We thank the reviewer for the helpful suggestion. Figures 1 and 2 in our paper already report the $W_2$ distance for each scheme across different step sizes, aligning closely with our theoretical analysis. In the revised version, we will include the FID for the MNIST data generation task to provide a more comprehensive evaluation of sample quality.
>
> 3. >`My major question is about the relationship between the convergence rate and the dimension of the target distribution.`
>
> We have explicitly quantified the dimensional dependence in the sample complexity for Theorems 1, 3, and 4 in our revision. For clarity, we summarize the results below:
>
> | | EM / EI / REM / REI | SO |
> |----| -------- | -------- |
> |Sample Complexity| $\tilde{\mathcal{O}}(d/\varepsilon^2)$ | $\tilde{\mathcal O}(d/\varepsilon)$  |
>
> 4. >`In the MNIST experiments, the running times for all five methods are reported to be close (within a 2-minute range for a >2 hour run) in Appendix G.1. Could the authors explain why the additional HVP computations in the second-order method did not lead to a noticeable increase in running time.`
>
> While second-order methods do introduce additional computational overhead due to HVP computations, in our implementation this overhead remains moderate. Specifically, we use PyTorch’s automatic differentiation to compute HVPs, which requires an additional backward pass to apply the Hessian to a given vector. As a result, the total cost per step (forward + backward) is indeed higher than that of first-order methods, which is both reasonable and expected. Moreover, since our experiments did not use GPU acceleration, the observed overhead may appear more pronounced than it would in a GPU-accelerated setting.

---

### Official Review · Reviewer_PXDo · 2025-07-13

**Clarity:** 3
**Significance:** 3
**Originality:** 3
**Rating:** 4
**Confidence:** 4

**Summary:**

This paper established a Wasserstein convergence theory for the score-based diffusion model. It provided upper error bounds for several popular discretization methods: Euler-Maruyama, exponential integrator, and randomized midpoint methods. All discretization methods offer the same rate asymptotically and require $\tilde{O}(\epsilon^{-2})$ steps to reach $\epsilon$ accuracy in $W_2$ distance, though midpoint randomization enables parallel computation and reduces computation time. The author also provides an acceleration method when the second-order information is available, and slashes the total sample steps to $\tilde{O}(\epsilon^{-1})$. The theoretical contributions were confirmed by the experiments on synthetic data and the MNIST dataset.

**Questions:**

1. As you mentioned, some other works have also focused on the convergence of diffusion in $W_2$ distance. Can you make a brief comparison of your result and theirs?

2. As mentioned in Weakness 1, is there a way to drop the Lipshitz continuity of the score function when analyzing convergence in $W_2$ distance?

3. Can you comment on the dimension dependence of the convergence result? Is it $\widetilde{O}(d/\epsilon^2)$ for non-accelerated version and $\widetilde{O}(d/\epsilon)$ for accelerated version?

**Ethical Concerns:**

["NO or VERY MINOR ethics concerns only"]

**Final Justification:**

The paper is comprehensive and solid, but there are a few weaknesses, especially in acceleration part, that limits the implement of the method. Therefore. I decide to maintain my original positive score.

**Limitations:**

Yes

**Quality:**

3

**Strengths And Weaknesses:**

Strengths:

1. The paper delivers a thorough Wasserstein-2 convergence analysis across multiple discretisation schemes, filling a gap left by prior work that centred on KL/TV metrics and relied almost exclusively on Euler steps.

2. The author proposed a second-order acceleration method to improve the convergence rate. This is the first acceleration that only requires $\tilde{O}(\epsilon^{-1})$ sample steps to achieve $\epsilon$ accuracy in $W_2$ distance.

3. Overall, the paper is well written and easy to follow, and the result seems correct.

Weaknesses:

1. The theoretical guarantees hinge on strong assumptions—especially Assumptions 2 and 6—that may not hold for many non-Gaussian or multimodal data distributions. Recent TV-distance analyses (e.g., Li et al., 2024) achieve convergence without such restrictive conditions.

2. The accelerated sampler needs accurate Hessian–vector products of $\nabla^2\log p_t(x)$ at every diffusion step. Computing these in high dimensions (e.g., $64\times64$ ImageNet) may be impossible. Alternative accelerations that approximate Hessians via finite differences of score functions (again Li et al., 2024) to avoid this bottleneck. As a result, I think the assumptions for this paper can be further weakened if this strategy is applied (though Li et al., 2024, focus on the convergence of TV distance).

3. Although the authors mention existing works focus on analyzing the convergence of $W_2$ distance, no comparison was given. It would be better to include a brief comparison of the setting/convergence rate between this paper and previous work.

Typos:

1. References [13] and [14] are the same.
2. Page 16, Line 493, missing one right bracket ")" in the first line.

References:
1. Li et al., 2024, "Accelerating Convergence of Score-Based Diffusion Models, Provably".

---

> ### Author Rebuttal · Authors · 2025-07-30
>
> We thank the reviewer for the feedback. Please find our point by point response below.
>
> 1. >`The theoretical guarantees hinge on strong assumptions—especially Assumptions 2 and 6—that may not hold for many non-Gaussian or multimodal data distributions. Recent TV-distance analyses (e.g., Li et al., 2024) achieve convergence without such restrictive conditions...As mentioned in Weakness 1, is there a way to drop the Lipshitz continuity of the score function when analyzing convergence in $W_2$ distance?`
>
> We thank the reviewer for this insightful comment. We agree that the Lipschitz continuity of the score function (Assumption 2) is a strong condition. However, this assumption is currently necessary to control the discretization error of the continuous-time backward process when analyzing convergence in the $W_2$ distance, as also indicated in [13].
>
> Similarly, for the SO scheme, we require Assumption 6, which imposes Lipschitz continuity of the Hessian. The only relaxation we are aware of is given in [28], where the Hessian Lipschitz condition is assumed only at the initial distribution $p_0$, not along the entire path. However, [28] focuses on accelerating DDPM under access to higher-order derivatives and analyzes convergence in KL divergence, relying on tools fundamentally different from those used in our Wasserstein-based analysis.
>
> Indeed, unlike analyses in the TV or KL divergence, convergence in the $W_2$ distance is particularly sensitive to the regularity of the score function due to its intrinsic dependence on the geometry of the space. Nonetheless, we acknowledge the importance of relaxing these assumptions and will add a discussion of this issue and related directions in our revision.
>
> 2. >`The accelerated sampler needs accurate Hessian–vector products of $\nabla^2 \log p_t(x)$ at every diffusion step. Computing these in high dimensions (e.g., $64\times 64$ ImageNet) may be impossible. Alternative accelerations that approximate Hessians via finite differences of score functions (again Li et al., 2024) to avoid this bottleneck. As a result, I think the assumptions for this paper can be further weakened if this strategy is applied (though Li et al., 2024, focus on the convergence of TV distance).`
>
> We thank the reviewer for raising this important point. We agree that computing accurate Hessian–vector products at every diffusion step can be challenging in high-dimensional settings. In our current implementation, we use PyTorch's automatic differentiation to compute Hessian–vector products efficiently without explicitly forming the full Hessian, which makes the approach feasible in moderate dimensions. We also appreciate the reviewer's reference to the finite-difference approximation strategy discussed in Li et al. (2024). That work focuses on the acceleration of DDPM and DDIM samplers and establishes convergence guarantees under the total variation distance by approximating the Jacobian of the score function through finite differences. While their framework and target metric differ from ours, extending such approximation techniques to our stochastic setting and analyzing their effect under Wasserstein-2 convergence is indeed a compelling direction. We view this as promising future work that could potentially help relax some of our current assumptions while maintaining theoretical guarantees.
>
> 3. >`Although the authors mention existing works focus on analyzing the convergence of $W_2$ distance, no comparison was given. It would be better to include a brief comparison of the setting/convergence rate between this paper and previous work...As you mentioned, some other works have also focused on the convergence of diffusion in $W_2$ distance. Can you make a brief comparison of your result and theirs?`
>
> We thank the reviewer for the helpful suggestion. Our work is indeed motivated by the limited understanding of Wasserstein convergence for score-based diffusion models. To our knowledge, the only prior comparable theoretical results in $W_2$ distance are for the Euler–Maruyama (EM) scheme, as presented in [14,38]. This comparison is briefly discussed in line 176 of the main text. While other works also study the Wassertein convergence for EM scheme, they typically do so under assumptions that differ substantially from ours. In contrast, no existing theoretical results are available under settings similar to ours for the EI, REM, REI, or SO schemes, making our work the first to provide such an analysis. We will revise the manuscript to clarify this point and include a more detailed comparison with existing EM-based results to highlight the distinctions in assumptions and convergence behavior.
>
> 4. >`Can you comment on the dimension dependence of the convergence result? Is it $\tilde O(d/\epsilon^2)$ for non-accelerated version and $O(d/\epsilon)$ for accelerated version?`
>
> We have explicitly quantified the dimensional dependence in the sample complexity for Theorems 1, 3, and 4 in our revision. For clarity, we summarize the results below:
>
> | | EM / EI / REM / REI | SO |
> |----| -------- | -------- |
> |Sample Complexity| $\tilde{\mathcal{O}}(d/\varepsilon^2)$ | $\tilde{\mathcal O}(d/\varepsilon)$  |
>
>
> 5. Typos: We have corrected the typos in the revised version.

---

> > ### Comment · Reviewer_PXDo · 2025-08-02
> >
> > Thanks to the authors for their responses. I have no further questions for now.

---

### Official Review · Reviewer_hQEd · 2025-07-15

**Clarity:** 3
**Significance:** 2
**Originality:** 3
**Rating:** 4
**Confidence:** 4

**Summary:**

The authors analyze the effects of the Euler discretization, exponential integrator, midpoint randomization, and a novel exponential integrator with midpoint randomization scheme in the convergence of score-based diffusion models within the Wasserstein distance. They show that midpoint randomization introduces a variance term that cancels out the improved estimation, offering no benefit in convergence. The authors also propose an accelerated sampler that uses second-order information, achieving an improved convergence rate of $O(1/\epsilon)$ iterations. A numerical study is offered, showing that the accelerated sampler allows for smaller error than the four previous discretizations (among which the randomized midpoint and Euler schemes perform the best).

**Questions:**

Can this requirement of exact second-order information be relaxed for a $O(1/\epsilon)$ rate, as is the case for convergence results under the TV distance as in Li et al. (2024) and Li et al. (2025)?

This is a good paper. I am on the fence between a score of 4 and a score of 5. My concerns regarding whether the accelerated sampler is truly accelerated or not lead me to lean towards a score of 4, as this heavily affects the significancy of the paper. I will be happy to raise my score if the authors alleviate my concerns.

**Ethical Concerns:**

["NO or VERY MINOR ethics concerns only"]

**Limitations:**

Yes

**Quality:**

3

**Strengths And Weaknesses:**

## Strengths
**Analysis of various discretization schemes.**
The authors study four different discretization schemes, and qualify the differences in performance therein from a theoretical standpoint. This analysis is novel, and welcome to the community, representing a step forward in understanding the behavior of discretization schemes within diffusion model sampling.
A nice numerical experiment (for a theory paper) is provided, allowing for an empirical comparison of these discretization schemes in practice that complements the theory provided.

**Well-written.** The paper is very well-written. I commend the authors with respect to the clear amount of effort taken to present the results in a clear and coherent manner. Intuition with respect to the results and methodology is readily provided, and overall enhances the quality of the paper.

## Weaknesses
**Acceleration requires second-order acceleration.** Previous work suggests that this may not be necessary — Li et al. (2024) achieve a speedup for a deterministic sample in https://arxiv.org/pdf/2403.03852, though their guarantees are in the TV distance.
In fact, the guarantee provided by the “accelerated” stochastic sampler of that paper mirrors what I suspect might be the case here. The rate for the “accelerated” stochastic sampler within Li et al. (2024) was shown to be achieved by the vanilla DDPM sampler by another paper written by the same first author (Li et al. (2025)) a year later. It may be possible that the first-order sampler may be able to achieve the $O(1/\epsilon)$ rate, though as Gao et al. (2025) remark, this may require additional assumptions.
The authors may have missed this remark, as seen in their remarks within Appendix A. The authors state that the “accelerated SDE sampler” of Li et al. (2024) “leverages higher-order expansions of the conditional density to enhance efficiency”. Yet, the rate achieved by the stochastic sampler of Li et al. (2024) is worse than the DDPM rate achieved by Li et al. (2025).
The authors do not quantify the dimensional dependence within the sample complexity explicitly. This hurts readability and clarity.

## References
Li et al. (2024), Accelerating Convergence of Score-Based Diffusion Models, Provably

Li et al. (2025), O(d/T) Convergence Theory for Diffusion Probabilistic Models under Minimal Assumptions

Gao et al. (2025), Wasserstein Convergence Guarantees for a General Class of Score-Based Generative Models

---

> ### Author Rebuttal · Authors · 2025-07-30
>
> We appreciate the reviewer's feedback. Please see our detailed responses to each comment below.
>
> 1. >`Acceleration requires second-order acceleration. Previous work suggests that this may not be necessary — Li et al. (2024) achieve a speedup for a deterministic sample in https://arxiv.org/pdf/2403.03852, though their guarantees are in the TV distance. In fact, the guarantee provided by the “accelerated” stochastic sampler of that paper mirrors what I suspect might be the case here. The rate for the “accelerated” stochastic sampler within Li et al. (2024) was shown to be achieved by the vanilla DDPM sampler by another paper written by the same first author (Li et al. (2025)) a year later. It may be possible that the first-order sampler may be able to achieve the $O(1/\epsilon)$
>  rate, though as Gao et al. (2025) remark, this may require additional assumptions...Can this requirement of exact second-order information be relaxed for a $O(1/\epsilon)$ rate?`
>
> We first consider the result of Li et al. (2025), which establishes a sharper convergence rate for the vanilla DDPM sampler. Their improvement arises from a novel proof technique: instead of relying on the standard inequality ${TV}^2(p_{X_1},p_{Y_1})\leqslant \frac{1}{2}{KL}(p_{X_1}\| p_{Y_1})$, they directly analyze the total variation (TV) distance. This direct control of TV allows for a tighter bound. Specifically, they introduce three auxiliary reverse processes to decompose the total error into estimation error and discretization error, with the improved rate stemming from tighter control of the latter. However, the analysis does not consider second-order samplers under the same proof framework, so it cannot be used to conclude that second-order information is ineffective. In fact, whether similar improvements can be achieved for second-order methods under this framework remains an open and interesting question.
>
> In our work, within the framework of Wasserstein distance and score-based diffusion models, both second-order information and smoothness assumptions contribute to improving the convergence rate. Our analysis adopts a unified proof skeleton for both first-order and second-order methods, introducing auxiliary processes to decompose and control the discretization error. The incorporation of second-order information plays a clear and essential role in reducing the discretization error in the backward dynamics. This improvement is not an artifact of the proof technique but reflects the numerical behavior of the sampler. Therefore, regardless of how the proof is structured, the benefit of second-order information remains fundamental and cannot be dismissed. Meanwhile, the smoothness assumptions are necessary to control the behavior of the time-dependent score function $s_t(x)$ for $t\in[0,T]$, which plays a central role in the backward dynamics and differs from the DDPM setting. While the optimal convergence rate in Wasserstein distance remains unknown, our results show that second-order acceleration is indeed effective in reducing the error introduced by discretizing the drift term in the backward process.
>
>
> 2. >`The authors may have missed this remark, as seen in their remarks within Appendix A. The authors state that the “accelerated SDE sampler” of Li et al. (2024) “leverages higher-order expansions of the conditional density to enhance efficiency”. Yet, the rate achieved by the stochastic sampler of Li et al. (2024) is worse than the DDPM rate achieved by Li et al. (2025). `
>
> We thank the reviewer for pointing out this important clarification. In Appendix A, our use of the term "accelerated SDE sampler" for Li et al. (2024) was intended to reflect the authors' design motivation, which involves higher-order expansions of the conditional density, rather than to claim superior convergence guarantees. We agree that, as noted in Li et al. (2025), the stochastic sampler proposed in Li et al. (2024) does not outperform the DDPM rate in terms of theoretical convergence. To avoid confusion, we will revise the wording in Appendix A to more accurately reflect this distinction and acknowledge the comparison raised by the reviewer.
>
> 3. `The authors do not quantify the dimensional dependence within the sample complexity explicitly. This hurts readability and clarity.`
>
>  We thank the reviewer for raising this point. We have explicitly quantified the dimensional dependence in the sample complexity for Theorems 1, 3, and 4 in our revision. For clarity, we summarize the results below:
>
> | | EM / EI / REM / REI | SO |
> |----| -------- | -------- |
> |Sample Complexity| $\tilde{\mathcal{O}}(d/\varepsilon^2)$ | $\tilde{\mathcal O}(d/\varepsilon)$  |

---

### Decision · Program_Chairs · 2025-09-17

**Decision:**

Accept (poster)

**Comment:**

This paper advances the theoretical understanding of score-based diffusion models by providing the first systematic Wasserstein convergence analysis across multiple discretization schemes (Euler, exponential integrator, randomized midpoint variants). A key contribution is the introduction of a second-order accelerated sampler leveraging Hessian information, shown to improve the convergence rate. Reviewers found the analysis rigorous and well written, and appreciated both the theoretical novelty and the supporting experiments on synthetic data and MNIST. While concerns remain regarding strong assumptions and practical feasibility of second-order methods, all reviewers ultimately leaned positive, and the consensus is that the paper makes a valuable theoretical contribution. I recommend acceptance as a poster.